# Isoginkgetin antagonizes ALS pathologies in its animal and patient iPSC models via PINK1-Parkin-dependent mitophagy

Ang Li[1,10], Sen Huang[2,10], Shu-qin Cao[3,10], Jinyi Lin[4], Linping Zhao [5], Feng Yu[1], Miaodan Huang [1], Lele Yang[1], Jiaqi Xin[1], Jing Wen[1], Lingli Yan[1], Ke Zhang[1], Maoyuan Jiang[1], Weidong Le[6], Peng Li[1], Yong U Liu[7], Dajiang Qin[5], Jiahong Lu [1], Guang Lu [4], Hanming Shen [8], Xiaoli Yao [2✉], Evandro F Fang [3,9✉] & Huanxing Su [1✉]

## Abstract

**Damaged mitochondria initiate mitochondrial dysfunction-associated senescence, which is considered to be a critical cause for amyotrophic lateral sclerosis (ALS). Thus, mitophagic elimination of damaged mitochondria provides a promising strategy in ALS treatment. Here, through screening of a large natural compound library ($n = 9555$), we have identified isoginkgetin (ISO), a bioflavonoid from *Ginkgo biloba*, as a robust and specific mitophagy inducer. ISO enhances PINK1–Parkin-dependent mitophagy via stabilization of the PINK1/TOM complex. In a translational perspective, ISO antagonizes ALS pathology in *C. elegans* and mouse models; intriguingly, ISO improves mitochondrial function and antagonizes motor neuron pathologies in three ALS patient-derived induced pluripotent stem cell systems (C9, SOD1, and TDP-43), highlighting a potential broad application to ALS patients of different genetic background. At the molecular level, ISO inhibits ALS pathologies in a PINK1–Parkin-dependent manner, as depletion or inhibition of PINK1 or Parkin blunts its benefits. These results support the hypothesis that mitochondrial dysfunction is a driver of ALS pathology and that defective mitophagy is a druggable therapeutic target for ALS.**

**Keywords** Amyotrophic Lateral Sclerosis; Drug Screening; Isoginkgetin; PINK1-Parkin; Mitophagy
**Subject Categories** Neuroscience; Pharmacology & Drug Discovery

## Introduction

The incidence of chronic neurodegenerative diseases increases with human age, and as the average age of human populations increases, the number of individuals affected by neurodegenerative diseases continues to increase (Fang et al, 2020; Hou et al, 2019). Amyotrophic lateral sclerosis (ALS) is characterized by the loss of motor neurons (MNs) in the brain and spinal cord, degeneration of corticobulbar/corticospinal tracts, and denervation of skeletal muscle, leading to progressive paralysis and death (Feldman et al, 2022; Larkin, 2022). Genetic heterogeneity in the ALS patient population, rare clinical morbidity, and the scarcity of postmortem samples from ALS patients have hindered progress in understanding the etiology, pathology, and disease progression of ALS.

Mitochondrial dysfunction occurs in ALS and contributes to disease progression (Amorim et al, 2022; Delic et al, 2018; Wilson et al, 2023), and cells with ALS pathogenic variants *SOD1, C9orf72,* and *TARDBP* (encoding TDP-43 protein) show mitochondrial defects (Van Daele et al, 2023). In the case of TDP-43 in ALS, mitochondrial dysfunction causes inflammation via release of mitochondrial DNA to the cytoplasm, followed by activation of the cGAS/STING-based inflammation pathway; inhibiting STING reduces inflammation and neurodegeneration in in vitro and in vivo models of ALS (Yu et al, 2020). These results suggest the possibility that mitochondrial dysfunction could be a therapeutic target for ALS. Mitophagy is a cellular process that removes and/or recycles damaged or superfluous mitochondrial components through a mechanism involving lysosomal degradation, playing critical roles in mitochondrial homeostasis, cell survival, and organismal longevity (Picca et al, 2023; Pickles et al, 2018). Mitophagy operates via two distinct but interconnected mechanisms: the ubiquitin-dependent pathway and the receptor-mediated

[1]State Key Laboratory of Mechanism and Quality of Chinese Medicine, Institute of Chinese Medical Sciences, University of Macau, Macao, China. [2]Department of Neurology, The First Affiliated Hospital, Sun Yat-sen University; Guangdong Provincial Key Laboratory of Diagnosis and Treatment of Major Neurological Diseases; National Key Clinical Department and Key Discipline of Neurology, Guangzhou, China. [3]Department of Clinical Molecular Biology, University of Oslo and Akershus University Hospital, Lørenskog 1478, Norway. [4]Department of Physiology, Zhongshan School of Medicine, Sun Yat-sen University, Guangzhou, China. [5]The Fifth Affiliated Hospital of Guangzhou Medical University, Guangzhou, China. [6]Liaoning Provincial Key Laboratory for Research on the Pathogenic Mechanisms of Neurological Diseases, The First Affiliated Hospital of Dalian Medical University, Dalian, China. [7]Laboratory for Neuroimmunology in Health and Diseases, Centre for Medical Research on Innovation and Translation, Institute of Clinical Medicine, the Second Affiliated Hospital, School of Medicine, South China University of Technology, Guangzhou, China. [8]Faculty of Health Sciences, University of Macau, Macau, China. [9]The Norwegian Centre on Healthy Ageing (NO-Age) and the Norwegian National anti-Alzheimer's Disease (NO-AD) Networks, Oslo, Norway. [10]These authors contributed equally: Ang Li, Sen Huang, Shu-qin Cao.✉E-mail: yaoxiaol@mail.sysu.edu.cn; e.f.fang@medisin.uio.no; huanxingsu@um.edu.mo

pathway. The PTEN-induced kinase 1 (PINK1) and parkin RBR E3 ubiquitin protein ligase (Parkin) pathway is one of the best characterized ubiquitin-mediated mitophagy pathways (Youle and Narendra, 2011). In this pathway, full-length PINK1 (FL-PINK1) stabilizes on the outer mitochondrial membrane (OMM), forming a high molecular weight complex with the translocase of the outer mitochondrial membrane (TOM) (Lazarou et al, 2012), where it dimerizes and auto-activates its kinase function via autophosphorylation (Okatsu et al, 2012). Then, the activated PINK1 phosphorylates ubiquitin at Serine 65 (pSer65-Ub) and the ubiquitin-like domain of Parkin, which activates a positive feed-forward loop, initiating mitophagy (Kane et al, 2014). Two major types of receptors, BNIP3 and BNIP3L (also known as NIX), as well as FUNDC1, are involved in the receptor-mediated mitophagy pathways (Onishi et al, 2021). BNIP3 and NIX contain a LC3-interacting region (LIR) motif that can interact with LC3. Under hypoxic conditions, BNIP3 and NIX are upregulated and anchored to the OMM, thereby participating in mitophagy (Yamashita et al, 2024). FUNDC1 is an OMM protein that contains a typical LIR motif and can act as a receptor for hypoxia-induced mitophagy (Liu et al, 2012).

Defects in mitophagy are associated with mitochondrial dysfunction such as oxidative stress and bioenergetic defects in Alzheimer's disease (AD) and Parkinson's disease (PD) (Fang et al, 2019; Sliter et al, 2018). Many studies have reported that agents that stimulate mitophagy could antagonize disease progression in animal models with AD and PD (Moskal et al, 2020; Xie et al, 2022). Although defects in mitophagy have been observed in distinct ALS disease models (Magrané et al, 2014; Palomo et al, 2018), a causal link between defects in mitophagy and ALS pathology has not been established. Notably, defects in mitophagy-related genes including *optineurin*, *p62/SQSTM1*, *Valosin-containing protein*, and *TANK-binding kinase 1* are reported to be associated with ALS (Freischmidt et al, 2017; Koppers et al, 2012; Teyssou et al, 2013; Wong and Holzbaur, 2014). Impaired PINK1-Parkin-dependent mitophagy, characterized by the decreased expression of FL-PINK1, has been reported in various ALS disease models (Knippenberg et al, 2013; Rogers et al, 2017; Zhang et al, 2025). The specific molecular mechanism remains unknown, and no available drugs are currently known to regulate mitophagy in ALS by activating PINK1 expression or by enhancing the activity of FL-PINK1. MNs are characterized by large cell bodies, unusually long axons, and high sensitivity to defects in mitochondrial quality, function, and bioenergetics (Evans and Holzbaur, 2019). Therefore, it is important to study mitophagy in MNs from ALS patients and to explore whether pharmacological restoration of MN mitophagy has therapeutic potential for ALS. In this study, we identified a mitophagy inducer, isoginkgetin (ISO), a bioflavonoid from *Ginkgo biloba*, for its therapeutic potential in ALS.

# Results

## Screen for mitophagy inducers

Using previously reported YFP-Parkin-mt-mKeima HeLa cells co-expressing Parkin and pH-sensitive monomeric mtKeima (Xie et al, 2022) and the Operetta CLS PerkinElmer high-content imaging system to monitor mitophagy signals (Fig. EV1A), a pool of 9555 natural compounds (Dataset EV1) was subject to drug screening for potential mitophagy inducers (Fig. 1A). Carbonyl cyanide 3-chlorophenylhydrazone (CCCP), a well-characterized mitophagy inducer (Katayama et al, 2020), was used as a positive control. Cells were incubated with test compound (10 μM, 24 h) or CCCP (5 μM, 4 h). The mitophagy index of each compound was normalized to the CCCP control. The screen identified 284 candidate compounds (normalized mitophagy index >1). Of these, 20 compounds with low cytotoxicity were selected for further study (Figs. 1A and EV1B).

Many mitophagy inducers, including CCCP, depolarize mitochondria and/or impair mitochondrial function, making them unsuitable as therapeutic drugs (Georgakopoulos et al, 2017). Therefore, the 20 top candidate mitophagy inducers were scored for relative ATP production in galactose- and glucose-containing medium, and 15% lower ATP production in galactose was used as a cut-off (i.e., maximum tolerable evidence of mitochondrial toxicity). Based on this criterion, 8 compounds were selected for further study (Fig. 1B). ISO was identified for having the strongest ability to activate mitophagy in CCCP-depolarized mitochondria (Fig. 1C). Its molecular structure is shown in Fig. 1D; its 50% cytotoxic concentration ($CC_{50}$) was also measured (Fig. 1E; 16.45 μM). Immunoblot analysis showed that application of ISO (10 μM, 24 h) led to a significant degradation of the outer and inner mitochondrial membrane proteins Mitofusin 2 (MFN2) and mitochondrial cytochrome c oxidase subunit 2 (MTCO2) (Fig. 1F), and flow cytometry detected mt-mKeima as evidence of mitophagy in ISO-treated cells (Fig. EV1C,D), confirming that ISO is a potent mitophagy agonist.

We found that ISO treatment remarkably increased the expression of pSer65-Ub and Parkin self-ubiquitination (Fig. 1F). Live-cell imaging showed that ISO induced substantial Parkin translocation (Fig. EV1E,F), indicating enhanced Parkin E3 ligase activity. Accordingly, ubiquitination of MFN2, a known Parkin substrate, also increased in ISO-treated cells (Fig. 1F). For three receptor proteins, BNIP3, NIX and FUNDC1, ISO treatment did not cause any changes in their expression levels (Fig. 1F). After siRNA knockdown (KD) of *PINK1*, *BNIP3*, *NIX*, *FUNDC1* and *BNIP3/NIX*, respectively, in YFP-Parkin-mt-mKeima HeLa cells, ISO-induced mitophagy was abolished only under PINK1 knocked down conditions (Fig. EV1G–I). Consistently, the *PINK1* KD in YFP-Parkin HeLa cells blocks ISO treatment-induced expression of pSer65-Ub, ubiquitination/degradation of MFN2, and translocation of Parkin to mitochondria (Fig. 1G–I), suggesting that ISO specifically stimulates PINK1-Parkin-dependent mitophagy. Moreover, while CCCP caused nearly complete loss of mitochondrial membrane potential (MMP), MMP was only 5.65% lower in the presence of 10 μM ISO than in its absence (Fig. 1J).

## ISO promotes PINK1–Parkin-dependent mitophagy of CCCP-damaged mitochondria

We found that 10 μM ISO can induce mitophagy and enhance the CCCP-induced mitophagy signal. Interestingly, 5 μM ISO, even when it cannot actively induce mitophagy, is still sufficient to significantly enhance the mitophagy signals in the presence of external mitochondrial stress (Fig. 2A,B). We then tested whether 5 μM ISO can enhance CCCP-induced mitophagy through the PINK1–Parkin pathway. Live-cell imaging and immunoblotting

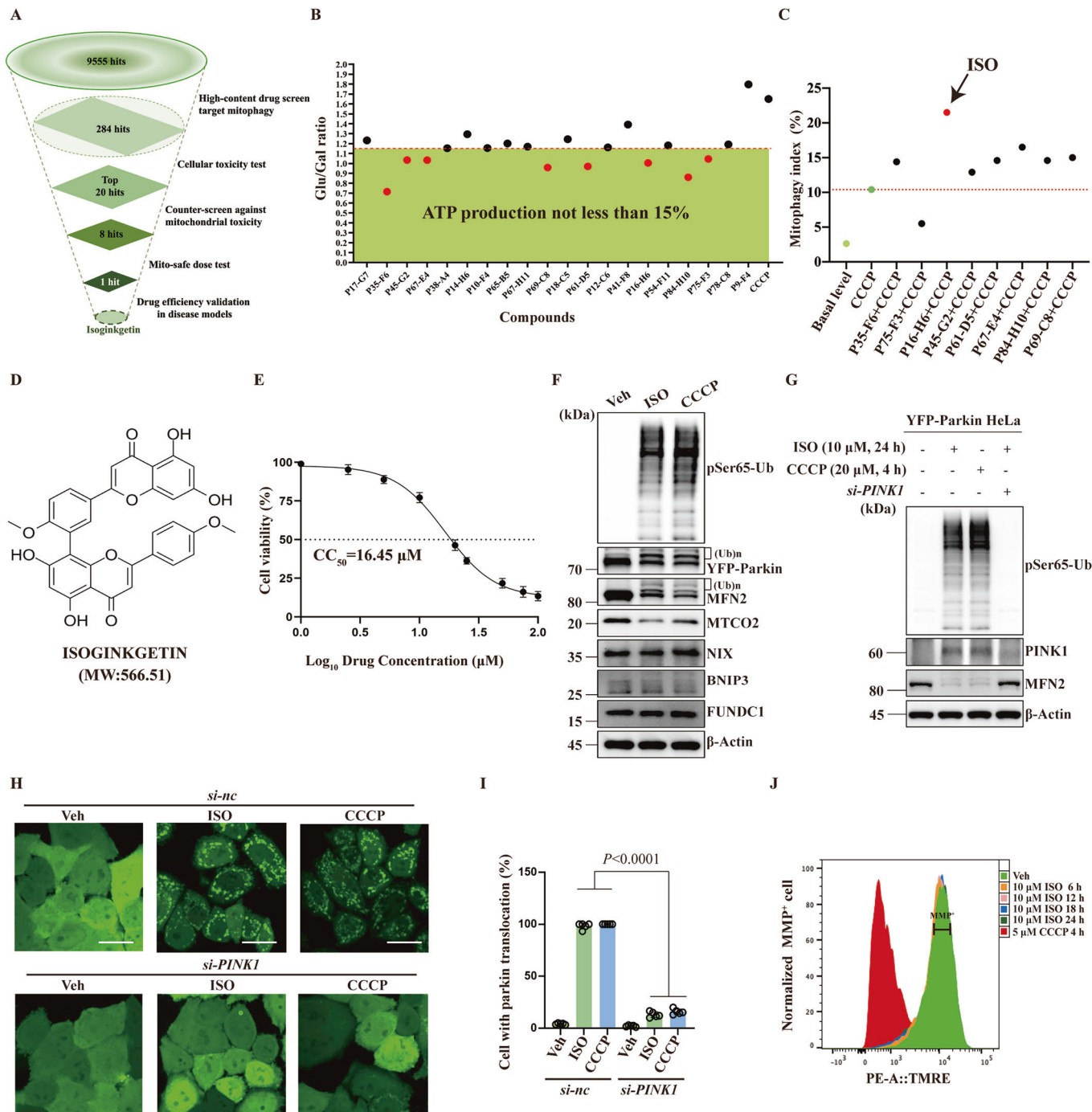

showed that ISO increased recruitment of Parkin to mitochondria and stimulated its degradation (Fig. 2C,D), increased abundance/expression of FL-PINK1 and pSer65-Ub, increased degradation of MFN2 and MTCO2, and increased LC3 lipidation (Fig. 2D). The same results were also found at different time points in YFP-Parkin HeLa (Fig. EV2A) and SH-SY5Y cells (Fig. EV2B). In addition, all these effects were strongly reduced by *PINK1* KD (Fig. 2E).

To further explore whether ISO participates in the activated receptor-mediated mitophagy pathway, we worked on receptor-mediated mitophagy using a hypoxia system. We have successfully established this system as evidenced by the high expression of hypoxia-inducible factor 1α (HIF-1α) and significant degradation of mitochondrial membrane proteins MFN2 and MTCO2 (Fig. EV2C,D). However, ISO did not increase the expression of BNIP3 and NIX or change FUNDC1 in the hypoxic YFP-Parkin HeLa cells (Fig. EV2C), which suggests that ISO does not enhance the receptor-mediated mitophagy. The same experiment was performed in SH-SY5Y cells with similar results (Fig. EV2D). Under normoxic conditions, 5 μM ISO did not increase the expression of receptor proteins in either of the two cell types. Thus, all these results suggest that 5 μM ISO can specifically

◄ **Figure 1. Identification of small-molecule compound ISO as a PINK1–Parkin-dependent mitophagy inducer.**

(**A**) Schematic of the screening funnel used to discover ISO from a total of 9555 hits. (**B**) Top 20 compounds were counter-screened for mitochondrial toxicity in a galactose/glucose assay. After 24 h, the retention of cells in glucose or galactose medium was quantified using ATP assays. The Glu/Gal ratio was plotted for each compound (10 μM), among which eight compounds were considered "mito-safe". (**C**) Flow cytometry was used to detect mitophagy signals in YFP-Parkin-mt-mKeima HeLa cells under co-treatment conditions. The level of mitophagy was plotted for each compound. Among the eight mito-safe compounds, ISO exhibited the strongest mitophagy activity. (**D**) The structure and molecular weight of ISO. (**E**) YFP-Parkin HeLa cells were treated with ISO at various concentrations for 24 h. CCK8 assay was used to detect cell viability ($n = 4$; four biological repeats). (**F**) YFP-Parkin HeLa cells were treated with ISO (10 μM) for 24 h. Immunoblots of the indicated proteins. Treatment with 20 μM CCCP (4 h) served as a positive control. (**G**) The si-PINK1 was transfected into YFP-Parkin HeLa cells with Lipo3000. Western blotting showed that knockdown of PINK1 inhibited the expression of pSer65-Ub and outer mitochondrial membrane protein (MFN2) degradation induced by ISO (10 μM). (**H**) The si-PINK1 was transfected into YFP-Parkin HeLa cells with Lipo3000. A live-cell imaging assay was used to monitor Parkin translocation. Scale bars, 20 μm. (**I**) Quantification of (**H**) showed that Parkin translocation induced by ISO was inhibited by knockdown of PINK1 ($n = 5$; five biological repeats). (**J**) YFP-Parkin HeLa cells were treated with 10 μM ISO, and the MMP levels were measured using flow cytometry with the TMRE probe every 6 h. CCCP treatment rapidly caused a dramatic decrease in MMP, but ISO treatment (10 μM) caused only a very marginal decrease in MMP at 24 h. Data are presented as the mean ± SD. Exact P values are reported in Appendix Table S1. One-way ANOVA followed by Dunnett's multiple comparisons test. Source data are available online for this figure.

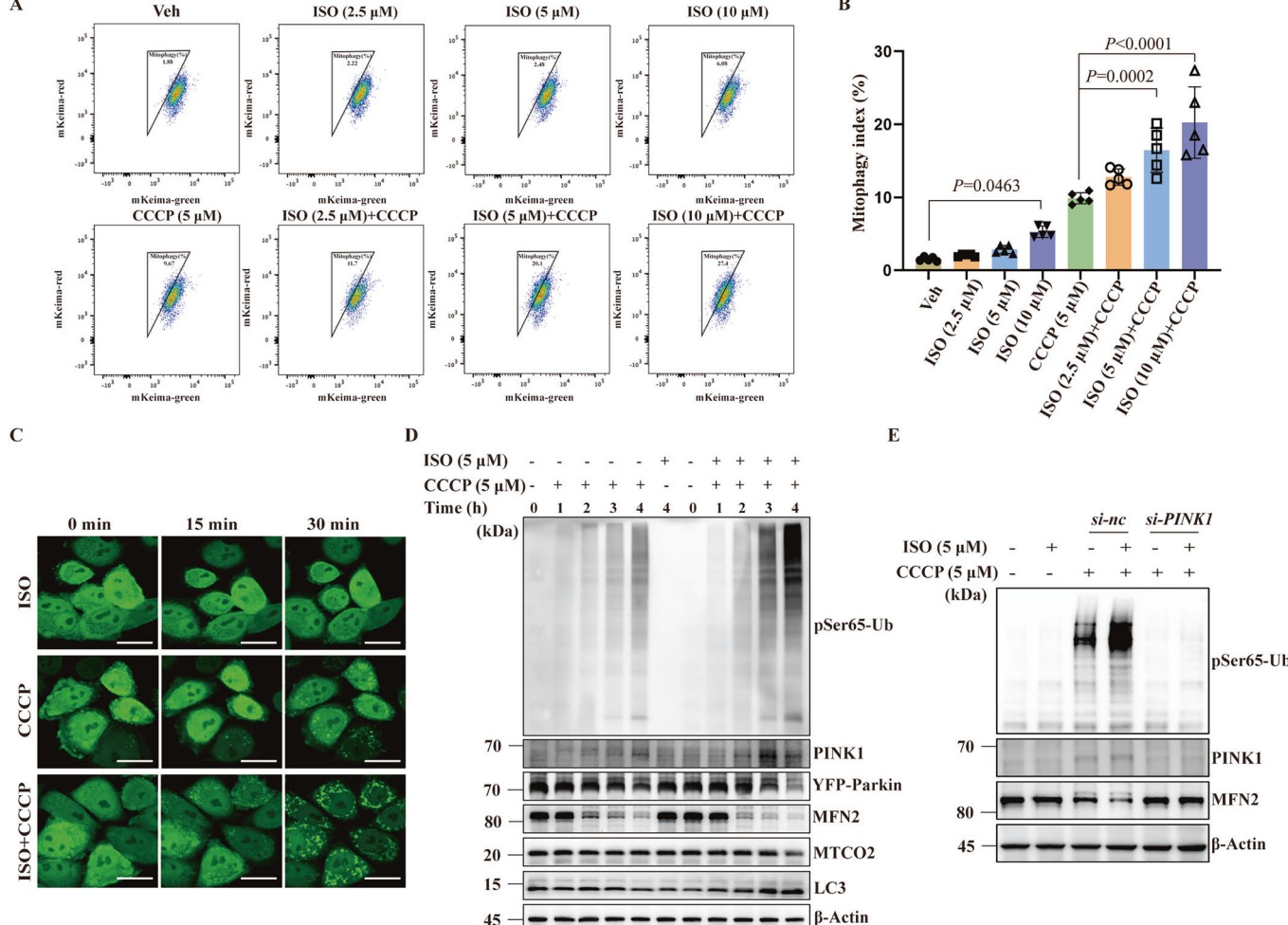

**Figure 2. ISO promotes PINK1–Parkin-dependent mitophagy under CCCP stress.**

(**A**) Flow cytometry was used to detect mitophagy signals in YFP-Parkin-mt-mKeima HeLa cells under the co-treatment of ISO (2.5, 5, 10 μM) and CCCP (5 μM) for 8 h. (**B**) Quantification of (**A**) showed that ISO promoted CCCP-induced mt-mKeima-induced mitophagy signals in a dose-dependent manner ($n = 5$; five biological repeats). (**C**) Representative image of Parkin translocation in YFP-Parkin HeLa cells. Scale bar, 25 μm. (**D**) YFP-Parkin HeLa cells were treated with CCCP (5 μM), CCCP + ISO (5 μM + 5 μM), and ISO (5 μM, 4 h), respectively. The expression level of PINK1-Parkin pathway proteins and mitochondrial membrane proteins was detected using immunoblotting. (**E**) The si-PINK1 was transfected into YFP-Parkin HeLa cells with Lipo3000. Western blotting results showed that the expression of PINK1 and pSer65-Ub and the degradation of MFN2 promoted by ISO (5 μM, 4 h) were significantly inhibited by knockdown of PINK1. Data are presented as the mean ± SD. Exact P values are reported in Appendix Table S1. One-way ANOVA followed by Dunnett's multiple comparisons test. Source data are available online for this figure.

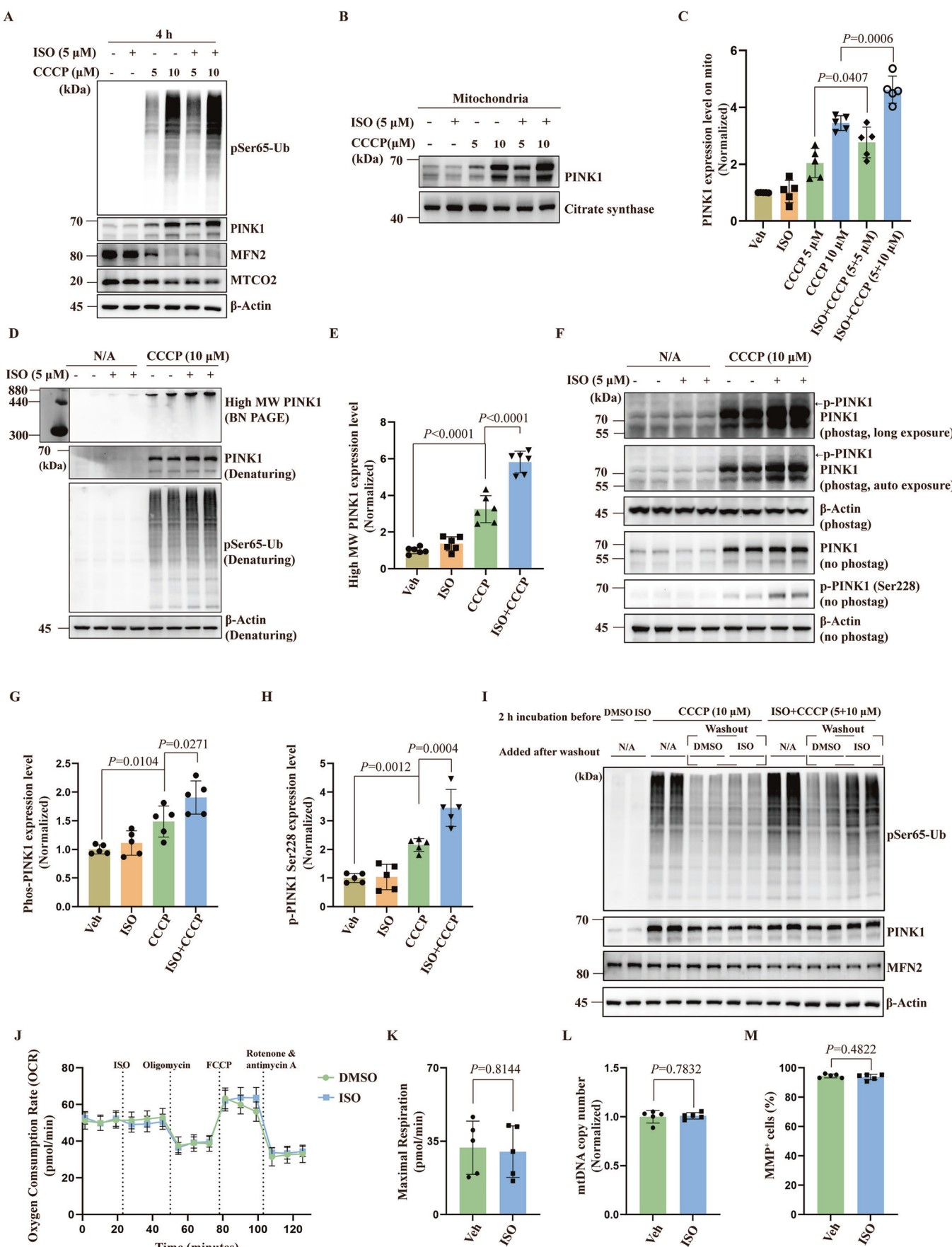

**Figure 3. ISO stabilizes the PINK1/TOM complex and sustains the active form of PINK1.**

(A) PINK1-Myc-YFP-Parkin HeLa cells were treated with ISO, CCCP, and CCCP + ISO for 4 h, respectively, and the expression of PINK1-Parkin pathway proteins and mitochondrial membrane proteins were detected by immunoblotting. (B) Isolated mitochondria from PINK1-Myc–YFP-Parkin HeLa cells were examined for the expression level of FL-PINK1 via immunoblotting. (C) Quantification of (B) showed that ISO stabilized more CCCP-induced FL-PINK1 on the OMM ($n = 5$; five biological repeats). (D) PINK1-Myc–YFP-Parkin HeLa cells were treated with ISO (5 µM), CCCP (10 µM), or CCCP + ISO (10 µM + 5 µM) for 4 h, respectively. Whole-cell lysates were analyzed by immunoblotting on BN-PAGE or denaturing gels. (E) The band intensities of the HMW complex from (D) were quantified. The expression of HMW complex was significantly increased in the co-treatment group ($n = 6$; six biological repeats). (F) Whole-cell lysates were analyzed by immunoblotting on Phos-tag SDS-PAGE or SDS-PAGE. (G) The band intensities of phospho-PINK1 in Phos-tag PAGE from (F) were quantified. The expression of phospho-PINK1 species was significantly increased in the co-treatment group ($n = 5$; five biological repeats). (H) The band intensities of p-PINK1 Ser228 in SDS-PAGE from (F) were quantified. The activated form of phosphorylated PINK1 at Ser228 was significantly increased in the co-treatment group ($n = 5$; five biological repeats). (I) PINK1-Myc-YFP-Parkin HeLa cells were treated with 10 µM CCCP or 10 µM CCCP + 5 µM ISO for 2 h. Cells were washed three times with DPBS to remove CCCP ("washout"), and then medium containing either DMSO or 5 µM ISO was added back to the cells. Cells were harvested for immunoblotting analysis before the washout and 1 h after the washout. Immunoblotting showed PINK1 stability and activity were sustained in cells co-treated with CCCP and ISO when ISO was added back after washout, compared to the DMSO treatment. (J) Seahorse was used to detect the OCR within the ISO treatment group, 5 µM ISO does not impair mitochondrial respiration ($n = 5$). (K) Based on (J), calculations of the maximum respiration values reached by the cells in the ISO and DMSO groups during the FCCP depolarization process, 5 µM ISO does not impair mitochondrial respiration ($n = 5$). (L) QRT-PCR was used to detect the mitochondrial DNA copy number within the ISO treatment group; 5 µM ISO does not decrease the mtDNA copy number ($n = 5$). (M) Flow cytometry was used to detect TMRE-labeled MMP within the ISO treatment group; 5 µM ISO does not cause the collapse of MMP ($n = 5$). Data are presented as the mean ± SD. Exact $P$ values are reported in Appendix Table S1. One-way ANOVA followed by Dunnett's multiple comparisons test in (C, E, G, H). Unpaired $t$ test in (K–M). Source data are available online for this figure.

promote PINK1–Parkin-dependent mitophagy in cells containing CCCP-damaged mitochondria.

## ISO stabilizes the PINK1/TOM complex

PRT062607 (PRT) is a recently reported potent PINK1 inhibitor (Rasool et al, 2024) that blocks the ability of ISO to enhance pSer65-Ub expression, MFN2 degradation, and Parkin translocation (Fig. EV2E,F). Here, we show that ISO stimulated PINK1-Parkin-dependent mitophagy in CCCP-treated YFP-Parkin-PINK1-Myc HeLa cells (Fig. 3A) with no increase in PINK1 mRNA (Fig. EV2G). However, mitochondria from CCCP/ISO-treated cells showed an increased level of FL-PINK1 on the OMM (Fig. 3B,C), suggesting that ISO might stabilize the PINK1/TOM complex on the OMM. This was confirmed by blue-native PAGE (an approach used to visualize the stabilization of PINK1 at the TOM complex) (Wittig et al, 2006), revealing that CCCP/ISO-treated cells stabilize more FL-PINK1/TOM high molecular weight (HMW) complex (Fig. 3D,E). Furthermore, ISO co-treatment also increased autophosphorylation of PINK1 (Fig. 3F–H), confirming that more PINK1 was stabilized and activated.

Mitochondrial stress activates PINK1 kinase, which in turn phosphorylates residue Ser65 in Ubiquitin forming pSer65-Ub and initiates mitophagy. When mitochondrial stress is resolved/removed, activated PINK1 is rapidly degraded and the abundance of pSer65-Ub decreases (Yi et al, 2024). We found that after removal (washout) of CCCP, pSer65-Ub expression in ISO-treated cells remained higher than in control cells (DMSO), supporting the concept that ISO stabilizes PINK1/TOM and prevents PINK1 inactivation/degradation (Fig. 3I). Importantly, no difference in MMP loss was found between DMSO- and ISO-treated cells, confirming that ISO is not a mitochondrial toxin (Fig. EV2H).

FL-PINK1 is quickly cleaved by PGAM5-associated rhomboid-like protease (PARL) at IMM after it is imported into mitochondria through the TOM complex at OMM. The cleaved PINK1 is extracted to the cytosol and degraded by proteasomes (Yi et al, 2024). Under physiological conditions, cells maintain a low level of PINK1. PINK1 P95A is resistant to cleavage, while PINK1 F104A is resistant to degradation (Deas et al, 2011; Yamano and Youle, 2013). We found that

ISO still increased CCCP-induced pSer65-Ub expression in both PINK1 mutants, suggesting that the effect of ISO on stabilizing PINK1/TOM is not due to directly disturbing PINK1 cleavage and degradation (Fig. EV2I,J). Combining our analysis within the ISO treatment group, through the Seahorse oxygen consumption rate detection (Fig. 3J,K), as well as the changes in mitochondrial DNA copy number (Fig. 3L) and mitochondrial membrane potential (Fig. 3M), we summarize that ISO promotes mitophagy by stabilizing the PINK1/TOM complex on the OMM without compromising the normal function of mitochondria.

## Defective PINK1–Parkin-dependent mitophagy in ALS MNs

We then investigated mitophagy in the spinal MNs of ALS patients. We analyzed published RNA sequencing (RNA-seq) data from the MN-enriched spinal cord samples of 23 ALS patients and 8 healthy controls (Krach et al, 2018; Nizzardo et al, 2020). High expression of Choline Acetyltransferase (ChAT), a biomarker for MNs, confirmed that samples were enriched with spinal cord MNs (Fig. 4A). Gene set enrichment analysis (GSEA) demonstrated upregulation of the genes/pathways involved in inflammatory response, hypoxia, apoptosis, and downregulation of oxidative phosphorylation, suggesting the presence of mitochondrial dysfunction in ALS spinal MNs (Fig. 4B). We used immunofluorescence to evaluate the co-localization of lysosome-associated membrane glycoprotein 2 (LAMP2) and MTCO2 in NeuN-positive (NeuN+) neurons in the spinal ventral horn of ALS patients and gender-matched healthy controls from the Netherland brain bank (Table EV1). The results showed 18.6% lower co-localization of LAMP2 and MTCO2 in the NeuN+ neurons of ALS patients than in healthy controls, suggesting significant impairment of mitophagy (Fig. 4C,D). Consistent with this, immunofluorescence showed that the expression of pSer65-Ub, an important substrate for the reaction of PINK1 activity, was significantly reduced in NeuN+ neurons in the ventral horn of ALS patients (Fig. 4E,F). Thus, we conclude that impaired PINK1–Parkin-dependent mitophagy is present in MNs from ALS patients.

Human iPSC-derived ALS MNs and controls were used to further examine mitophagy in ALS. ALS patient-derived iPSCs included three

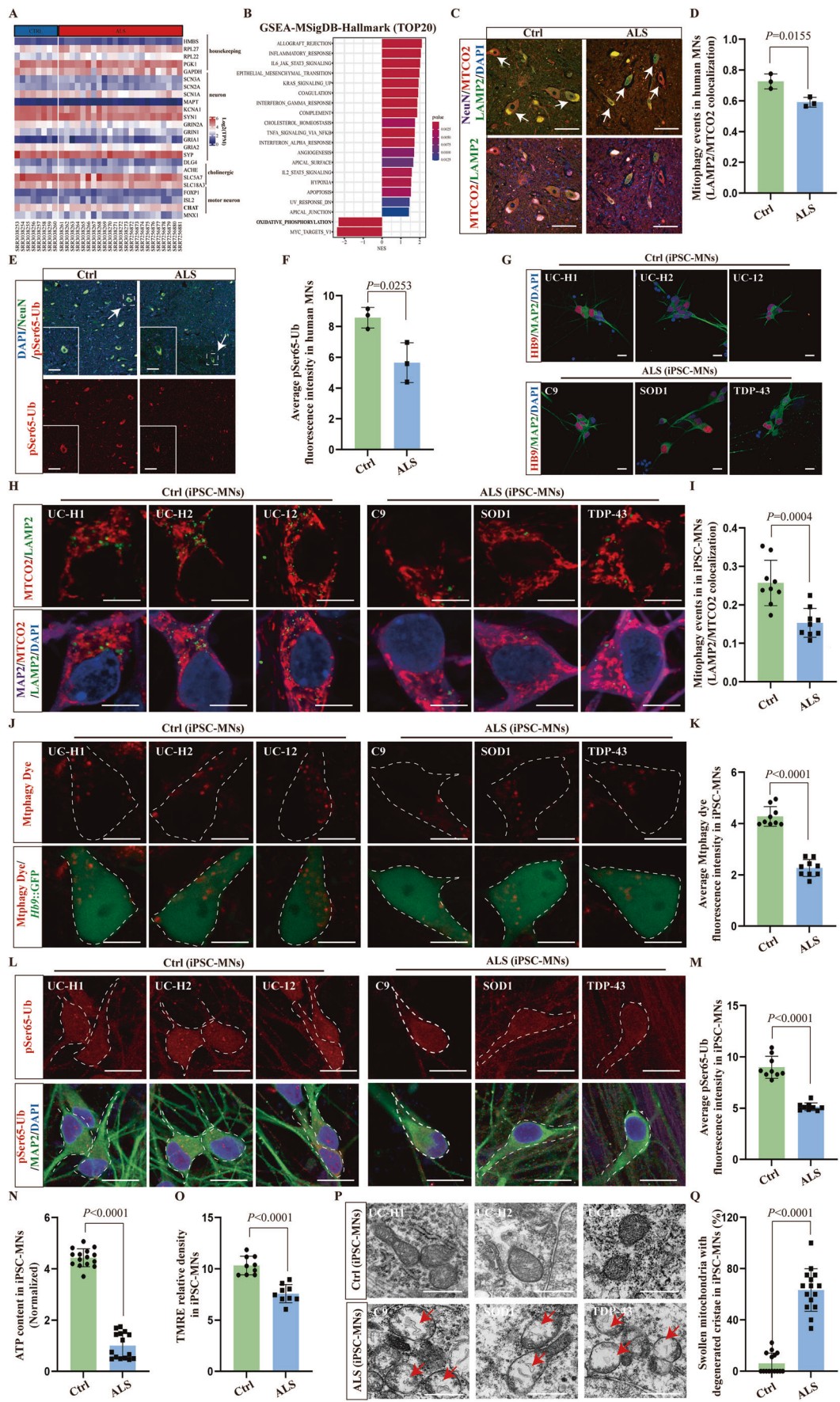

◄

**Figure 4. Accumulation of dysfunctional mitochondria and impaired PINK1–Parkin-dependent mitophagy in ALS MNs.**

(A) Heatmap of RNA-seq derived gene expression signature of laser capture microanatomy (LCM)-captured MNs from control and ALS patient samples. TPM was applied to filter expressed genes. Color bar indicates log2-transformed TPM expression levels. (B) Top 20 GSEA results. All pathways are enriched in significantly upregulated and downregulated genes. (C) Representative images of co-localization of MTCO2 and LAMP2 in NeuN⁺ MNs (NeuN staining) of the ventral horn of postmortem ALS and gender-matched healthy control spinal cords. NeuN⁺ motor neurons with reduced co-localization of MTOC2 and LAMP2 are marked with white arrows. Scale bars, 50 μm. (D) Quantification of MTCO2 and LAMP2 co-localization using Pearson's correlation coefficient demonstrating that the basal level of mitophagy was decreased in NeuN⁺ MNs of the ventral horn in postmortem ALS spinal cords. Each point represents the average value of 10 MNs for each patient sample; $n = 3$ patient samples for each group. (E) Representative images of pSer65-Ub expression in NeuN⁺ MNs of the ventral horn of postmortem ALS and gender-matched healthy control spinal cords. NeuN⁺ motor neurons with reduced pSer65-Ub expression are marked with the white arrow. Scale bars, 50 μm. (F) Quantification of (E) showing that pSer65-Ub protein expression was reduced in ALS NeuN⁺ MNs. Each point represents the average value of 10 MNs for each patient sample; $n = 3$ patient samples for each group. (G) Immunostaining of ALS patient iPSC lines-derived MNs and healthy controls on day 10 at stage 5 demonstrating the differentiation of MAP2⁺/HB9⁺ MNs. Cellular nuclei were counterstained with DAPI. Scale bars, 20 μm. (H) Representative images of co-localization of MTCO2 and LAMP2 in three ALS patient iPSC lines-derived MNs (MAP2 staining) and three healthy controls. Scale bars, 10 μm. (I) Quantification of (H) MTCO2 and LAMP2 co-localization using the Pearson's correlation coefficient demonstrating that the basal level of mitophagy was decreased in three ALS patient iPSC-derived MAP2⁺ MNs. Each point represents the average value of 10 MNs for each iPSC-derived MNs; each group contains three biological replicates of three types of MNs. (J) Representative living cell images of the mtphagy dye in three ALS patient iPSC-derived MNs (Hb9::GFP⁺) and three healthy controls. Hb9::GFP⁺ motor neurons are marked with white dotted borders. Scale bars, 10 μm. (K) Quantification of (J) the fluorescence intensity demonstrating that the basal level of mitophagy was decreased in three ALS patient iPSC-derived Hb9::GFP⁺ MNs. Each point represents the average value of 10 MNs for each iPSC-derived MNs; each group contains three biological replicates of three types of MNs. (L) Representative images of the expression of pSer65-Ub in three ALS patient iPSC lines derived MNs (MAP2⁺ staining) and three healthy controls. MAP2⁺ motor neurons are marked with white dotted borders. Scale bars, 20 μm. (M) Quantification of (L) showing pSer65-Ub protein expression was reduced in three ALS patient iPSC lines derived MAP2⁺ MNs. Each point represents the average value of 10 MNs for each iPSC-derived MNs; each group contains three biological replicates of three types of MNs. (N) Quantification of ATP stock in MNs on day 28 at stage 5 showing ATP content was reduced in ALS MNs ($n = 5$ for each MNs). (O) Quantification of MMP in Hb9::GFP⁺ MNs showing that MMP was significantly reduced in ALS Hb9::GFP⁺ MNs on day 28 at stage 5. Each point represents the average value of 10 MNs for each iPSC-derived MNs; each group contains three biological replicates of three types of MNs. (P) Ultrastructure of mitochondria in three ALS patient iPSC lines-derived MNs and three healthy controls. MNs were identified by their shape and size under EM with a low magnification (×5800,10–20 μm). The morphology of mitochondria was observed by high-power electron microscope (×37,000). Massive swollen mitochondria with significantly degenerated cristae and filamentous structures (Red arrow) were present in three ALS MNs. Scale bars, 0.5 μm. (Q) Quantification of (P) ultrastructure of mitochondria in three ALS patient iPSC lines-derived MNs and three healthy controls showed that massive swollen and degenerated mitochondria were present in three ALS MNs ($n = 5$ for each MNs). Data are presented as the mean ± SD. Exact P values are reported in Appendix Table S1. Unpaired t test. Source data are available online for this figure.

commercially available cell lines as follows: UCLi004-A containing GGGGCC repeat expansion, PFIZi013-A containing *TARDBP A382T* mutation, and WC034i containing *SOD1 D90A* mutation (WiCell Research). Three control iPSC lines, UC-H1-iPSC, UC-H2-iPSC, and UC-12-iPSC, were prepared as previously described (Chong et al, 2018; Chong et al, 2022; Tan et al, 2019). All iPSC cell lines were differentiated into MAP2⁺/HB9⁺ MNs (Figs. 4G and EV3A) and GFP-labeled using the lentivirus *Hb9*::GFP reporter system (Hung et al, 2023) (Fig. EV3B). Differentiated MNs were cultured for 28 days to allow full maturation, at which point disease-associated phenotypes, such as swollen neurites, were observed in the ALS-iPSC-derived MNs (Fig. EV3B,C, white arrow). We found that the co-localization frequency of LAMP2 and MTCO2 in the ALS MAP2⁺ motor neurons was 40.3% lower than that in the healthy control group, indicating a mitophagy disorder (Fig. 4H,I). To further explore the mitophagy levels in ALS MNs, we used a commercial mitophagy dye (mtphagy) to conduct live-cell imaging within *Hb9*::GFP-positive motor neurons. We found that the mitophagy levels indicated by the mtphagy dye in ALS MNs were also 47% lower than those in the healthy control group, further indicating the presence of damaged mitophagy in ALS MNs (Fig. 4J,K). We also noted that both the expression of pSer65-Ub and the ATP content were lower than in control MNs (Fig. 4L–N). In addition, TMRE staining (Yi et al, 2024) demonstrated that MMP was lower in ALS MNs than in controls (Figs. 4O and EV3D). Electron microscopic (EM) studies revealed swollen mitochondria with severe degeneration of cristae and aggregated filamentous structures in the soma of ALS iPSC-derived MNs (Fig. 4P,Q). These results suggest that ALS MNs are characterized by defective PINK1-Parkin-dependent mitophagy and more abundant dysfunctional/damaged and/or depolarized mitochondria than control MNs.

## ISO improves mitochondrial function and antagonizes ALS-like cellular pathology in ALS patient iPSC-derived MNs

ISO-enhanced mitophagy and protective effect experiments were performed in MNs derived from three previously used independent lines of ALS patient-derived iPSCs and appropriate controls. Initial dose–response control experiments revealed no evidence of cellular toxicity in iPSC-derived MNs in the presence of 0.5 μM ISO for 7 days (Fig. EV4A). Three ISO-treated types of ALS MNs showed higher levels of pSer65-Ub than untreated controls (Fig. 5A,B), as well as increased co-localization of LAMP2 and MTCO2 in MAP2⁺ MNs (Fig. 5C,D). Increased mtphagy dye signals indicated improved mitophagy in *Hb9*::GFP⁺ MNs (Fig. 5E,F), with abundant mitolysosome-like structures (Fig. 5G, green arrow), and fewer damaged mitochondria (Fig. 5G, red and yellow arrows), suggesting that ISO activates mitophagy in ALS MNs. In addition, ISO increased ATP content in the three types of ALS iPSC-derived MNs (Fig. 5H), with significantly reduced neurite swelling relative to controls (Fig. 5I,J). Lastly, MNs labeled using the *Hb9*::GFP reporter system (Hung et al, 2023) were detected at a much higher level after 35 days in the presence of ISO than in its absence (Fig. 5K–M), with concomitant increase in pSer65-Ub expression, mitophagy signals and decreased neurite swelling, while all these effects were abolished by co-treatment with PINK1 inhibitor PRT (Fig. EV4B–G). In summary, ALS-patient and control human iPSCs were differentiated into MNs, and studies of these MNs demonstrate that ISO stimulates PINK1–Parkin-dependent mitophagy and antagonizes ALS-like pathology in MNs.

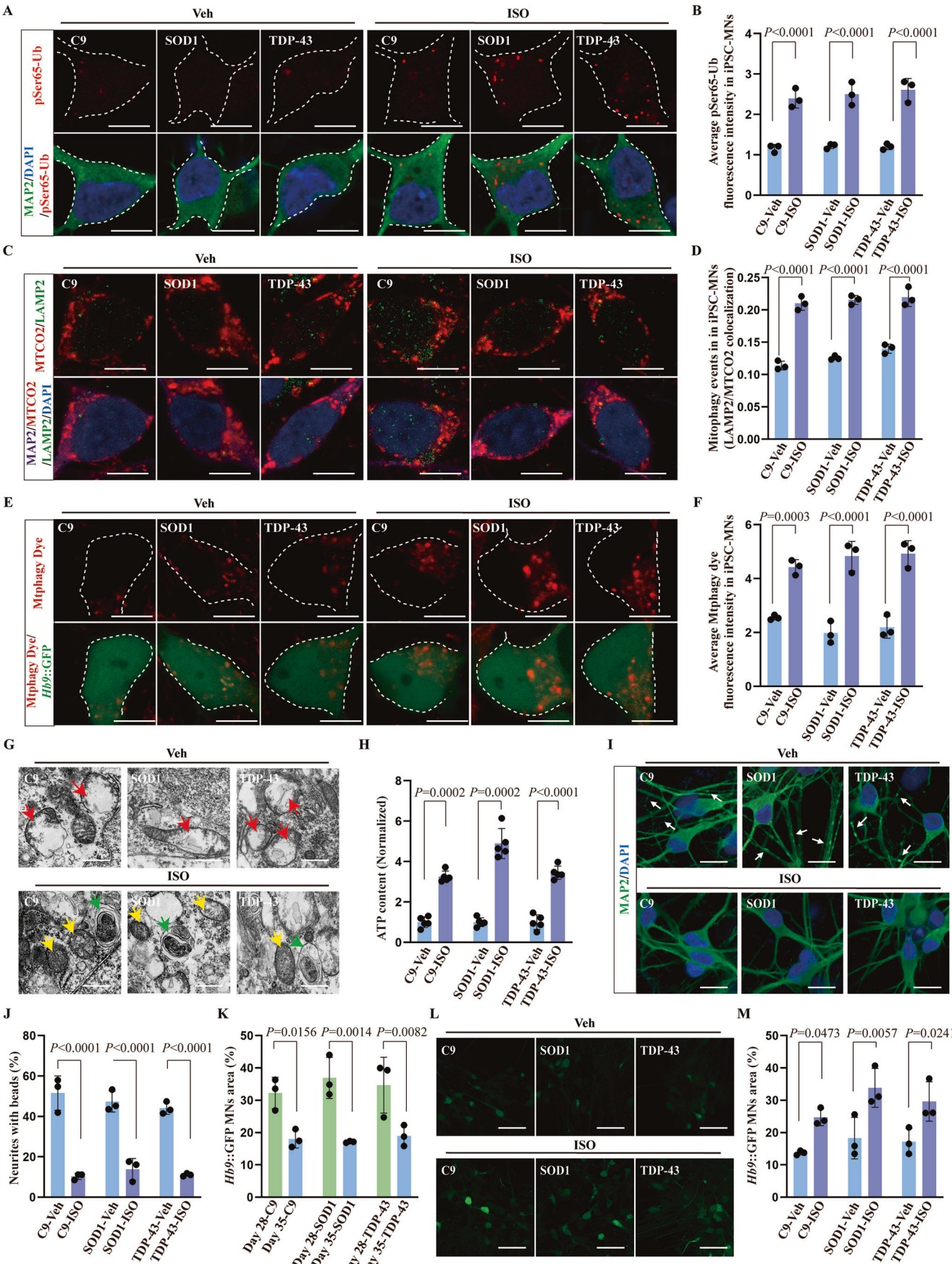

**Figure 5. ISO improves ALS-related phenotypes in three ALS iPSC lines-derived MNs by promoting PINK1-dependent mitophagy.**

(A) Representative images of pSer65-Ub protein expression in three ALS patient iPSC lines derived MAP2[+] MNs. MAP2[+] motor neurons are marked with white dotted borders. Scale bars, 10 μm. (B) Quantification of (A) showed that ISO increased pSer65-Ub protein expression in three ALS patient iPSC lines derived MAP2[+] MNs. Each point represents the average value of 10 MNs for each iPSC-derived MNs; three biological replicates for each group. (C) Representative images of MTCO2 and LAMP2 co-localization levels in three ALS patient iPSC lines derived MAP2[+] MNs. Scale bars, 10 μm. (D) Quantification of (C) MTCO2 and LAMP2 co-localization using Pearson's correlation coefficient showed that ISO induced mitophagy in three ALS patient iPSC lines derived MAP2[+] MNs. Each point represents the average value of 10 MNs for each iPSC-derived MNs; three biological replicates for each group. (E) Representative living cell images of the mtphagy dye in three ALS patient iPSC-derived MNs (Hb9::GFP[+]). Hb9::GFP[+] motor neurons are marked with white dotted borders. Scale bars, 10 μm. (F) Quantification of (E) the fluorescence intensity demonstrating that ISO enhanced mitophagy in three ALS patient iPSC-derived Hb9::GFP[+] MNs. Each point represents the average value of 10 MNs for each iPSC-derived MNs; three biological replicates for each group. (G) EM showing that ISO improved mitochondrial morphology (yellow arrow) and induced mitolysosome-like structures (green arrow) in three ALS MNs after 7 days of treatment. Scale bars, 0.5 μm. (H) Quantification of ATP stock in ALS MNs showing that ISO administration increased ATP content in ALS MNs (n = 5 for each MNs). (I) Representative images of neurites swelling with bead-like structures (white arrow) in three ALS patient iPSC-derived MAP2[+] MNs. Scale bars, 20 μm. (J) Quantification of (I) showing that neurites swelling with bead-like structures in three ALS patient iPSC-derived MAP2[+] MNs were reduced after 7 days of ISO treatment. Each point represents the average value of 10 images for each iPSC-derived MNs; three biological replicates for each group. (K) Quantification of GFP[+] area in three ALS patient iPSC lines derived GFP[+] MNs from day 28 to 35. Each point represents the average value of ten images for each iPSC-derived MNs; three biological replicates for each group. (L) Representative images of Hb9::GFP area in three ALS patient iPSC-derived Hb9::GFP[+] MNs after 7 days of ISO treatment. Scale bars, 50 μm. (M) Quantification of (L) Hb9::GFP[+] area in ALS MNs showing ISO administration increased ALS MNs survival. Each point represents the average value of ten images for each iPSC-derived MNs; three biological replicates for each group. Data are presented as the mean ± SD. Exact P values are reported in Appendix Table S1. Two-way ANOVA followed by Tukey's multiple comparisons test. Source data are available online for this figure.

## ISO ameliorates ALS-related phenotypes in ALS C. elegans model by activating PINK1–Parkin-dependent neuronal mitophagy

A dose–response study with wild-type (WT) N2 worms from hatching through adult day 1 revealed that ISO was non-toxic up to 75 μM, with no significant adverse effects on egg laying/hatching or larval development/maturation (Fig. EV5A). The ability of ISO to promote mitophagy in vivo was investigated using our previously reported transgenic worms expressing either mt-Rosella or DCT-1/LGG-1 (Cao et al, 2022; Fang et al, 2019). mt-Rosella transgenic worms were generated by expressing the mitochondrial-targeted biosensor protein Rosella, which is fused with GFP and DsRed in worm neurons. Due to the sensitivity of GFP in the acidic lysosomal environment, an increase in the DsRed/DsRed+GFP ratio indicates an increased level of neuronal mitophagy. DCT-1/LGG-1 transgenic worms were generated by expressing the DAF-16/FOXO-controlled germline-tumor affecting-1 (DCT-1) mitophagy receptor fused with GFP together with the autophagosome marker protein LGG-1 fused with DsRed in neurons, and the increased number of the co-localization of DCT-1 and LGG-1 indicates the increased level of neuronal mitophagy. Furthermore, 15 μM (but not 5 μM) ISO induced significant neuronal mitophagy in both transgenic worm strains (Fig. EV5B–D).

ISO-promoted mitophagy was also quantified in nematodes using the unc-25 promoter to drive expression of DCT-1/LGG-1 in GABAergic D-type dorsal and ventral MNs of WT (control) worms (Fig. 6A) or transgenic worms co-expressing human mutant SOD1 G93A in GABAergic D-type MNs (a worm model of human ALS) (Li et al, 2014). Consistent with findings in ALS patient samples and iPSC-MNs, basal mitophagy was lower in SOD1 G93A worms than in WT controls (Fig. 6B,C). Notably, exposure to ISO stimulated mitophagy in MNs in SOD1 G93A transgenic worms from egg stages to adult day 1 (Fig. 6B,C). This suggests that ISO can induce mitophagy in nematode MNs in vivo.

SOD1 G93A transgenic worms exhibit several pathological features of human ALS, including progressive paralysis, progressive loss of function in MNs, and progressive neurodegeneration (Fig. 6D–I). Remarkably, exposure to 15 μM ISO from egg hatching

to adult day 1 significantly improved MN function, swimming performance, and paralysis in ALS G93A nematodes (Fig. 6D,E); unc-25::GFP worms were crossed with WT or G93A worms to label the MNs and the connections between MNs. When exposure to 15 μM ISO continued from egg laying to adult day 8, ISO-treated worms began to experience MN loss/degeneration later than untreated (control) worms (Fig. 6F–H). More importantly, ISO improved the 15.4% median survival of G93A worms over the time course relative to untreated control worms (Fig. 6I). These results confirm that ISO protects against MN degeneration in an in vivo nematode model of human ALS.

Using pink-1 or pdr-1(C. elegans homolog of the mammalian Parkin) knockdown conditions in mt-Rosella worms crossed with SOD1 G93A, ISO-induced neuronal mitophagy was abolished (Fig. 6J,K), confirming that ISO induces PINK1-Parkin-dependent mitophagy. Similarly, knockdown of neuronal pink-1 and pdr-1 abrogated the beneficial effect of ISO on swimming ability in SOD1 G93A nematodes (Fig. 6L). These results provide evidence that ISO stimulates PINK1-Parkin-dependent neuronal mitophagy in SOD1 G93A nematodes.

We then wondered whether the protective effect of ISO in NMs might be mediated by a bacterial metabolite produced by E. coli OP50 in nematode media, rather than by ISO itself. To check this, SOD1 G93A nematodes were cultured in either UV-inactivated or control untreated E. coli OP50 (Fig. EV5E), and the results showed that ISO had the same protective effect in both conditions, suggesting that its activity is not dependent on bacterial metabolism. In addition, ISO did not interfere with the foraging behavior of N2 nematodes (Fig. EV5F) while it increased pharyngeal pumping in SOD1 G93A nematodes (Fig. EV5F), which could indicate that ISO protects the D-type MNs in the nematode head.

## Intranasal delivery of ISO nanoparticles to ALS SOD1 G93A transgenic mice antagonizes neurodegeneration and ALS-like phenotypes

The therapeutic potential of ISO was investigated in C57BL/6 J (control) and SOD1 G93A transgenic mice, a classical ALS mouse model. Initial studies quantified ISO by ultra-performance liquid chromatography-tandem mass spectrometry (UPLC-MS/MS) in blood and brain samples

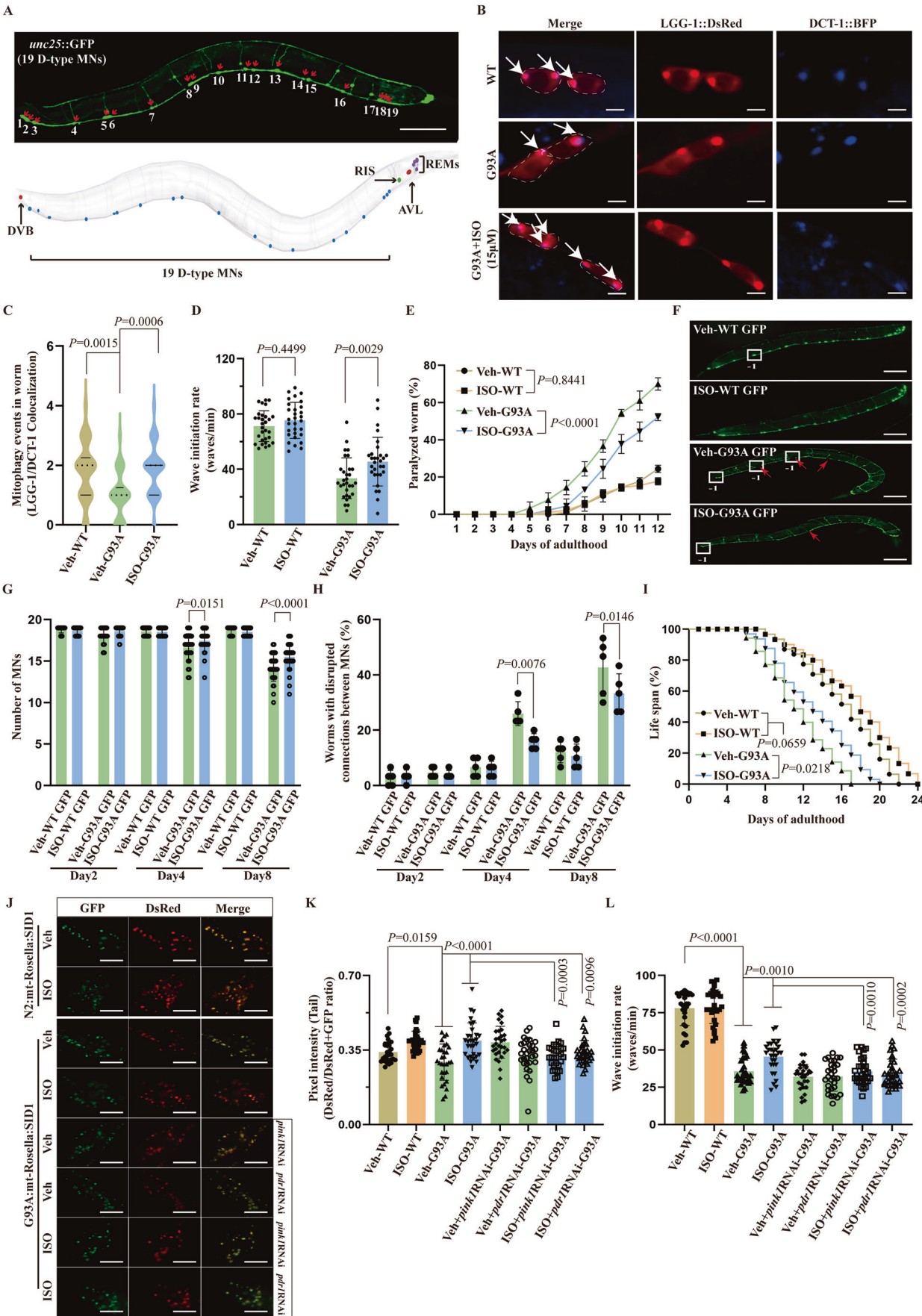

**Figure 6. ISO ameliorates ALS-related phenotypes in ALS *C. elegans* by activating *pink1-pdr1*-dependent neuronal mitophagy.**

(A) A representative image and schematic of *unc-25* promoter expression in 26 GABAergic neurons, including 19 D-type MNs in the ventral cord (blue), 4 RMEs (purple), one AVL (red), and one RIS (green) in the head, and one DVB (red) in the tail. Scale bar, 100 μm. (B) Representative images showing the co-localization of LGG-1::DsRed and DCT-1::BFP in D-type MNs of N2 or SOD1 G93A transgenic worms. *unc25*[+] D-type motor neurons are marked with white arrows. Scale bar, 1 μm. (C) Quantification of (B) showed lower basal mitophagy level in SOD1 G93A transgenic worms MNs than WT. ISO (15 μM) treatment induced mitophagy in MNs of SOD1 G93A transgenic worms. n = 30 D-type MNs from 30 worms for each group. (D–I) ISO treatment improved ALS-related phenotypes in SOD1 G93A worms, including (D) swimming ability, (E) paralysis, (F, G) MN loss (white square in (F)), scale bar, 100 μm, (H) MN degeneration defined by loss of connections between MNs (red arrows in (F)), and (I) lifespan. WT N2 worms as control in (D, E, I); WT N2 worms crossed with *unc-25*::GFP worms to visualize MNs as control in (F–H). n = 30 worms for each group. RNAi technology was used to knock down *pink-1* and *pdr-1* in worm MNs. TU3401 worms expressing SID-1 were crossed to each strain, ensuring gene knockdown within neurons. (J) Representative images of mt-Rosella indicated mitophagy in WT N2 worms and SOD1 G93A transgenic worms crossed with mt-Rosella worms and SID-1 worms. The effect of ISO on promoting neuronal mitophagy (K) and improving swimming ability (L) was inhibited by knockdown of *pink-1* and *pdr-1*. n = 30 worms for each group. Data are presented as the mean ± SD. Exact *P* values are reported in Appendix Table S1. One-way ANOVA followed by Dunnett's multiple comparisons test in (C, K, L). Log-rank (Mantel–Cox) test in (I). Two-way ANOVA followed by Tukey's multiple comparisons test in (D, E, G, H). Source data are available online for this figure.

0, 1, 2, 4, or 8 h after intraperitoneal injection of control mice with 0.3 mg/kg or 3 mg/kg ISO (Suhua, 1993). ISO was not detected in the brain (assay sensitivity 0.5 ng/mL) under these conditions (Fig. EV6A), suggesting that ISO is unable to cross the blood brain barrier (BBB) in mice. No adverse effects were observed in treated mice, but ISO was detected at expected levels in blood at all time points (Fig. EV6B).

Previous studies have shown that flavonoids can self-assemble into nanoparticles that are able to cross the BBB in rodents and other species (Yue et al, 2023). Consistent with this, self-assembled ISO nanoparticles (Nano-ISO) were successfully prepared using a published method (Fig. EV6C) (Liu et al, 2020). UPLC and EM analyses revealed that Nano-ISO contained uniform, well-defined, stable, monodispersed nanospheres (hydrodynamic diameter, 167.4 nm; polydispersity index, 0.121) carrying on average 390 μg/mL ISO at an estimated 43.6% maximal drug loading capacity (Fig. EV6D,E). Nano-ISO particles released ISO slowly over 6 days (Fig. EV6F). These results show that Nano-ISO is a self-assembling pure nanodrug with excellent drug loading capacity and stability that promotes sustained release of its bioactive payload.

Subsequently, Nano-ISO containing indocyanine green (ICG), an FDA-approved near-infrared absorbing fluorescence tracer (Chauhan et al, 2023), was administered intranasally (Fig. 7A), and treated mice were subject to fluorescence imaging for 120 h. The results indicate that the fluorescence signal of Nano-ISO#ICG in the brain peaks 24 h after dosing, decreasing steadily, but is still detectable at 120 h (Fig. 7B). After 19.5 μg Nano-ISO#ICG in 50 μL volume was delivered intranasally, no adverse effects on body weight or ability to hang onto a wire were observed over 6 days. Furthermore, ISO molecules were directly detected in the brain tissue (Fig. 7C), and Nano-ISO#ICG in the brain of anesthetized mice showed a decline between its highest point (72 h) to 96 h using in vivo optical imaging experiment (Fig. EV6G). In subsequent studies, SOD1 G93A mice were administered with Nano-ISO through intranasal dosing every 72 h.

In many cases, ALS is diagnosed after significant muscle atrophy, when MN loss has already occurred, resulting in a poor prognosis (Grad et al, 2017). Because electromyography (EMG) can detect neurogenic damage before pathological features of neurodegenerative disease emerge (Hulisz, 2018), EMG was used to monitor early stages of disease in SOD1 G93A mice (relative to control mice) (Fig. EV7A). More specifically, the compound muscle action potential (CMAP) of the gastrocnemius muscle was monitored after electro-stimulation of the sciatic nerve (Pollari et al, 2018). The results showed normal muscle function in 50-day-

old mice, while CMAP was lower and continued to decrease in animals 60 days or older (Fig. EV7B–D). However, it is encouraging that EMG detected evidence of neurological dysfunction ~4 weeks earlier than the average onset of behavioral dysfunction in SOD1 G93A mice (Wen et al, 2021).

Histological studies of the anterior horn of the spinal cord in 60-day-old SOD1 G93A mice revealed more activated microglia and astrocytes as well as smaller somatic ChAT[+] and NeuN[+] in MNs than in control mice, with no reduction in the overall number of MNs (Fig. EV7E–J). Interestingly, male and female SOD1 G93A mice showed significant differences in disease onset, disease progression, and lifespan (Alves et al, 2011). In our laboratory, female mice exhibit less individual variability than male mice. Therefore, in subsequent studies, the efficacy of Nano-ISO was examined in 60-day-old female SOD1 G93A or control mice (WT female littermate mice) after intranasal dosing with Nano-ISO, Riluzole, or vehicle, where Riluzole was used as a positive control (Fig. EV7K). Neurobiological (hind limb tremor) and behavioral (hanging time) assessments were performed to evaluate disease progression. Animal lifespan/survival was also tracked. The results showed that the onset of hind limb tremor occurred later in Nano-ISO-treated mice than in vehicle-treated mice (Fig. 7D). SOD1 G93A mice treated with Nano-ISO for 4 or 5 weeks had longer hang time and an increase in lifespan than vehicle-treated controls (Fig. 7E,F, +8.02%, difference between mean values). In contrast, disease progression and lifespan showed no statistically significant differences in mice treated with Riluzole or vehicle (Fig. 7D–F). Subsequently, SOD1 G93A mice were dosed with Nano-ISO for 60 days and MN abundance and neuroinflammation in the lumbar spinal cord were assessed by immunofluorescence, based on the observation that SOD1 G93A mice experience progressive motor function loss in hind- and forelimbs (Gento-Caro et al, 2022). This revealed 38.5% fewer ChAT[+] MNs in SOD1 G93A mice than in WT mice, and this deficit decreased to 20.6% in Nano-ISO-treated mice (Fig. 7G,H). In addition, microglia and astrocytes in Nano-ISO-treated mice had significantly lower levels of neuroinflammation than vehicle-treated mice, based on staining for inflammation markers Iba-1 and GFAP in MNs (Fig. 7I–K).

Next, co-expression of ChAT[+], MTCO2, and LAMP2 (markers of mitophagy) was quantified in MNs in the anterior horn of the lumbar spinal cord. These markers were approximately 28.2% less frequently co-expressed in SOD1 G93A mice than in WT mice, with a similar reduction of pSer65-Ub, strongly suggesting abnormally low levels of PINK1-Parkin-dependent mitophagy

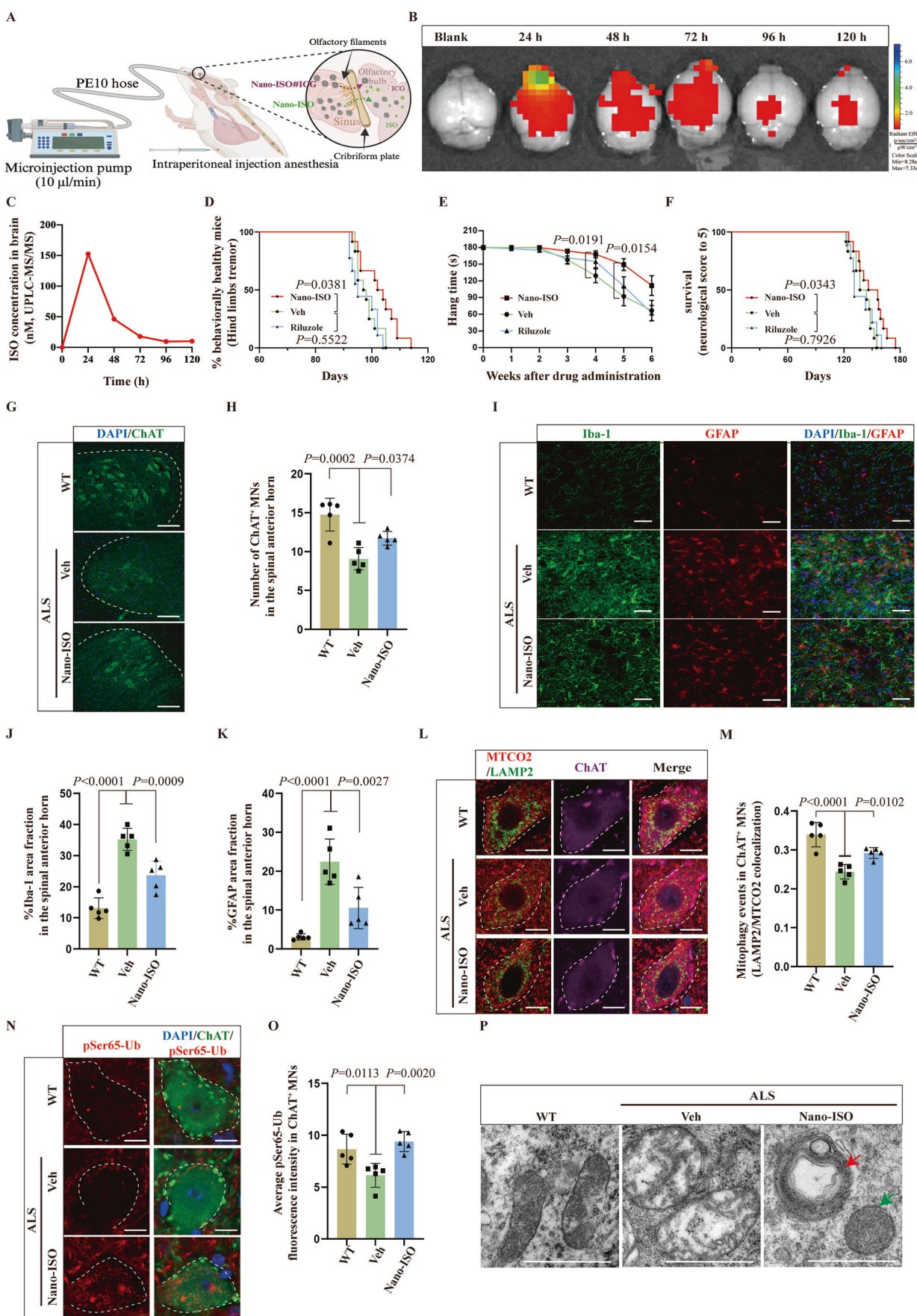

**Figure 7. Protective effects of intranasal administration of Nano-ISO in ALS SOD1 G93A transgenic mice.**

(A) Schematic of intranasal administration. Magnified schematic showing the transport of Nano-ISO and Nano-ISO#ICG from olfactory filaments across the cribriform plate into the olfactory bulb and release into the brain. (B) Representative IVIS images of the distribution of Nano-ISO#ICG in the brain after a single intranasal administration. (C) ISO concentrations in mice within 120 h detected by the UPLC-MS/MS assay. (D) The onset of tremor in SOD1 G93A mice was delayed in the Nano-ISO group, $n = 12$ mice in the Veh group, $n = 12$ mice in the Nano-ISO group, and $n = 9$ mice in the Riluzole group. (E) The hanging time of the animals in the Nano-ISO group after 4 and 5 weeks of drug administration was longer than that animals in the Veh group. (F) Nano-ISO prolonged the lifespan of SOD1 G93A mice compared to the Veh treatment. (G) Representative images of ChAT staining showing lumbar MNs of wild-type (WT) mice, Veh-treated SOD1 G93A mice, and Nano-ISO-treated SOD1 G93A mice. The ventral horn of the spinal cord is marked with white dotted lines. Scale bars, 200 μm. (H) Quantification of (G) showed that the number of ChAT$^+$ MNs was significantly increased in the Nano-ISO group. Each point represents the average value of ten images for each mouse, $n = 5$ mice/group. (I) Representative images of microglia and astrocyte activation revealed by Iba-I staining and GFAP staining in WT, Veh group, and Nano-ISO group. Scale bars, 50 μm. (J) Quantification of (I) showed microglia activation was inhibited in the Nano-ISO group. Each point represents the average value of 10 images for each mouse, $n = 5$ mice/group. (K) Quantification of (I) showed astrocyte activation was inhibited in the Nano-ISO group. Each point represents the average value of ten images for each mouse, $n = 5$ mice/group. (L) Representative images of MTCO2 and LAMP2 co-localization levels in lumbar ChAT$^+$ MNs. ChAT$^+$ motor neurons are marked with white dotted borders. Scale bars, 5 μm. (M) Quantification of (L) MTCO2 and LAMP2 co-localization using Pearson's correlation coefficient showed that ISO induced mitophagy in lumbar ChAT$^+$ MNs. Each point represents the average value of 10 MNs for each mouse, $n = 5$ mice/group. (N) Representative images of pSer65-Ub protein expression in lumber ChAT$^+$ MNs. ChAT$^+$ motor neurons are marked with white dotted borders. Scale bars, 5 μm. (O) Quantification of (N) showed that ISO increased pSer65-Ub protein expression in MNs. Each point represents the average value of ten MNs for each mouse, $n = 5$ mice/group. (P) The ultrastructure of mitochondria was observed using EM. ISO improved mitochondrial morphology (green arrow) and induced mitolysosome-like structures (red arrow) in lumbar MNs. Scale bars, 0.5 μm. Data are presented as the mean ± SD. Exact $P$ values are reported in Appendix Table S1. One-way ANOVA followed by Dunnett's multiple comparisons test in (E, H, J, K, M, O). Log-rank (Mantel–Cox) test in (D, F). Source data are available online for this figure.

(Fig. 7L–O). However, in SOD1 G93A mice treated with Nano-ISO for 60 days, co-localization of these mitophagy markers increased by 19.8% relative to vehicle-treated mice (Fig. 7L,M), with a concomitant increase in the level of pSer65-Ub (Fig. 7N,O), a reduction in mitochondrial swelling, and an increased abundance of mitolysosome-like structures adjacent to damaged mitochondria (Fig. 7P). These results suggest that ISO stimulates PINK1–Parkin-dependent mitophagy in spinal MNs thereby antagonizing ALS disease progression.

## Discussion

This study proposes and tests the hypothesis that PINK1-Parkin-dependent mitophagy is a druggable therapeutic target in ALS. To test the hypothesis, we screened an "in-house" library containing 9555 potential small-molecule mitophagy inducers in human cells expressing a mitophagy reporter. This screen identified ISO as the lead mitophagy agonist. Here, we show that ISO robustly activates PINK1, stimulates PINK1-Parkin-dependent mitophagy, and stabilizes the PINK1/TOM complex with no associated mitochondrial membrane depolarization and low cytotoxicity. Furthermore, ISO antagonizes ALS-related phenotypes in worm, mouse, and iPSC-MN models of ALS while protecting against MN loss/dysfunction across species. These results suggest that ISO could potentially lead to novel therapeutic options for ALS patients, as it likely acts through a different mechanism than drugs currently used to treat ALS.

ISO is a bioactive bioflavonoid from *Ginkgo Biloba* that reduces oxidative stress and inhibits expression of matrix metallopeptidase 9 (MMP9) (Wu et al, 2022; Yoon et al, 2006). Previous studies suggest that inhibition of MMP9 could protect against MN loss in ALS (Kaplan et al, 2014). Although ISO does not cross the BBB, self-assembled Nano-ISO does, allowing further exploration of its therapeutic efficacy via intranasal delivery to mice. It is believed that the excitatory amino acid neurotransmitter glutamate is involved in the pathogenesis of ALS, and accordingly one US FDA-approved anti-ALS drug is the anti-glutamate agent Riluzole (Bensimon et al, 1994). While oxidative stress plays a role in ALS progression, Edaravone, an

antioxidant compound, was approved as a 2nd ALS drug (Bajpai et al, 2025). However, these two anti-ALS drugs do not show strong effects for stopping or even slowing down the progression of ALS, possibly due to ALS high heterogeneity and fast disease progression (Kraft et al, 2023). Thus, it is important to identify novel and robust anti-ALS drugs. Our etiological focus on ALS is mitochondrial dysfunction in ALS motor neurons, and correspondingly, its links to excitotoxicity and oxidative stress (Arnold et al, 2024; Zuo et al, 2021). Our ISO data enable us to propose that combining antioxidants and/or anti-excitotoxic agents with strategies that enhance mitophagy to remove damaged mitochondria may offer additive/synergistic therapeutic outcomes in some ALS patients, such as those who are suffering from mutations of specific genes. In addition, the intranasal delivery employed by Nano-ISO can enhance cerebral distribution and reduce the risk of peripheral side effects, thereby providing improved benefits and adaptability in long-term treatment.

It has previously been suggested that pSer65-Ub could be a useful biomarker for mitochondrial damage in aging and disease (Hou et al, 2018). In the present study, low pSer65-Ub and low levels of mitophagy were associated with presumed functional impairment of PINK1. This is consistent with reports of reduced or absent pSer65-Ub in the brains of carriers of *PRKN* or *PINK1* mutations (Hou et al, 2018). However, the molecular mechanisms leading to impaired PINK1-Parkin-dependent mitophagy in ALS MNs are still not clear. We therefore propose two potential hypotheses describing the reduction in PINK1-mitophagy activity in ALS: a) increased cleavage of PINK1 by PARL (Liu et al, 2023) and/or protein degradation systems (such as UPS and autophagy itself), and b) compromised homeostasis of FL-PINK1 and its different forms (e.g., pathological TDP-43 may impair PINK1 degradation resulting in increased abundance of cytoplasmic cleaved-PINK and compromised mitophagy) (Sun et al, 2018). These hypotheses should be tested in the future. Mitophagy levels and the alterations in mitochondrial membrane potential varied in different pathogenic genotypes of ALS and at different stages of disease progression (Rogers et al, 2017). Due to practical challenges, we did not systematically monitor mitophagy dynamics throughout the entire progression of ALS in our models. However, it has been

reported that mitophagy may be higher in the early stages of ALS, and due to various reasons such as mitochondrial permeability changes, hyperpolarized mitochondria may also appear (Onesto et al, 2016; Tan et al, 2013; Tan et al, 2014). It has become possible to measure mitophagy in vivo through models such as mt-mKeima mice (Sun et al, 2015): crossing this mitophagy reporting mouse model with an ALS mouse strain will enable the recording of tissue- and cell-type-specific changes in mitophagy across the ALS disease continuum. The direct molecular target(s) of ISO, including potential binding sites on/in mitochondria, remain unidentified. In the context of PINK1-Parkin-mediated mitophagy, PINK1 is normally imported into mitochondria via the TOM and TIM complexes and subsequently cleaved and degraded in healthy mitochondria. However, PINK1 instead accumulates on the outer mitochondrial membrane when mitochondrial membrane potential is lost, and forms a super complex with TOM subunits such as TOM20, which facilitates PINK1 activation, thereby initiating Parkin recruitment and mitophagy (Raimi et al, 2024). There are no interactions between the ISO and TOM complex subunit that have been established, but it is noteworthy that compounds targeting the TOM complex can enhance PINK1 stabilization (Yi et al, 2024), suggesting a potential, but as yet untested, mechanistic link. Furthermore, the present study does not fully address the specific role(s) of other mitophagy pathways in ALS pathology and progression, such as impaired FUNDC1-dependent (Guo et al, 2024) and BNIP3-dependent mitophagy pathways (Rogers et al, 2017). Thus, further mechanistic and interventional studies (e.g., with ISO) on other mitophagy pathways within ALS motor neurons are needed. To note, our unpublished data also shows lysosomal dysfunction in ALS MNs, and whether and how ISO functions in lysosomal biogenesis/homeostasis in ALS MNs are under investigation.

It is well recognized that increased mitochondrial dysfunction is associated with aging. In tissue from ALS patients and in animal models of ALS, mitochondrial dysfunction is prevalent and may be exacerbated by reduced capacity for mitophagy. The present study identifies ISO as a safe and effective small-molecule compound that could have therapeutic use as a mitophagy inducer. ISO could potentially enhance selective degradation of damaged mitochondria without toxic side effects to improve therapeutic options for ALS. It is worth noting that ISO may also have great therapeutic potential in other neurodegenerative diseases, such as AD and PD, given that mitochondrial dysfunction is recognized as a common feature of many neurodegenerative diseases.

# Methods

### Reagents and tools table

| Reagent/resource | Reference or source | Identifier or catalog number |
| --- | --- | --- |
| **Experimental models** | | |
| Patient samples | Netherlands Brain Bank | |
| B6SJL-Tg (SOD1*G93A) 1Gur/J | Jackson Laboratory | No. 002726 |
| C57BL/6J | Jackson Laboratory | No. 000664 |

| Reagent/resource | Reference or source | Identifier or catalog number |
| --- | --- | --- |
| YFP-Parkin-mt-mKeima HeLa cell | This study | |
| PINK1-Myc-YFP-Parkin cell | This study | |
| YFP-Parkin HeLa cell | This study | |
| SH-SY5Y cell | Cell Bank of the Committee on Type Culture Collection, Chinese Academy of Sciences | |
| Health ctrl-iPSC#1 | This study | UC-H1 |
| Health ctrl-iPSC#2 | This study | UC-H2 |
| Health ctrl-iPSC#3 | This study | UC-12 |
| ALS-C9-iPSC | WiCell Research Institute Inc. | UCLi004-A |
| ALS-TDP-43 A382T-iPSC | WiCell Research Institute Inc. | PFIZi013-A |
| ALS-SOD1 D90A-iPSC | WiCell Research Institute Inc. | WC034i |
| N2 worm | Caenorhabditis Genetics Center | N2 |
| mt-Rosella worm | This study | |
| LGG-1/DCT-1 worm | This study | IR2160 |
| unc::25 GFP worm | This study | ngIs19 |
| unc::25 SOD1 G93A worm | This study | ngIs27 |
| unc::25 LGG-1/DCT-1 worm | SunyBiotech | sybIs5301 |
| sid-1 worm | Caenorhabditis Genetics Center | TU3401 |
| Escherichia coli (OP50) | Caenorhabditis Genetics Center | |
| **Recombinant DNA** | | |
| PINK1 P95A mutant plasmids | This study | |
| PINK F104A mutant plasmids | This study | |
| Hb9::GFP lentiviral vector | Addgene | 37080 |
| **Antibodies** | | |
| Anti-ChAT (1:50 dilution) | Sigma | AB144P |
| Anti-GFAP (1:300 dilution) | Cell Signaling Technology | 3670 |
| Anti-Iba-1 (1:200 dilution) | Wako | 019-19741 |
| Anti-NeuN (1:500 dilution) | Millipore | MAB377 |
| Anti-NeuN (1:500 dilution) | Abcam | ab134014 |
| Anti-Citrate synthase (1:1000 dilution) | Proteintech | 16131-1-AP |
| Anti-Phospho-Ubiquitin (Ser65) (1:1000 dilution) | Cell Signaling Technology | 70973 |
| Anti-LAMP2 (1:1000 dilution) | Abcam | ab25631 |
| Anti-MTCO2 (1:1000 dilution) | Proteintech | 55070-1-AP |

| Reagent/resource | Reference or source | Identifier or catalog number |
| --- | --- | --- |
| Anti-MAP2 (1:2000 dilution) | Sigma | M4403 |
| Anti-MAP2 (1:5000 dilution) | Abcam | ab5392 |
| Anti-HB9 (1:200 dilution) | Thermo Fisher Scientific | PA5-102932 |
| Anti-Phospho-Ubiquitin (Ser65) (1:500 dilution) | Cell Signaling Technology | 62802 |
| Anti-PINK1 (1:1000 dilution) | Cell Signaling Technology | 6946 |
| Anti-phospho-PINK1(Ser228) (1:1000 dilution) | Thermo Fisher Scientific | PA5-105356 |
| Anti-Parkin (1:1000 dilution) | Cell Signaling Technology | 4211 |
| Anti-Mitofusin2 (1:1000 dilution) | Proteintech | 12186-1-AP |
| Anti-Mitofusin2 (1:1000 dilution) | Cell Signaling Technology | 9482 |
| Anti-LC3B (1:1000 dilution) | Abcam | ab192890 |
| Anti-BNIP3L/Nix (1:1000 dilution) | Cell Signaling Technology | 12396 |
| Anti-FUNDC1 (1:1000 dilution) | Abcam | ab224722 |
| Anti- HIF-1α (1:1000 dilution) | Cell Signaling Technology | 36169 |
| Anti- BNIP3 (1:1000 dilution) | Cell Signaling Technology | 44060 |
| Anti-β-Actin Antibody (1:5000 dilution) | Cell Signaling Technology | 4967 |
| Anti-rabbit IgG, HRP-linked antibody (1:2000 dilution) | Cell Signaling Technology | 7074 |
| Anti-mouse IgG, HRP-linked Antibody (1:2000 dilution) | Cell Signaling Technology | 7076 |
| Alexa Fluor 488 goat anti-rabbit IgG (1:200 dilution) | Invitrogen™ | A11008 |
| Alexa Fluor 488 goat anti-mouse IgG (1:200 dilution) | Invitrogen™ | A11029 |
| Alexa Fluor 568 goat anti-mouse IgG (1:200 dilution) | Invitrogen™ | A11004 |
| Alexa Fluor 568 goat anti-rabbit IgG (1:200 dilution) | Invitrogen™ | A11011 |
| Alexa Fluor 647 goat anti-mouse IgG (1:200 dilution) | Invitrogen™ | A11004 |
| Alexa Fluor 647 goat anti-rabbit IgG (1:2000 dilution) | Invitrogen™ | A21235 |
| Alexa Fluor 647 goat anti-rabbit IgG (1:200 dilution) | Invitrogen™ | A21244 |
| Goat Anti-Chicken IgY H&L (Alexa Fluor® 488) preadsorbed (1:5000 dilution) | Abcam | ab150173 |

| Reagent/resource | Reference or source | Identifier or catalog number |
| --- | --- | --- |
| **Oligonucleotides and other sequence-based reagents** | | |
| si-pink1-1 | Guangzhou IGE Biotechnology Co., Ltd. | 5'-CGAAGCCAUC UUGAACACAA UdTdT-3' |
| si-pink1-2 | Guangzhou IGE Biotechnology Co., Ltd. | 5'-GCCGCAAAUG UGCUUCAUC UAdTdT-3' |
| si-pink1-3 | Guangzhou IGE Biotechnology Co., Ltd. | 5'-GUUCCUCGU UAUGAAGAA CUAdTdT -3' |
| si-bnip3-1 | Guangzhou IGE Biotechnology Co., Ltd. | 5'-GGAAGAU GAUAUUG AAAGAtt-3' |
| si-bnip3-2 | Guangzhou IGE Biotechnology Co., Ltd. | 5'-GAACUGCACU UCAGCAAUAtt-3' |
| si-bnip3-3 | Guangzhou IGE Biotechnology Co., Ltd. | 5'-CCAAGGAGU UCCUCU UUAAtt-3' |
| si-nix-1 | Guangzhou IGE Biotechnology Co., Ltd. | 5'-AGAAGAAGA AGUUGU AGAAtt-3' |
| si-nix-2 | Guangzhou IGE Biotechnology Co., Ltd. | 5'-GAUCAUGUU UGAUGUG GAAtt-3' |
| si-nix-3 | Guangzhou IGE Biotechnology Co., Ltd. | 5'-GUCAGAAGA AGAAGU UGUAtt-3' |
| si-fundc1-1 | Guangzhou IGE Biotechnology Co., Ltd. | 5'-GCUCGGACUU GCAUCUUAAtt-3' |
| si-fundc1-2 | Guangzhou IGE Biotechnology Co., Ltd. | 5'-GAUUUAACU GAGUAUGC AAtt-3' |
| si-fundc1-3 | Guangzhou IGE Biotechnology Co., Ltd. | 5'-CCAAGACUA UGAAAGU GAUtt-3' |
| PINK1-F | Guangzhou IGE Biotechnology Co., Ltd. | 5'-GCCTCATCG AGGAAAA ACAGG-3' |
| PINK1-R | Guangzhou IGE Biotechnology Co., Ltd. | 5'-GTCTCGTGTC CAACGGGTC-3' |
| β-actin-F | Guangzhou IGE Biotechnology Co., Ltd. | 5'-TTTGAAT GATGAGCC TTCGTGCCC-3' |
| β-actin-R | Guangzhou IGE Biotechnology Co., Ltd. | 5'-GGTCTCAA GTCAGTGTACAG GTAAGC-3' |
| **Chemicals, enzymes, and other reagents** | | |
| DMEM high-glucose medium | Gibco | 11965092 |
| DMEM without glucose medium | Gibco | 11966025 |
| Fetal bovine serum | Gibco | 10099-141 |
| Lipofectamine 3000 | Thermo Fisher Scientific | L3000075 |
| Matrigel | Corning | 354277 |
| mTeSR Plus basal medium | STEMCELL | 100-0276 |
| Dispase | Life Technologies | 17105 |

| Reagent/resource | Reference or source | Identifier or catalog number |
|---|---|---|
| DMEM/F12 | Gibco | C11330500BT |
| Neurobasal medium | Gibco | 21102-049 |
| N2 | Gibco | 17502-048 |
| B27 | Gibco | 17504-044 |
| L-Ascorbic acid | TOCRIS | 4055-50-81-7 |
| Glutamax | Gibco | 35050061 |
| CHIR99021 | Tocris | 252917-06-9 |
| DMH-1 | Tocris | 4126 |
| SB431542 | Tocris | 1614 |
| Retinoic acid | Stemgent | 04-0021 |
| Purmorphamine | Stemgent | 04-0009 |
| Valproic acid | Stemgent | 04-0007 |
| Compound E | Tocris | 6476 |
| Insulin-like growth factor | MCE | HY-P7018 |
| Brain-derived neurotrophic factor | MCE | HY-P7116A |
| Ciliary neurotrophic factor | MCE | HY-P7146 |
| Y-27632 | SELLECKCHEM | S1049 |
| Hexadimethrine bromide | MCE | HY-112735 |
| 4′,6-diamidino-2-phenylindole | MCE | HY-D0814A |
| Triton™ X-100 | Sigma-Aldrich | 9036-19-5 |
| Paraformaldehyde | MCE | HY-Y0333 |
| Imidazole/HCl | MCE | HY-D0837R |
| 6-Aminohexanoic acid | MCE | HY-B0236 |
| EDTA | MCE | HY-Y0682 |
| Glycerol | Sigma-Aldrich | 56-81-5 |
| Small molecule libraries | Topscience | L4150, L4000 and L6000 |
| Glutaraldehyde | Sigma-Aldrich | 111-30-8 |
| CCCP | MCE | HY-100941 |
| Isoginkgetin | MCE | HY-N2117 |
| pyruvate | Gibco | 11360070 |
| D-Glucose | MCE | HY-B0389 |
| L-Glutamine | Gibco | 25030081 |
| Pluronic F-127 | Sigma-Aldrich | 9003-11-6 |
| Zoletil 50 | Virbac | Zoletil™50 |
| Xylazine | Sigma | X1126 |
| PRT062607 | MCE | MCE |
| Indocyanine Green | Sigma-Aldrich | 3599-32-4 |
| Riluzole | MCE | HY-B0211 |
| D-Galactose | MCE | HY-N0210 |
| DMSO | Sigma-Aldrich | 67-68-5 |
| **Software** | | |
| FlowJo V 10.06.1 | FlowJo | |
| GraphPad Prism 9.5 | GraphPad Software | |

| Reagent/resource | Reference or source | Identifier or catalog number |
|---|---|---|
| Wave | Agilent | |
| BD FACSDiva™ | BD Biosciences | |
| Image Lab | Bio-Rad | |
| BioRender | | https://BioRender.com |
| **Other** | | |
| Human Mitochondrial DNA (mtDNA) Monitoring Primer Set | Takara | 7246 |
| DNeasy blood and tissue kit | Qiagen | 69506 |
| Hieff® qPCR SYBR Green Master Mix | Yeasen | 11202ES08 |
| Mitophagy Detection Kit | Dojindo | MD01 |
| Seahorse XF Cell mitochondrial stress test kit | Agilent | 103015-100 |
| Cell Counting Kit 8 (CCK8) | Beyotime | C0048S |
| CellTiter-Glo assay | Promega | G7570 |
| CyQUANT™ LDH Cytotoxicity Assay | Invitrogen™ | C20301 |
| Mitochondrial Membrane Potential Assay Kit with TMRE | Beyotime | C2001S |
| Cell mitochondria isolation kit | MCE | HY-K1060 |
| BCA protein assay kit | Thermo Fisher Scientific | 07920 |
| Phos-tag SDS-PAGE | FUJIFILM Wako | 195-17991 |
| Non-denaturing gel electrophoresis kit | Real-Times (Beijing) Biotechnology Co., Ltd | RTD6139-0312 |
| Enhanced chemiluminescence (ECL) kit | Millipore | WBKLS0500 |

## Postmortem ALS patient samples

Postmortem spinal cord and brain samples of ALS patients and sex-matched healthy controls were obtained from the Netherlands Brain Bank (https://www.e-nbb.org). Sample details are available in Table EV1. The related experiments were performed at Akershus University Hospital (Norway) with approval from the Regional Committee for Medicine and Health Research Ethics (REK# 412997). Informed consent was obtained from all human subjects (next of kin), and the experiments conformed to the principles set out in the WMA Declaration of Helsinki and the Department of Health and Human Services Belmont Report.

## Animals and ethics statement

SOD1 G93A transgenic mice that express a G93A mutant form of human SOD1 (B6SJL-Tg (SOD1*G93A)1Gur/J, JAX stock No. 002726) and C57BL/6J mice (JAX stock No. 000664) were

purchased from the Jackson Laboratory. All mouse strains used in this study were housed in the Laboratory Animal Centre of Sun Yat-Sen University, where they were maintained in a constant temperature and humidity environment with a 12h-light/12h-dark cycle. Hemizygous female mice were used in these experiments. Wild-type female littermate mice were used as controls. Nano-ISO was administered with Tribromoethyl alcohol (MCE, HY-B1372) as an anesthetic. For tissue sampling and brain imaging, mice were treated with Zoletil 50 (Virbac) and Xylazine (Sigma, X1126) and then euthanized using carbon dioxide. Animal experiments were carried out using protocols approved by the Institutional Animal Care and Use Committee (IACUC), Sun Yat-Sen University (SYSU-IACUC-2022-001202).

## Differential analysis

A total of 31 LCM-seq samples (including 23 spinal motor neuron samples from ALS patients and 8 from healthy controls) were used in this study; the relevant data were downloaded from the GEO database under accession numbers GSE76220 and GSE115130 (Krach et al, 2018; Nizzardo et al, 2020). Quality control was performed using SAMtools, and the results were collated using MultiQC. Adapter sequences and low-quality reads were removed using Trimmomatic. The remaining reads were aligned to hg38 build reference using STAR v2.5.2b with default parameters, followed by quantification through RSEM. Initially, Principal Component Analysis (PCA) was employed to scrutinize the impact of batch effects on gene expression data derived from two distinct studies (Subramanian et al, 2005). Once batch effects were mitigated, differential gene expression analysis was conducted using DESeq2. Differentially expressed genes (DEGs) met the criteria of an unadjusted $P$ value $< 0.05$ and fold-change (FC) $> |1.2|$. GSEA was performed using the Bioconductor clusterProfiler R package (version 4.2.0). Significantly altered LFC values were analyzed by GSEA using the Molecular Signatures Database (hallmark gene sets). Heatmaps and bar plots were generated by R4.2.0 software.

## Cell lines and C. elegans strains

YFP-Parkin-mt-mKeima, PINK1-Myc-YFP-Parkin, YFP-Parkin HeLa and SH-SY5Y cells were cultured in DMEM high-glucose medium (Gibco,11965092) containing 10% fetal bovine serum (FBS, Gibco, 10099-141) and grown at 37 °C in 5% $CO_2$. In hypoxic cultures, cells were cultured with normal medium under a hypoxic condition (1% $O_2$, 94% $N_2$, and 5% $CO_2$) using a multi-gas incubator (APM-30D; Astec Corporation). Cells were transfected with PINK1 P95A and PINK F104A expression plasmids using Lipofectamine 3000 (Thermo Fisher Scientific, L3000075). For gene silencing, cells were transfected overnight with 50 pmol targeted siRNAs using Lipofectamine 3000. Three iPSC lines, UC-H1-iPSC, UC-H2-iPSC, and UC-12-iPSC, were maintained in our laboratory. Three familial ALS iPSC cell lines, purchased from WiCell Research Institute Inc., harbored the following mutations: C9 (UCLi004-A, GGGGCC repeat expansion), TDP-43 (PFIZi013-A, TDP-43 A382T), and SOD-1 (WC034i, SOD1-D90A). iPSC cells were cultured in plastic microtiter plates or confocal dishes pre-coated with Matrigel (Corning, 354277) in mTeSR Plus basal medium (STEMCELL, 100-0276) in an incubator under 5% $CO_2$ at 37 °C. The culture medium was replaced daily.

*C. elegans* strains used in this study included mt-Rosella: N2; Ex[*Punc-119*TOMM-20::Rosella; *rol-6(su1006)*], IR2160: N2:Ex(002[*Prab-3*::dsRed::LGG-1*Prab-3*DCT-1::gfp;P*myo-2*::gfp], ngIs19 (*unc-25p*::GFP), ngIs27 (*unc-25p*::G93A SOD1) and TU3401: sid-1(pk3321) V;uIs69[pCFj90(*myo-2p*::mCherry)+*unc119p*::sid-1]V. *C. elegans* cultures were maintained at 20 °C on NGM plates with *Escherichia coli* OP50 using standard methods. Bristol strain N2 worms were used as a control. Crosses were verified by observing the co-transfer of markers or by PCR. To monitor the level of mitophagy in MNs, a PHX5301 sybIs5301[*Punc-25*::DCT-1::GGGGS::ceBFP::unc-54 3'UTR, P*unc-25*::LGG1::GGGGS::Dsred2::unc-54 3'UTR, Lin44] stable integrated transgenic strain was generated by SunyBiotech with X-ray irradiation. The ceBFP plasmid was kindly provided by Dr. Ou Guangshuo.

## iPSC-derived motor neurons

Motor neurons were generated from iPSCs as described previously (Du et al, 2015). iPSCs were dissociated with dispase (1 mg/mL, Life Technologies, 17105) and split 1:6 on Matrigel-coated plates. On the following day, the medium was replaced with a chemically defined neural medium, containing Dulbecco's modified Eagle's medium (DMEM)/F12 (Gibco, C11330500BT), Neurobasal medium (Gibco, 21102-049) at a 1:1 ratio, $0.5 \times$ N2 (Gibco, 17502-048), $0.5 \times$ B27 (Gibco, 17504-044), 0.1 mM L-Ascorbic acid (TOCRIS, 4055-50-81-7) and $1\times$ Glutamax (Gibco, 35050061). 3 μM CHIR99021 (Tocris, 252917-06-9), 2 μM DMH-1 (Tocris, 4126), and 2 μM SB431542 (Tocris, 1614) were added to the medium. The culture medium was changed every day. Human iPSCs were maintained under these conditions for 6 days and induced into neuroepithelial (NEP) cells. The NEP cells were then dissociated with dispase and split at a 1:6 ratio with the same medium as described above, and 0.1 μM Retinoic acid (RA) (Stemgent, 04-0021), 0.5 μM Purmorphamine (Stemgent, 04-0009), 1 μM CHIR99021, 2 μM DMH-1, and 2 μM SB431542 were added. The culture medium was changed every day. The NEP cells were maintained under these conditions for 6 days and differentiated into OLIG2+ motor neuron precursors (MNPs). The OLIG2+ MNPs were expanded in the same medium containing 3 μM CHIR99021, 2 μM DMH-1, 2 μM SB431542, 0.1 μM RA, 0.5 μM purmorphamine, and 0.5 mM valproic acid (VPA) (Stemgent, 04-0007) and split at a 1:6 ratio once weekly with dispase. OLIG2+ MNPs were frozen in 70% DMEM/F12, 20% fetal bovine serum, and 10% DMSO in liquid nitrogen until use. After thawing, cells were grown in expansion medium.

To induce motor neuron differentiation, OLIG2+ MNPs were dissociated with dispase and cultured in suspension in neural medium containing 0.5 μM RA and 0.1 μM purmorphamine. The culture medium was changed daily. After 6 days, OLIG2+ MNPs were differentiated into HB9+ motor neurons. HB9+ motor neurons were cultured with 0.5 μM RA, 0.1 μM purmorphamine, 0.1 μM Compound E (Tocris, 6476), and three neurotrophic factors, including insulin-like growth factor (IGF) (10 ng/mL, MCE, HY-P7018), brain-derived neurotrophic factor (BDNF) (10 ng/mL, MCE, HY-P7116A), and ciliary neurotrophic factor (CNTF) (10 ng/mL, MCE, HY-P7146) for 10 days. During differentiation, Y-27632 (SELLECKCHEM, S1049) was added for 12 h during each passage. Neurotrophic factors were removed from the culture medium containing mature MNs on day 20 to promote the appearance of ALS-related phenotypes.

## Mitochondrial function and neurite swelling in iPSC-derived motor neurons

The *Hb9*::GFP lentiviral vector was purchased from Addgene (ID: 37080). Lentivirus production was carried out by Guangzhou IGE Biotechnology Co., Ltd. For lentivirus transfection, 100,000 OLIG2$^+$ MNPs were seeded in the presence of 10 µg/mL polybrene per confocal dish and incubated with 1 mol virus for 8 h. Cells were washed twice with culture medium, incubated for 48 h, and then allowed to differentiate into HB9$^+$ MNs. Green fluorescence protein began to appear after 10–12 days. On day 28, cells were stained with 1× TMRE (Beyotime, C2001S) for 15 min, washed twice with medium, and analyzed by visual inspection for GFP$^+$ expression via confocal microscopy. ATP content was estimated in $2 \times 10^4$ MNPs in 96-well plates at day 28 of maturation using CellTiter-Glo assay (Promega G7570). Alternatively, ATP content was evaluated on day 7 of ISO treatment. The CellTiter-Glo assay was performed according to the manufacturer's instructions. Swelling of neurites was analyzed using MAP2 immunostaining after 28 days of differentiation with or without 7 days of exposure to ISO. Swollen neurites were evaluated using confocal microscopy under a ×40 objective.

## Immunofluorescence

Spinal cord MNs were examined in 20-µm tissue after embedding with OCT. Sections were blocked with 10% donkey serum in 0.3% Triton 100X for 60 min and then incubated with primary antibody in blocking buffer at 4 °C overnight. iPSC-derived motor neurons were grown in 35-mm confocal glass-bottom dishes (NEST Biotechnology), fixed at room temperature for 10 min in 4% paraformaldehyde (PFA), and then incubated in blocking buffer (1% BSA, 0.1% Triton X-100 in PBS) for 1 h. After blocking, cells or sections were incubated overnight at 4 °C with the primary antibody, and washed three times in PBS at room temperature for 10 min. After overnight incubation at 4 °C or incubation at room temperature for 60 min with the fluorochrome-conjugated secondary antibody (Thermo Fisher Scientific), cells or sections were washed three times in PBS for 10 min at room temperature. Cell nuclei were stained with 4',6-diamidino-2-phenylindole (DAPI; 1:2000) and visualized using confocal laser scanning microscopy and ImageJ software (National Institute of Health, Bethesda, MD, USA).

## Immunoblotting

For the SDS-PAGE and Phos-tag SDS-PAGE, the cell culture medium was removed and rinsed with DPBS (1×) twice. A cell mitochondria isolation kit (MCE, HY-K1060) was used to isolate mitochondria within cells. Then Laemmli SDS buffer (62.5 mM Tris-HCL, pH 6.8, 25% glycerol, 2% SDS) containing 1x protease and phosphatase inhibitor cocktail was added to the cells. The phos-tag SDS-PAGE (FUJIFILM Wako, 195-17991) experiment was performed as previously reported (Sugiyama and Uezato, 2022). The native PAGE procedure was similar to that reported previously (Wittig et al, 2006). Briefly, solubilization buffer A (50 mM Sodium chloride, 50 mM Imidazole/HCl, 2 mM 6-Aminohexanoic acid, 1 mM EDTA, pH 7.0 at 4 °C) was added to cells, and samples were adjusted to 50% glycerol and 5%

Coomassie blue G-250. For mammalian mitochondria, Triton X-100 was used to control the ratio at 20%. Non-denaturing gel electrophoresis kit and marker were from Real-Times (Beijing) Biotechnology Co., Ltd (Real-Times, RTD6139-0312 and RTD6137). Total protein was quantified using a BCA protein assay kit (Thermo Fisher Scientific, 07920). After electrophoresis, proteins were transferred to 0.2 mm polyvinylidene fluoride (PVDF) membrane and blocked with 5% non-fat milk or 5% BSA in 1× tris-buffered saline-Tween 20 (TBST) for 2 h. The membranes were incubated overnight at 4 °C with the primary antibody. Secondary antibodies were added to the incubation/blotting buffer and incubated with membranes for 1 h at room temperature. The blots were visualized using an enhanced chemiluminescence (ECL) kit (Millipore, WBKLS0500) and quantified using Bio-Rad Image Lab 5.2.1 software.

## High-content drug screening

YFP-Parkin-mt-mKeima HeLa cells were plated in 96-well plates at approximately 8,000 cells per well. After incubating at 37 °C overnight, the medium was replaced with medium containing the test compound at 10 µM. Test compounds were selected from three small-molecule libraries (L4150, L4000, and L6000 purchased from Topscience) containing 9555 compounds. Cells were incubated for 24 h, and then four randomly selected fields per sample were imaged to measure mKeima expression (Xie et al, 2022) using a PE high content imaging system and FPbase. The mitophagy indices were quantified using ImageJ software (i.e., red pixels divided by total green plus red pixels. Data were normalized to control sample treated with 5 µM CCCP for 4 h). Candidate mitophagy inducers had a mitophagy index >1.

## Transmission electron microscope for mitochondria

Sample preparation for electron microscopy was as described previously (Ruffoli et al, 2015). Briefly, mice were anesthetized and perfused through the left ventricle with 0.9% normal saline and 5% glutaraldehyde/4% paraformaldehyde in PBS. The lumbar spinal cord was removed quickly and post fixed in 5% glutaraldehyde at 4 °C overnight. MNs were incubated with 5% glutaraldehyde for 30 min at room temperature, followed by 5% glutaraldehyde overnight. Samples were washed with PBS and fixed in 1% OsO$_4$ at 4 °C for 2 h, dehydrated in an alcohol gradient series from 30 to 100%, embedded in resin, and cut into ultrathin transverse sections. Sections were stained with lead citrate and uranyl acetate, visualized, and scanned by transmission electron microscopy. Mitochondrial parameters were quantified as described previously (Fang et al, 2019). The percentage of damaged mitochondria and the number of mitophagy-like events were estimated.

## Real-time quantitative PCR (qPCR) analysis of mitochondrial DNA content and PINK1 mRNA level

For the measurement of mtDNA content, we used a human mitochondrial DNA (mtDNA) monitoring primer set (Takara, 7246). Briefly, the cellular DNA was first extracted from the treated HeLa cells using the DNeasy blood and tissue kit (Qiagen, 69506). Then, according to the instructions, the mitochondrial DNA ND1 gene and the ND5 gene, or the nuclear DNA SLCO2B1 gene and

the SERPINA1 gene were utilized as mtDNA or nDNA markers, respectively, and their levels were quantified by a CFX96 real-time PCR detection system (Bio-Rad) with Hieff® qPCR SYBR Green Master Mix (Yeasen, 11202ES08). For the measurement of PINK1 mRNA level, Hieff® qPCR SYBR Green Master Mix was utilized for qPCR.

## Seahorse oxygen consumption rate (OCR) detection

OCR measurements were achieved using an Agilent Seahorse XF Cell mitochondrial stress test kit. Cells were plated with 50,000 cells per XF24 well to ensure ~90% surface coverage at the time of the experiment the next day. For OCR analysis, medium was exchanged for mitochondrial stress medium (Seahorse XF DMEM Medium (Agilent, 103575-100) supplemented with 1 mM pyruvate, 2 mM glutamine and 10 mM glucose) at 1 h before the assay. The cultures were similarly equilibrated in a non-$CO_2$ incubator. Substrates and selective inhibitors (Agilent, 103015-100) were injected to achieve final concentrations of oligomycin at 1.5 μM, carbonyl cyanide 4-(trifluoromethoxy) phenylhydrazone (FCCP) at 1.0 μM, and rotenone/antimycin A at 0.5 μM, according to the manufacturer's instructions. The samples in each well were quantified using BCA assay, and then the data were normalized using the Wave software.

## CCK8 assays

Cell viability was determined using Cell Counting Kit 8 (CCK8). Briefly, approximately 10,000 cells were seeded per well in a 96-well microtiter plate, pre-incubated for 24 h, and then incubated with or without drug for 24 h. Viable cells were estimated by adding 10 μL CCK8 solution per well, incubating for 1 h at 37 °C, and measuring absorbance at 450 nm.

## Flow cytometry

Mitophagy index was assessed in mKeima-expressing cells as described previously (Yi et al, 2024). Briefly, mt-mKeima HeLa cells were harvested with trypsin and analyzed by flow cytometry (BD LSRFortessa X-20) equipped with a 405-nm and 561-nm laser. Lysosomal mito-Keima indicated that mitophagy index was detected using dual-excitation ratiometric pH measurements at 405 nm (pH 7) and 561 nm (pH 4) lasers with 695 nm and 670 nm emission filters, respectively. A minimum of 10,000 events was counted and analyzed using Diva software (BD Biosciences). Alternatively, to assess mitochondrial membrane potential, cells were stained in 1× TMRE for 15 min, washed twice, harvested with trypsin, and analyzed using flow cytometry. A minimum of 10,000 events was counted. Data were analyzed using FlowJo v10.0.

## Galactose/glucose assay

The galactose/glucose assay was performed as previously reported (Hertz et al, 2024). YFP-Parkin HeLa cells were isolated by trypsinization, washed three times in DMEM with glucose (MCE, HY-N0210) and 10% FBS or DMEM without glucose (Gibco 11966025) containing 10% FBS, 1 mM pyruvate (Gibco, 11360070), seeded in 96-well plates (Shanghai Jingan Biotechnology Co., LTD, J09604) at 8000 cells/well, and incubated for 24 h. Cells were then incubated in medium containing

a test compound (10 μM or 5 μM) or CCCP (positive control) for 20 h, rinsed with 50 μL glucose-containing medium per well, and assessed for ATP production/cell viability using the CellTiter-Glo assay (Promega G7570) according to the manufacturer's instructions. Cell survival was normalized to the survival of cells treated with vehicle (DMSO), and the ratio of cell survival in the presence of galactose vs glucose was calculated. Drugs were considered toxic when they caused a 10–15% reduction in the production of ATP in galactose vs glucose-containing medium.

## Live-cell imaging

YFP-Parkin HeLa cells were plated in 96-well plates at ~8000 cells per well and analyzed using high content imaging at 37 °C with 5% $CO_2$. Images were processed and analyzed using ImageJ software.

## Drug administration

For cells, ISO or an equal volume of DMSO or PRT was added to the culture medium, and then the cells were harvested. For C. elegans, worms were cultivated on NGM plates containing ISO or solvent control (DMSO). For mice, 3 mg/kg ISO was administered by intraperitoneal injection, or nano-ISO was administered by intranasal delivery (Liu et al, 2020). In the latter case, mice were anesthetized with 1.25% tribromoethyl alcohol to sustain anesthesia, a micro syringe was inserted about 8 mm into the nostril, and a sustained release pump was used to deliver 50 μl at 5 μl/min once every three days over 60 days. After each treatment, mice were placed on a thermal pad for 10–20 min. Normal saline was used for the vehicle group, and Riluzole (5 mg/kg in drinking water) was used as a positive control. For iPSC-derived MNs, because the culture medium at stage 5 was half-changed every other day, a final concentration of 1 μM ISO or 4 μM PRT or equal volume of DMSO was added to the remaining half of the culture medium to achieve the final working concentration.

## C. elegans viability

Fecundity (3 h egg-laying), egg hatching, and larval development were assessed in N2 C. elegans as described previously (Xie et al, 2022). Briefly, synchronized eggs were placed on NGM plates seeded with Escherichia coli (OP50, Caenorhabditis Genetics Center). L4 larvae were transferred onto fresh OP50-seeded NGM plates and allowed to grow to adulthood on day 1. Ten adult day 1 worms were transferred onto NGM plates with OP50 without or with ISO (5, 15, 75, 150 μM) or vehicle control (DMSO). Five plates of worms were prepared and analyzed for each group. The gravid worms were incubated for 3 h, removed from the plates, and the number of eggs per plate was counted. The following day, the number of unhatched eggs and L1 larvae was counted. Larval development to stage L4 was assessed after 36 h. L4 larvae were incubated for 16 h, and the number of viable adult worms was assessed.

## Evaluation of ALS-related phenotypes in C. elegans

Worms were treated with ISO or vehicle from egg hatching. For swimming experiments, the method was the same as previously reported (Vaccaro et al, 2012). Briefly, 1-day-old adult worms were transferred to 20 μl M9 buffer, preincubated for one min, after which swimming motion was recorded for 30 s using a

stereomicroscope and the number of body swings counted. If a worm moved its nose but not its body in response to physical contact, the animal was considered paralyzed (Li et al, 2014). To measure viability, transgenic worms were cultured in the NGM plates with 15 μM ISO or vehicle (DMSO), and viable animals were counted daily starting on adult day 1. Worms were considered dead if they did not respond to external stimuli. Pharyngeal pumping rate was evaluated manually every 2 days from day 2 to day 14. To quantify MN loss, animals were immobilized in M9 with 5 mM sodium azide and mounted on slides with 2% agarose pads. MNs were visualized and assessed for degeneration with confocal microscopy as described previously (Vaccaro et al, 2012).

## RNA interference by feeding

*C. elegans* were treated with RNAi as previously reported (Fang et al, 2019). Briefly, worms were co-cultured with bacteria engineered to facilitate knockdown of *pink-1* or *pdr-1* in *C. elegans*. Engineered bacteria were grown in LB media in the presence of 50 mg/mL ampicillin and added to NGM plates containing 1% ampicillin and 1% IPTG with/without ISO. Animals were synchronized by bleaching or egg laying and grown on RNAi plates until adult day 1. Assessments of swimming capacity and neuronal mitophagy were performed on adult day 1.

## Preparation and functionalization of self-assembled ISO nanoparticles

1 mg of ISO and 3 mg of Pluronic F-127 were separately dissolved in 100 μl of DMSO, thoroughly mixed, added to 2 mL ultra-pure water, exposed to ultrasound at room temperature for 30 min, and filtered using a 100 kDa ultrafiltration tube. Nanoparticles were characterized using a particle size analyzer (Malvern), transmission electron microscopy (TEM, JOEL), high-performance liquid chromatography (HPLC), and, in the brains of anesthetized mice, by fluorescence using an animal in vivo optical imaging system (IVIS, Tanon ABL X6).

## Quantification of ISO using UPLC-MS/MS

Quantitative methods were previously reported (Zhao et al, 2019). In brief, plasma and brain samples were collected from mice. Plasma was centrifuged at 3000 rpm for 10 min to obtain serum samples. Brain tissue was homogenized with triploid saline, centrifuged at 12,500 rpm for 10 min, and the supernatant was extracted with three volumes of cold acetonitrile. The supernatant was analyzed using an ACQUITY I-Class UPLC system connected to a Xevo TQD Mass Spectrometry (Waters Corp., USA). Samples were analyzed on a Hypersil GOLD column (50 × 2.1 mm, 1.9 μm, Thermo Fisher) in water:acetonitrile (35:65, v/v). Multiple reaction monitoring (MRM) transition for iso was $m/z$ $567.345 > 135.062$ generated by IntelliStart optimization program. The parameters of mass spectrometry were: desolvation temperature of 350 °C, capillary voltage of 3.0 kV, tune cone voltage of 80 V, tune collision energy of 48 V. Data were recorded and analyzed using Masslynx 4.2 software (Waters).

## Animal grouping and pathological evaluation

Five litters of SOD1-positive female mice were randomly assigned to different experimental groups ($n = 12$ in the Nano-ISO group,

$n = 12$ in the vehicle group, and $n = 9$ in the Riluzole group). SOD1-negative mice were used as the wild-type for unified sampling and pathological evaluation after 60 days of drug treatment. ChAT, GFAP, Iba-1, and other mitophagy markers (MTCO1/LAMP2/ChAT/DAPI and pSer65-Ub/ChAT/DAPI) were assessed using standard immunofluorescence methods and ImageJ software (NIH, USA).

## Evaluation of ALS-related phenotypes in mice

Assessments of neurological functions used a six-point scoring system (0–5), as described previously (Obrador et al, 2021). Mice were tested daily and animals with score of 5 (highest score) were euthanized for ethical reasons. The motor function of mice was assessed using the hanging wire test, which measures latency to fall from a suspended wire. The maximum latency was 180 s. Latency was the average of three attempts per mouse, measured and recorded weekly. Electrophysiological assessment (described previously (Pollari et al, 2018)) calculated the compound motor action potential (CMAP) of the gastrocnemius muscle after stimulation of the sciatic nerve. Mice were anesthetized with 1.25% tribromoethyl alcohol to sustain anesthesia. Stimulating electrodes were placed subcutaneously on each side of the sciatic notch, separated by ~2 cm. The recording electrode was placed subcutaneously, aligned with the gastrocnemius muscle. The reference electrode was embedded subcutaneously 2 to 5 mm under the skin next to the Achilles tendon at a 30-degree angle. The ground electrode was placed subcutaneously in a similar manner as the stimulating electrodes. The stimulus intensity started at 1.0 V and gradually increased to a maximum value. The CMAP amplitude at this time was recorded. The EMG equipment was from iWorx (iWorx Systems Inc.).

Animal/samples (mice) were assigned randomly to the Veh or the Nano-ISO groups, and mice were randomly selected for the behavioral experiments. In data collection and analysis (for example, mouse behavioral studies, mouse neurological functions assessment, mouse imaging data analysis, as well as imaging and data analysis of electron microscopy), the researcher(s) was (were) blinded as to the experimental design.

## Lactate dehydrogenase (LDH) cytotoxicity assay

MNPs were seeded at 10,000 cells per well in a 96-well plate at stage 5. Culture medium was replaced with 200 μl of fresh media on day 28 after 7 days of ISO treatment. Culture medium was then collected for quantification of LDH leakage using CyQUANTTM LDH Cytotoxicity Assay (Invitrogen) following the manufacturer's instructions. Levels of leaked LDH were recorded, normalized, and quantified based on the manufacturer's instructions.

## Induced motor neuron survival assay

The experimental procedures used for lentivirus transfection of MNs were described previously. In brief, 50,0000 MNPs were seeded into six-well plates and transfected with *Hb9*::GFP lentivirus. MN survival was assessed using three independent biological replicates. High content imaging was performed 8 days after transfection, 15 days after neurotrophic factor withdrawal, and after 28 and 35 days of maturation. ImageJ was used to detect

**The paper explained**

**Problem**

The currently approved clinical treatment drugs for ALS, which focus on excitotoxicity and oxidative stress, have a poor therapeutic effect. It is necessary to identify new, more effective drugs targeting different molecular mechanisms and to explore their protective effect on ALS motor neurons.

**Results**

Our research has identified a new compound, isoginkgetin (ISO) is able to promote PINK1-Parkin-dependent mitophagy by stabilizing the PINK1/TOM complex. Through promoting mitophagy, ISO can degrade the accumulated damaged mitochondria and demonstrate its protective effects in our ALS cross-species platform that includes nematodes, mice, and ALS patient iPSC-derived MNs.

**Impact**

We observe that PINK1–Parkin-dependent mitophagy is impaired in ALS motor neurons; reactivation of the mitophagy pathway may serve as a druggable target in ALS.

*Hb9*::GFP-labeled MNs. ISO or DMSO was administered as described above.

### Neuronal mitophagy events detection

Mitophagy was detected in multiple model organisms using several methods as reported previously (Fang et al, 2019; Takahashi et al, 2019; Yang et al, 2025). Postmortem spinal cord samples of ALS patients and sex-matched healthy controls were obtained from the Netherlands Brain Bank (https://www.e-nbb.org, n = 3 per group), and the mitochondrial protein marker MTCO2 and the lysosomal protein marker LAMP2 were detected using immunofluorescence. The Pearson's correlation coefficient of co-localization between MTCO2 and LAMP2 was calculated in NeuN-positive neurons using ImageJ software (http://punias.free.fr/ImageJ/colocalization-finder.html) to assess the mitophagy events. In iPSC-derived MNs models, two methods were employed to assess the level of mitophagy. The first method involves using immunofluorescence to mark the MAP2-positive motor neurons and then analyzing the Pearson's correlation coefficient of MTCO2 and LAMP2 proteins to assess the mitophagy events. The second method involves the use of a commercial mitophagy detection kit (Dojindo; MD01). Mtphagy dye is a mitochondrion-localizing pH-sensitive probe. When mitophagy occurs, damaged mitochondria fuse with lysosomes, causing the mtphagy dye to produce a stronger fluorescence. Mitophagy events were evaluated by analyzing the average fluorescence intensity of mtphagy dye in *Hb9*::GFP lentivirus-labeled motor neurons. Neuronal mitophagy events were detected in mice using immunofluorescence to mark the ChAT-positive motor neurons and then analyzing the Pearson's correlation coefficient of MTCO2 and LAMP2 proteins to assess the mitophagy events. Neuronal mitophagy in *C. elegans* was measured using two strains. First, an mt-Rosella transgenic worm expressing a pan-neuronal mt-Rosella biosensor that combines a GFP variant sensitive to the acidic environment of the lysosomal lumen, fused to the pH-insensitive DsRed. Mitophagy was calculated as DsRed/DsRed+GFP, thus the higher the ratio of pixel intensity, the higher the level of mitophagy. Second, we generated transgenic nematodes expressing the DCT-1 mitophagy receptor fused with GFP or BFP, together with the autophagosome marker LGG-1 fused with DsRed in D-type motor neurons or pan-neurons. Mitophagy events were assessed using the co-localization puncta number between LGG-1 and DCT-1.

### Graphics

Graphics including synopsis, Figs. 7A, EV6C, and EV7K were created with BioRender.com.

### Statistical analysis

All values are presented as mean ± SD, unless otherwise specified. Unpaired *t* test was used for comparison between two independent samples. One-way analysis of variance (ANOVA) was used for comparison among multiple groups, followed by Dunnett's multiple comparisons test. Group differences were analyzed with two-way ANOVA followed by Tukey's multiple comparisons test. Survival curves were produced and compared using the Log-rank (Mantel–Cox) test. Prism 9.5 (GraphPad Software) was used for statistical analysis. The significance level is indicated in all statistical analyses. $P$ values < 0.05 were considered statistically significant.

## Data availability

This study includes no data deposited in external repositories.

The source data of this paper are collected in the following database record: biostudies:S-SCDT-10_1038-S44321-025-00323-2.

## Peer review information

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

## Acknowledgements

This work was supported by funding from the Macau Science and Technology Development Fund (0093/2024/AFJ), the Science and Technology Development Fund, Macau SAR (SKL-QRCM(UM)-2023-2025; 001/2023/ALC), the National Natural Science Foundation of China (#82271448), and MYRG-GRG2023-00152-ICMS-UMDF. EFF is supported by the Cure Alzheimer's Fund (#282952; #284930), Helse Sør-Øst (#2020001, #2021021, #2023093), the Research Council of Norway (#262175, #334361), Molecule AG/VITADAO (#282942), NordForsk Foundation (#119986), the National Natural Science Foundation of China (#81971327), Akershus University Hospital (#269901, #261973, #262960), the Civitan Norges Forskningsfond for Alzheimers sykdom (#281931), the Czech Republic-Norway KAPPA programme (with Martin Vyhnálek, #TO01000215), Horizon-TMA-MSCA-DN (#101073251, with Riekelt Houtkooper), and Wellcome Leap's Dynamic Resilience Program (jointly funded by Temasek Trust) (#104617).

## Author contributions

**Ang Li**: Data curation; Formal analysis; Investigation; Writing—original draft; Writing—review and editing. **Sen Huang**: Resources; Software; Investigation; Writing—original draft. **Shu-qin Cao**: Resources; Data curation; Investigation; Methodology. **Jinyi Lin**: Investigation; Methodology. **Linping Zhao**: Investigation; Methodology. **Feng Yu**: Data curation; Methodology. **Miaodan Huang**: Investigation. **Lele Yang**: Investigation; Methodology. **Jiaqi Xin**: Investigation. **Jing Wen**: Validation; Investigation. **Lingli Yan**: Investigation; Methodology. **Ke Zhang**: Data curation; Investigation; Methodology. **Maoyuan Jiang**: Investigation; Methodology. **Weidong Le**: Resources. **Peng Li**: Investigation; Methodology. **Yong U Liu**: Investigation; Methodology; Project administration. **Dajiang Qin**: Funding acquisition; Project administration. **Jiahong Lu**: Investigation; Methodology. **Guang Lu**: Investigation; Methodology; Writing—review and editing. **Hanming Shen**: Resources; Software; Supervision. **Xiaoli Yao**: Supervision; Funding acquisition; Methodology; Project administration. **Evandro F Fang**: Supervision; Funding acquisition; Investigation; Methodology. **Huanxing Su**: Conceptualization; Supervision; Funding acquisition; Project administration; Writing—review and editing.

Source data underlying figure panels in this paper may have individual authorship assigned. Where available, figure panel/source data authorship is listed in the following database record: biostudies:S-SCDT-10_1038-S44321-025-00323-2.

## Disclosure and competing interests statement

EFF is co-owner of Fang-S Consultation AS (Organization number 931 410 717) and NO-Age AS (Organization number 933 219 127). EFF also has an MTA with LMITO Therapeutics Inc (South Korea), a CRADA with ChromaDex (USA), a commercialization agreement with Molecule AG/VITADAO, and is a consultant to MindRank AI (China), NYO3 (Norway), and AgeLab (Vitality Nordic AS, Norway). SQC has a commercialization agreement with Molecule AG/VITADAO.

# Expanded View Figures

**Figure EV1. High-content drug screening identifies ISO as a potent PINK1-Parkin mitophagy agonist.**

(A) Flowchart for high-content drug screening. Scale bar, 25 μm. (B) Representative images of Top 20 mitophagy inducers. Scale bar, 50 μm. (C) YFP-Parkin-mt-mKeima HeLa cells were treated with ISO (10 μM) for 24 h. Flow cytometry was used to detect mt-mKeima indicated mitophagy signals. (D) The statistical results showed that ISO (10 μM) activated mitophagy ($n = 5$; five biological repeats). (E) Representative images of Parkin translocation induced by ISO (10 μM). Scale bar, 15 μm. (F) Quantification of (E) showed that ISO (10 μM) induced substantial Parkin translocation ($n = 5$; five biological repeats). (G) Three siRNAs targeting *PINK1* were transfected into YFP-Parkin-mt-mKeima HeLa cells with Lipo3000, and after 72 h transfection, CCCP was added into cells to induce PINK1-dependent mitophagy. Western blotting showed that PINK1 was successfully knocked down, and the expression of pSer65-Ub was also significantly inhibited. (H) siRNAs targeting *BNIP3*, *NIX*, *FUNDC1* and *BNIP3/NIX* double knock-down (KD) were transfected into YFP-Parkin-mt-mKeima HeLa cells with Lipo3000, and after 72 h transfection, Western blotting showed that each receptor protein was successfully knocked down. (I) Only *PINK1* KD in YFP-Parkin-mt-mKeima HeLa cells can block 10 μM ISO (8 h) induced mitophagy signals ($n = 5$; five biological repeats). Data are presented as the mean ± SD. Exact *P* values are reported in Appendix Table S1. One-way ANOVA followed by Dunnett's multiple comparisons test. Source data are available online for this figure.

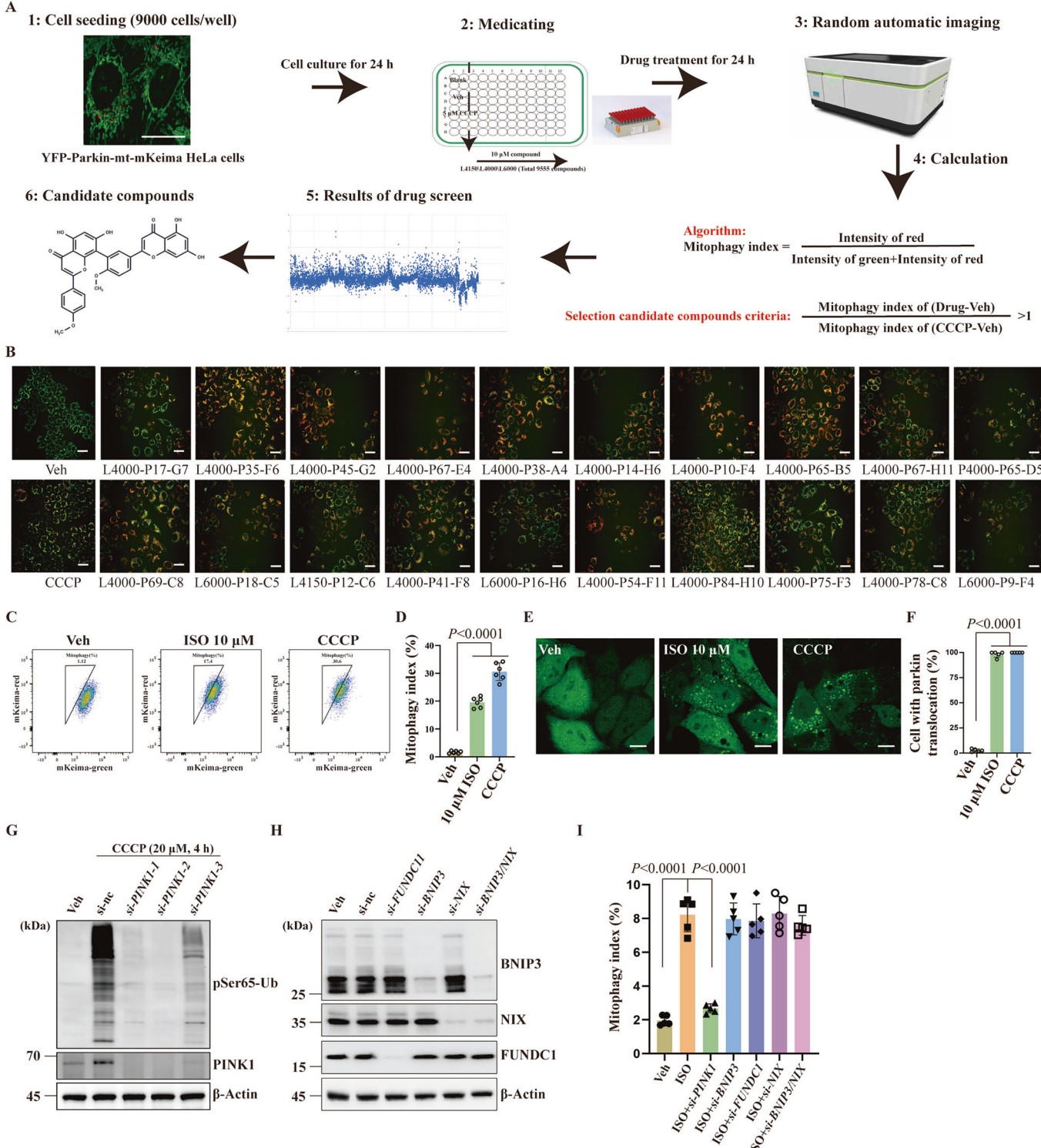

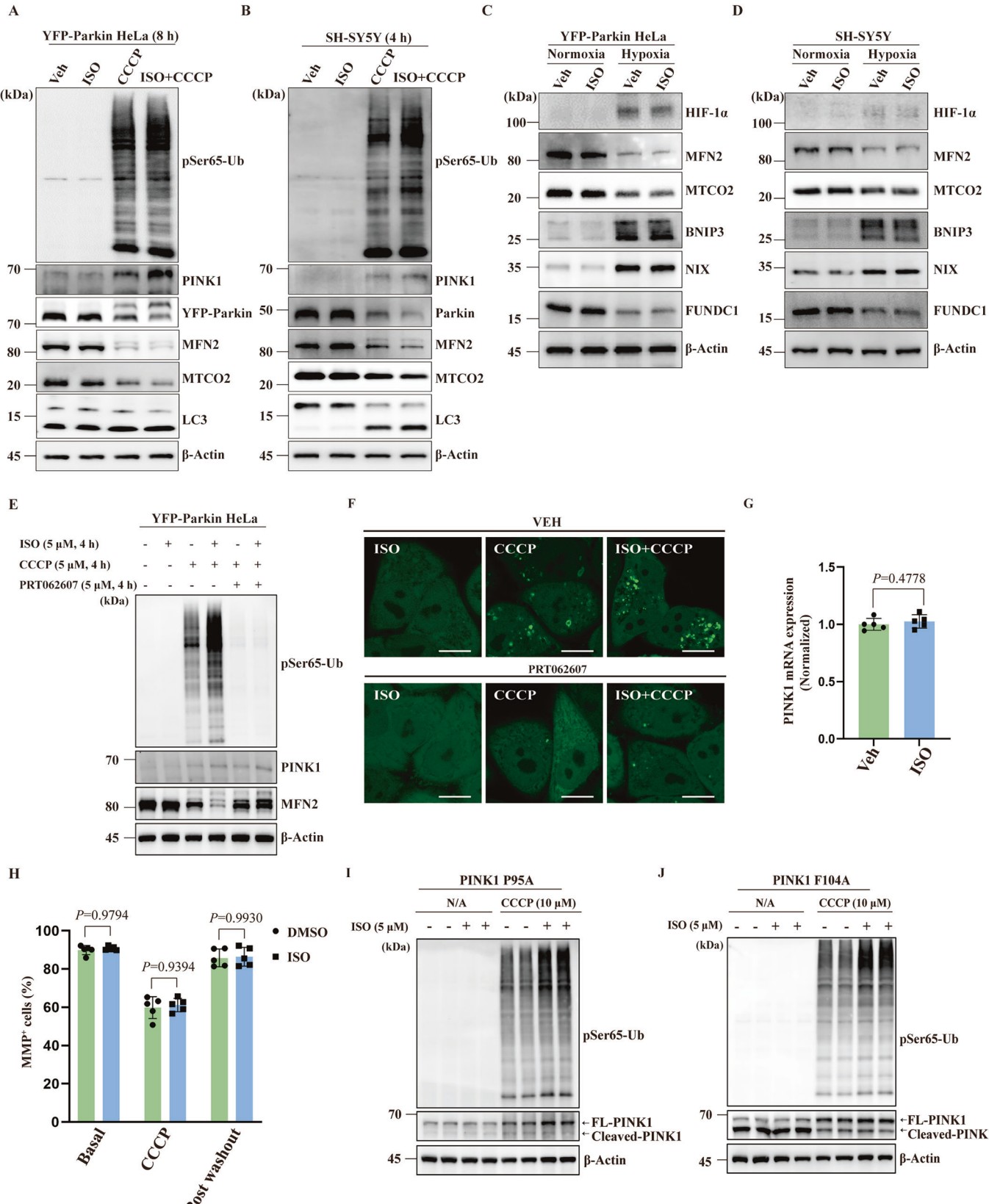

**Figure EV2.  ISO promotes mitophagy via mediating PINK1 function.**

(A) 5 μM ISO treatment for 8 h can promote CCCP-induced PINK1-Parkin-dependent mitophagy protein expression, mitochondrial membrane protein degradation and LC3 lipidation in YFP-Parkin HeLa cells. (B) 5 μM ISO treatment for 4 h can promote CCCP-induced PINK1-Parkin-dependent mitophagy protein expression, mitochondrial membrane protein degradation and LC3 lipidation in SH-SY5Y cells. (C) 5 μM ISO treatment for 24 h cannot promote hypoxia-induced receptor-mediated mitophagy protein expression in YFP-Parkin HeLa cells. (D) 5 μM ISO treatment for 24 h cannot promote hypoxia-induced receptor-mediated mitophagy protein expression in SH-SY5Y cells. (E) Immunoblotting showed that PRT, a PINK1 inhibitor, prevented the enhancing effect of ISO on CCCP-induced pSer65-Ub expression and MFN2 degradation, and Parkin translocation (F). (G) Relative mRNA expression levels of PINK1 in YFP-Parkin-PINK1-Myc HeLa cells between the Veh and ISO (5 μM, 4 h) group ($n = 5$; five biological repeats). (H) In a CCCP washout experiment, MMP was quantified by flow cytometry with TMRE probe at the indicated timepoints. Quantification of the TMRE intensity showed that ISO stabilized PINK1 without causing MMP collapse ($n = 5$; five biological repeats). (I, J) Immunoblotting showed that ISO was still able to increase pSer65-Ub expression in YFP-Parkin HeLa cells transfected with PINK1 P95A and F104A mutation plasmids. Data are presented as the mean ± SD. Exact *P* values are reported in Appendix Table S1. Unpaired *t* test in (G). Two-way ANOVA followed by Tukey's multiple comparisons test in (H). Source data are available online for this figure.

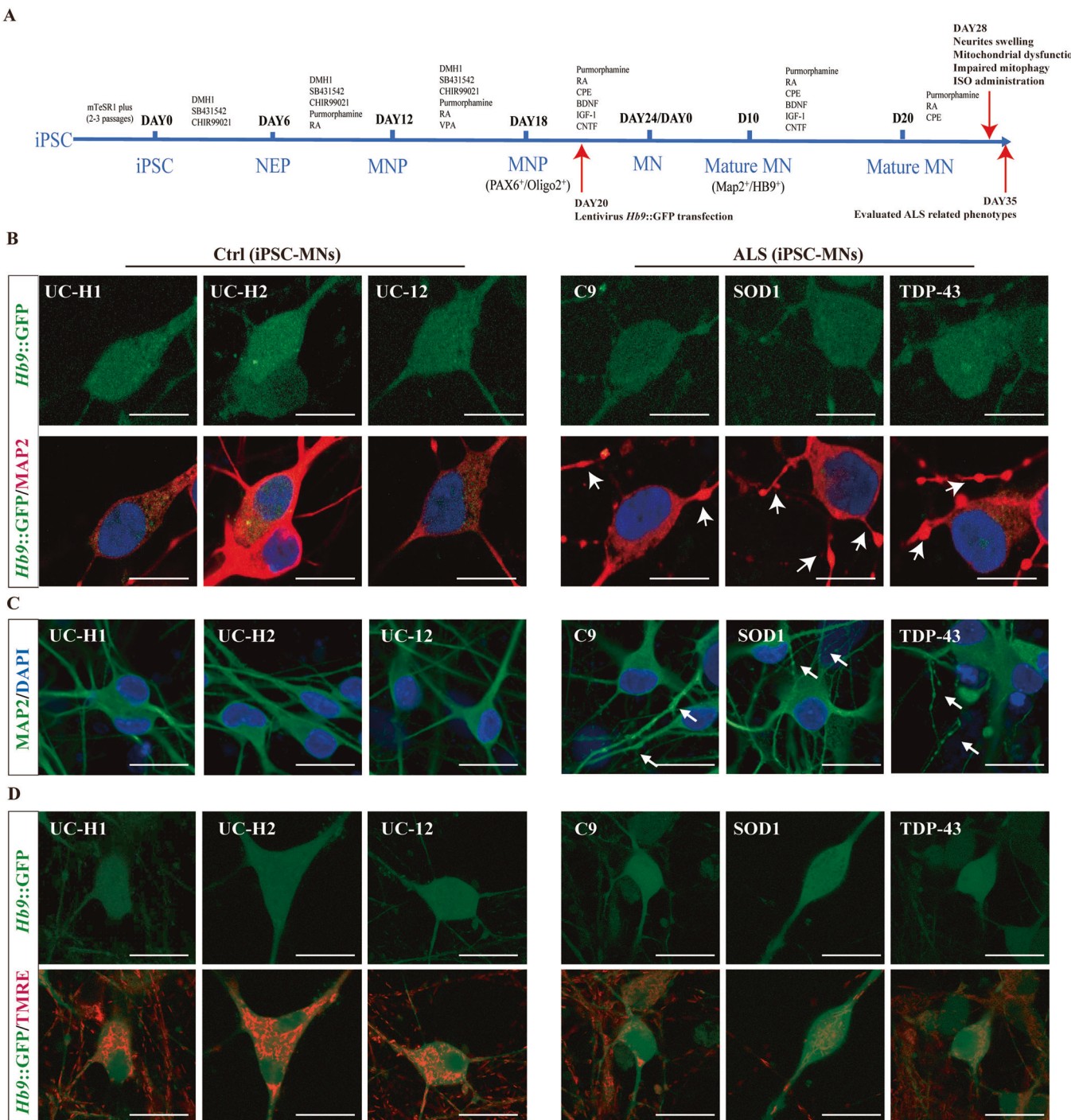

**Figure EV3. ALS-related phenotypes in three ALS patient iPSC lines derived MNs.**

(A) Schematic of the MN differentiation protocol. (B) OLIG2⁺ motor neuron progenitors (MNPs) were transfected with lentivirus *Hb9*::GFP and further induced to differentiate into MNs. Immunostaining showed MAP2⁺/GFP⁺ MNs were successfully differentiated, and neurites swelling with bead-like structures appeared in ALS MNs (white arrow) on day 28 at stage 5. Scale bars, 15 µm. (C) Representative images of neurites with bead-like structures (white arrow) in MAP2⁺ MNs on day 28 at stage 5. Scale bars, 20 µm. (D) Representative images of TMRE staining of GFP⁺ MNs showed that MMP was significantly reduced in ALS *Hb9*::GFP⁺ MNs on day 28 at stage 5. Scale bars, 20 µm.

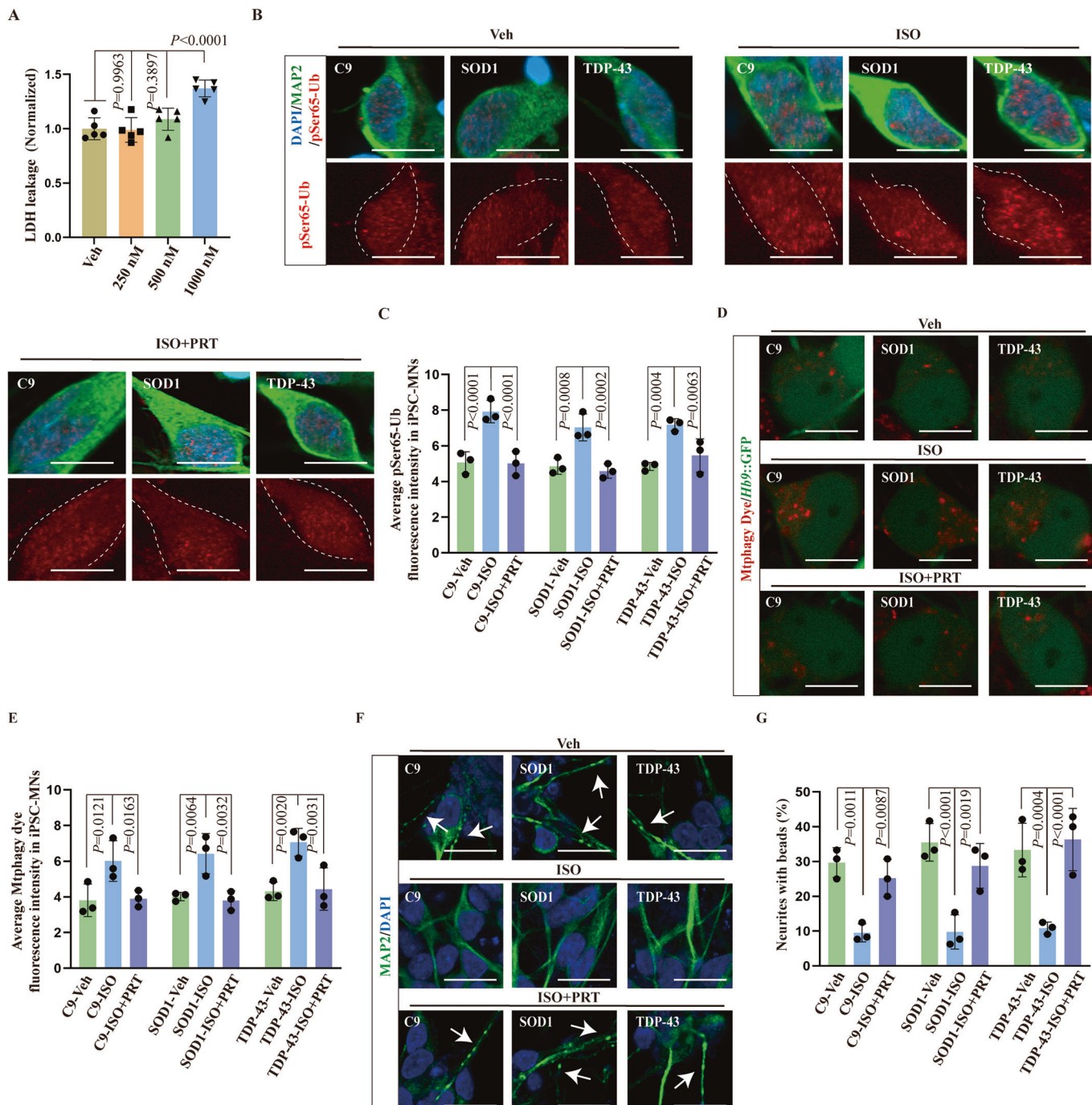

**Figure EV4.  PRT abolishes the beneficial effects of ISO in three ALS patient iPSC-derived MNs.**

(**A**) LDH leakage assay determined upon a working concentration of ISO (without neurotoxicity) of 0.5 μM (n = 5; three biological repeats). (**B, C**) ISO increased pSer65-Ub expression in three ALS patient iPSC lines derived MAP2+ MNs and PRT, a PINK1 inhibitor blocked the effect of ISO on increasing pSer65-Ub expression. MAP2+ motor neurons are marked with white dotted borders. Scale bar, 15 μm. Each point represents the average value of 10 MNs for each iPSC-derived MNs; three biological replicates for each group. (**D**) Representative living cell images of the mtphagy dye in three ALS patient iPSC-derived MNs (*Hb9*::GFP+). Scale bars, 15 μm. (**E**) Quantification of (**D**) the fluorescence intensity demonstrating that ISO enhanced mitophagy in three ALS patient iPSC-derived *Hb9*::GFP+ MNs, and PRT blocked this effect. Each point represents the average value of 10 MNs for each iPSC-derived MNs; three biological replicates for each group. (**F**) Representative images of neurites with swelling bead-like structures (white arrow) in three ALS patient iPSC lines derived MAP2+ MNs. Scale bars, 50 μm. (**G**) Quantification of (**F**) showing that ISO decreased neurites with swelling bead-like structures in three ALS patient iPSC line-derived MAP2+ MNs after 7 days of treatment and PRT abolished this effect. Each point represents the average value of 10 images for each iPSC-derived MNs; three biological replicates for each group. Data are presented as the mean ± SD. Exact *P* values are reported in Appendix Table S1. One-way ANOVA followed by Dunnett's multiple comparisons test in (**A**). Two-way ANOVA followed by Tukey's multiple comparisons test in (**C, E, G**).

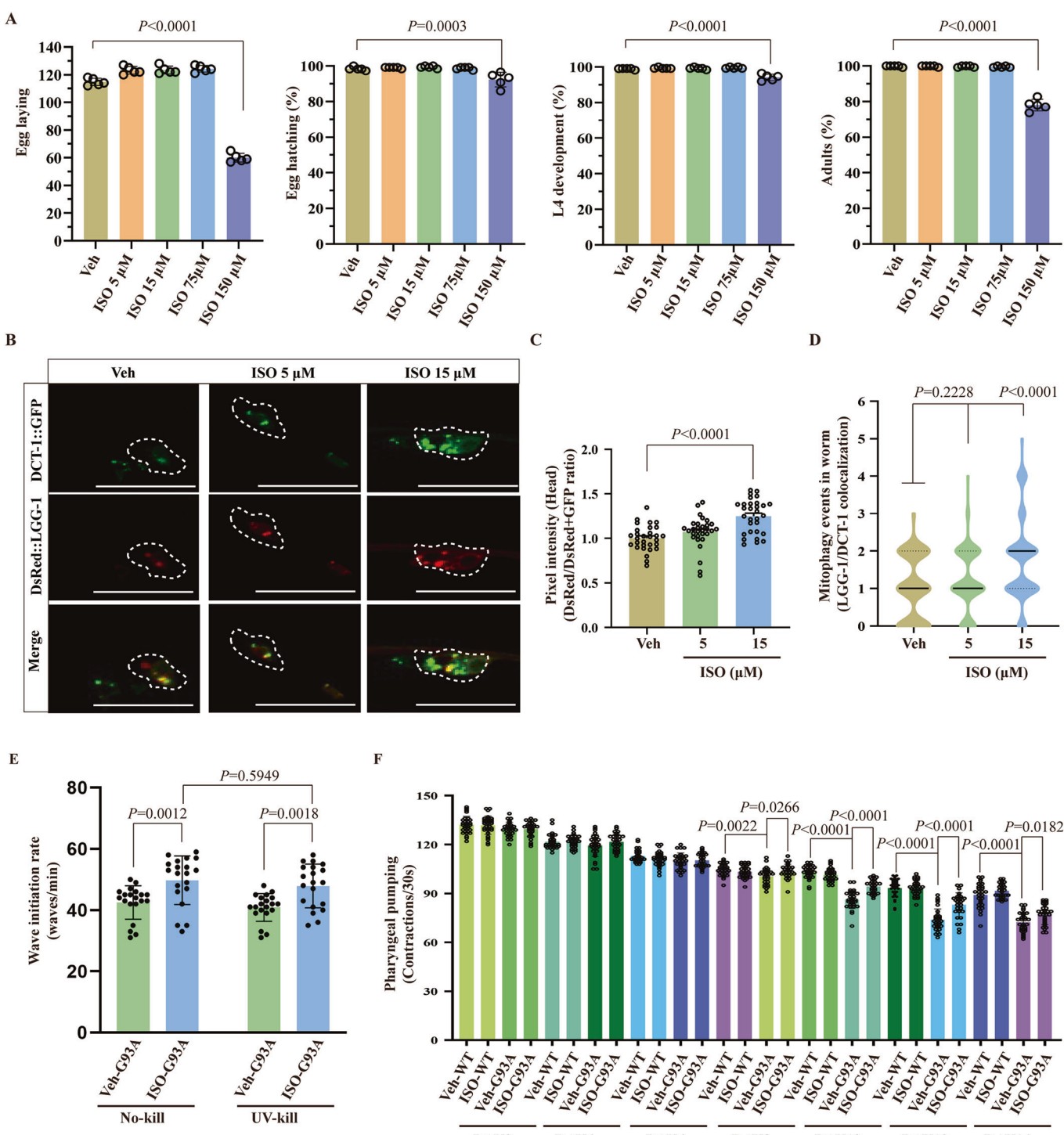

**Figure EV5. ISO induces neuronal mitophagy and ameliorates ALS-related phenotypes in *C. elegans*.**

(A) Effects of ISO (5 to 150 μM) on the egg laying rate, egg hatching rate (L1), L4 development rate, and the adulthood rate (Adult day 1) in the WT N2 nematodes. Toxicity tests revealed a safe concentration range of ISO between 0 and 75 μM ($n = 5$; 10 worms for each biological repeats). (B) Representative images showing the LGG-1 and DCT-1 co-localization in DMSO or ISO treatment group on adult day 1. *unc119*+ neurons are marked with white dotted borders. Scale bar, 5 μm. (C, D) Statistical results of mt-Rosella (C) and LGG-1/DCT-1 (D) transgenic worms showed that 15 μM ISO activated neuronal mitophagy ($n = 30$ neurons from 30 worms for each group). (E) UV-killed OP50 experiment showing that the protective effect of ISO on the motility of SOD1 G93A worms was independent of microbial metabolism ($n = 20$ worms for each group). (F) Effect of ISO on pharyngeal pumping speed from adult day 2 to day 14 in WT (N2) and SOD1 G93A worms. The results showed that ISO did not restrict the nematode diet ($n = 30$ worms for each group). Data are presented as the mean ± SD. Exact P values are reported in Appendix Table S1. One-way ANOVA followed by Dunnett's multiple comparisons test in (A, C, D, F). Two-way ANOVA followed by Tukey's multiple comparisons test in (E).

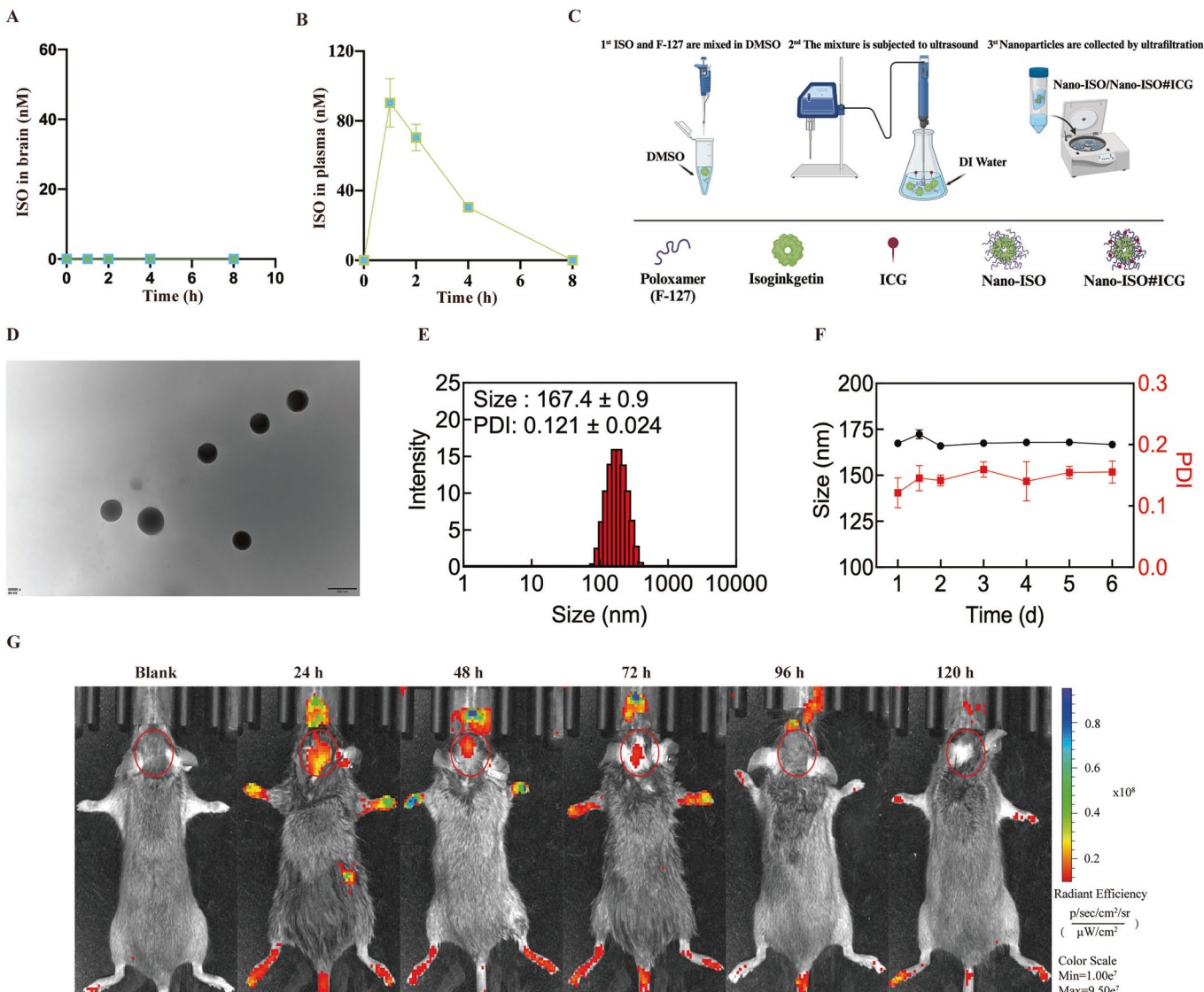

**Figure EV6. Synthesis and characterization of self-assembled Nano-ISO.**

(**A, B**) ISO concentrations in brain tissue (**A**) and plasma (**B**) detected by the UPLC-MS/MS assay after a single intraperitoneal injection of ISO ($n = 3$; three biological repeats). (**C**) Schematic illustration of the synthesis process of Nano-ISO and Nano-ISO#ICG. (**D**) TEM images of Nano-ISO in aqueous solutions at pH 7.4 for 24 h. (**E**) The size value and PDI of Nano-ISO were detected by particle size analyzer. (**F**) Within 6 days, the size of the Nano-ISO particles decreased gradually, and the PDI value increased gradually ($n = 3$; three biological repeats). (**G**) Representative IVIS images of the distribution of Nano-ISO#ICG over time in the mice after a single intranasal administration.

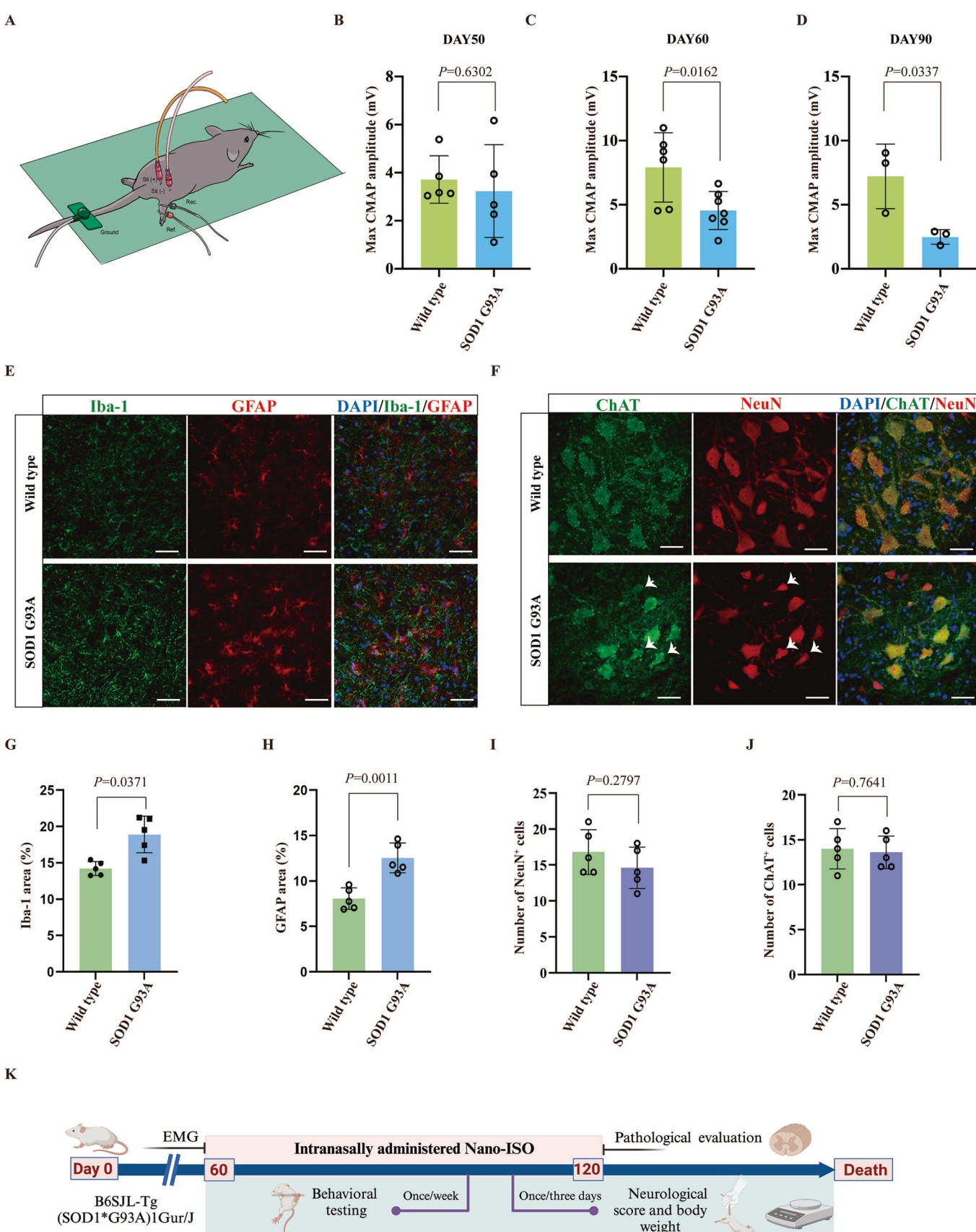

◀    **Figure EV7.   SOD1 G93A mice exhibit neurogenic damage and neuroinflammation at Day 60.**

(**A**) Schematic diagram of electrophysiological test in mice. (**B–D**) Statistical results of the CMAP amplitude of gastrocnemius muscle by stimulating sciatic nerve in SOD1 G93A mouse and WT group on Day 50 (**B**), Day 60 (**C**), and Day 90 (**D**). The maximum CMAP amplitude was noted to decrease significantly in SOD1 G93A mice on day 60, $n = 5$ mice/group on day 50; $n = 6$ mice for WT and $n = 7$ mice for G93A on day 60; $n = 3$ mice/group on day 90. (**E**) Representative images of microglia and astrocyte activation revealed by Iba-I staining and GFAP staining. Scale bar, 25 μm. (**F**) Representative images of ChAT and NeuN staining. White arrows indicate motor neurons with shrunken cell bodies. Scale bar, 25 μm. (**G, H**) Statistical results of Iba-I staining and GFAP staining in (**E**) demonstrating that both the microglia and astrocyte activation were detected in the ventral horn of lumbar spinal cord of SOD1 G93A mice on day 60. Each point represents the average value of 10 images for each mouse, $n = 5$ mice/group. (**I, J**) Statistical results of the number of MNs by NeuN staining and ChAT staining. No loss of MNs was detected in the ventral horn of lumbar spinal cord of SOD1 G93A mice on day 60. Each point represents the average value of 10 images for each mouse, $n = 5$ mice/group. (**K**) Schematic diagram of the exploration of Nano-ISO therapeutic effects in SOD1 G93A mice. Data are presented as the mean ± SD. Exact *P* values are reported in Appendix Table S1. Unpaired *t* test.

                                                                       