## [Peer Review File · EMBO Molecular Medicine]

Isoginkgetin antagonizes ALS pathologies in animal and patient iPSC models via PINK1-Parkin-dependent mitophagy

Huanxing Su, Ang Li, Sen Huang, Shuqin Cao, Jinyi Lin, Linping Zhao, Feng Yu, Miaodan Huang, Lele Yang, Jiaqi Xin, Jing Wen, Lingli Yan, Ke Zhang, Maoyuan Jiang, Weidong Le, Peng Li, Yong Liu, Dajiang Qin, Jia-Hong Lu, Guang Lu, Han-Ming Shen, Xiaoli Yao, and Evandro Fang

Corresponding author(s): Huanxing Su (huanxingsu@um.edu.mo), Evandro Fang (e.f.fang@medisin.uio.no), Xiaoli Yao (yaoxiaol@mail.sysu.edu.cn)

Review Timeline:

Submission Date:	1st May 25
Editorial Decision:	22nd May 25
Revision Received:	21st Aug 25
Editorial Decision:	22nd Sep 25
Revision Received:	26th Sep 25
Accepted:	1st Oct 25

Editor: Zeljko Durdevic

Transaction Report:

22nd May 2025

Dear Dr. Su,

Thank you for the submission of your manuscript to EMBO Molecular Medicine. We have now received feedback from the three reviewers who agreed to evaluate your manuscript. All three referees recognize interest of the study but also raise serious and partially overlapping concerns that should be addressed in a major revision. If you would like to discuss further the points raised by the referees, I am available to do so via email or video. Let me know if you are interested in this option.

We would welcome the submission of a revised version within three months for further consideration. Please let us know if you require longer to complete the revision.

I look forward to receiving your revised manuscript.

Yours sincerely,

Zeljko Durdevic

Zeljko Durdevic
Senior Editor
EMBO Molecular Medicine

We require:

- 1) A .docx formatted version of the manuscript text (including legends for main figures, EV figures and tables). Please make sure that the changes are highlighted to be clearly visible.
- 2) Individual production quality figure files as .eps, .tif, .jpg (one file per figure). For guidance, download the 'Figure Guide PDF': (<https://www.embopress.org/page/journal/17574684/authorguide#figureformat>).
- 3) A .docx formatted letter INCLUDING the reviewers' reports and your detailed point-by-point responses to their comments. As part of the EMBO Press transparent editorial process, the point-by-point response is part of the Review Process File (RPF), which will be published alongside your paper.
- 4) A complete author checklist, which you can download from our author guidelines (<https://www.embopress.org/page/journal/17574684/authorguide#submissionofrevisions>). Please insert information in the checklist that is also reflected in the manuscript. The completed author checklist will also be part of the RPF.
- 5) Please note that all corresponding authors are required to supply an ORCID ID for their name upon submission of a revised manuscript.
- 6) It is mandatory to include a 'Data Availability' section after the Materials and Methods. Before submitting your revision, primary datasets produced in this study need to be deposited in an appropriate public database, and the accession numbers and

database listed under 'Data Availability'. Please remember to provide a reviewer password if the datasets are not yet public (see <https://www.embopress.org/page/journal/17574684/authorguide#dataavailability>).

12) Author contributions: You will be asked to provide CRediT (Contributor Role Taxonomy) terms in the submission system. These replace a narrative author contribution section in the manuscript.

13) A Conflict of Interest statement should be provided in the main text.

14) Every published paper now includes a 'Synopsis' to further enhance discoverability. Synopses are displayed on the journal webpage and are freely accessible to all readers. They include a short stand first (maximum of 300 characters, including space) as well as 2-5 one-sentences bullet points that summarizes the paper. Please write the bullet points to summarize the key NEW findings. They should be designed to be complementary to the abstract - i.e. not repeat the same text. We encourage inclusion of key acronyms and quantitative information (maximum of 30 words / bullet point). Please use the passive voice. Please attach these in a separate file or send them by email, we will incorporate them accordingly.

15) Include a Reagents and Tools Table as part of the Methods section, which can be downloaded from our author guidelines (<https://www.embopress.org/page/journal/17574684/authorguide#structuredmethods>)

***** Reviewer's comments *****

Referee #1 (Comments on Novelty/Model System for Author):

ALS-associated phenotypes in *C. elegans*, SOD1-G93A transgenic mice, and patient-derived iPSC motor neurons across three genotypes (C9orf72, SOD1, TDP-43).

Referee #1 (Remarks for Author):

This study identifies Isoginkgetin (ISO), a bioactive flavonoid derived from Ginkgo biloba, as a novel and potent inducer of PINK1-Parkin-dependent mitophagy. The authors demonstrate that ISO robustly stabilizes the PINK1/TOM complex, enhances mitophagy, and improves mitochondrial quality control without inducing overt mitochondrial toxicity. Notably, ISO ameliorates ALS-associated phenotypes in *C. elegans*, SOD1-G93A transgenic mice, and patient-derived iPSC motor neurons representing three major genotypes (C9orf72, SOD1, and TDP-43). The development of self-assembled Nano-ISO overcomes blood-brain barrier limitations, providing a compelling translational delivery strategy. Overall, this is a well-designed, mechanistically insightful, and translationally relevant study. The identification of ISO as a mitophagy activator with broad efficacy across ALS models is significant.

Comments and Suggestions Clarify the direct molecular target of ISO:

The mechanistic data convincingly show that ISO promotes mitophagy through stabilization of the PINK1/TOM complex and upregulation of phosphorylated ubiquitin (pSer65-Ub). While ISO's effect on the PINK1/TOM complex is clearly demonstrated, its direct molecular target remains undefined. A brief discussion of known or proposed mechanisms of ISO, or acknowledgement of this gap, would strengthen the manuscript.

The inclusion of Nano-ISO as a delivery system adds further innovation and clinical relevance. However, the discussion would benefit from a more detailed comparison between ISO and currently approved ALS treatments, particularly in terms of mechanism and potential for synergy or additive effects.

Finally, the study is focused on PINK1-Parkin, but omits analysis of other mitophagy pathways (e.g., BNIP3/NIX, FUNDC1). These factors can be tested to improve mechanistic specificity.

Referee #2 (Remarks for Author):

This is an interesting paper that explores a novel therapeutic approach for ALS, focusing on enhancement of mitophagy - the process of selectively removing damaged mitochondria from cells. An extensive screening of nearly 10,000 compounds, lead the authors to identify isoginkgetin (ISO) as a potent mitophagy inducer, which proved to be protective in several in vitro and in vivo models of ALS. The study, spanning from molecular mechanisms to animal models and human cell lines, is convincing and sheds light on the role of impaired mitophagy in ALS pathology, opening up new avenues for developing treatments based on mitophagy enhancement.

In this respect, the study is novel and deserves publication.

We do have however a number of crucial requests before the paper can be accepted for publication.

GENERAL/MAJOR COMMENTS:

-One major concern is that the authors did not describe the molecular mechanism underlying PINK1/Parkin impairment in ALS models. This is crucial to clearly dissect the molecular mechanism of protection triggered by ISO. In fact, if PINK1 activity is impaired in models of ALS, and ISO-dependent protective effect depends on the mitophagic effect of PINK1, unfunctional PINK1 needs to become functional upon ISO treatment. How is this achieved? The molecular mechanism triggered by ISO underlying

this inactive/active switch of PINK1 is completely unexplored.

-Authors did not take into consideration the existence of mitophagic pathways other than PINK1/Parkin, which can be activated by ISO. The activation of these alternative pathways can compensate for PINK1 impairment in ALS models, and be protective in the absence of PINK1. Although authors did show that ISO promotes PINK1/Parkin mitophagy, this does not exclude the possibility that ISO might also activate alternative mitophagic pathways that can compensate for impaired PINK1 in ALS models.

-Stat needs to be seriously improved. Most experiments have been repeated only two to three times, a minimum of five independent biological replicates are mandatory to perform reliable stat analysis.

-Introduction and Discussion lacks a comprehensive analysis of pertinent literature and an in-depth examination of the study's findings. Both sections would benefit from a more rigorous approach, enhanced clarity in language use, and improved sentence structure.

ADDITIONAL POINTS

-The order used to present the results is not clear. In the first part of the results the authors showed that ISO is able to enhance mitophagy in vitro and they provide a possible molecular mechanism, subsequently in the fourth chapter of the results (Defective PINK1-Parkin-dependent mitophagy in ALS MNs) they showed that mitophagy is impaired in MNs derived from ALS patient, however the results that showed that ISO enhances mitophagy in these models are at the very end, after all the in vivo studies. Consider moving the last chapter after the fourth to improve clarity.

-Please describe in more details in Materials and Methods how the different mitophagy indexes/mitophagy events are calculated.

-In Figure 4H the number of LAMP2 puncta in ALS models seems to be lower compared to controls, an effect that appears to be reversed by ISO treatment (Figure 7C). The same effect is also visible in the mouse model (Figure 6L). Could you please comment on this? Does the number of LAMP2 puncta in these models depend on mitophagy levels? Could ISO also affect general autophagy and/or lysosomal biogenesis?

-The DCT-1/LGG-1 and mt-Rosella C. elegans models should be better described.

-Experiments in Figure 5 (from D to J) are performed only on SOD1 G93A transgenic worms. It is important to also add WT as control (treated with vehicle as well as ISO). Please add representative images for quantification in Figure 5J. It is not clear if the KD of Pink-1 and pdr-1 in worm MNs (Figure 5J-K) is performed on WT or SOD1 G93A mutant, please clarify this point and add the appropriate controls (vehicle-Pink-1, vehicle-pdr-1, vehicle-WT, ISO-WT), which are missing in the analysis. Figure S4C-D the higher concentration of ISO used as reported in the text is 15 μ M, correct in the figure.

-Please clarify in the text that pdr-1 is the homolog of Parkin.

-Figure 6D: the y-axis is labeled "Time to disease onset", but it should refer to the percentage of "healthy" mice. Please modify.

-A very critical point arises from the data of the experiments performed using iPSCs-derived MNs (Figure 4H-O and Figure 7). In Figure 4H-O it appears that the 3 cell lines deriving from patients with 3 different mutations are pooled together and used as biological replicates. The same is true for Figure 7A-C, while in Figure 7E-K the results from the 3 cell lines are kept separate. Moreover, based on the figure legend description, the dots in the quantification images correspond to single images/MNs, they do not represent biological replicates so it is not possible to perform a proper statistical analysis on these data. Please revise all data from Figure 4 and 7 taking into account that to perform a statistical analysis it is required to have three to five biological replicates for each sample, and each biological replicate should come from the analysis of at least 10 images/MNs. Ideally, all data should be presented as in Figure 7F-H-I-K. It might be acceptable to pool together data from the 3 different cell lines and consider them as ALS patients, however it is necessary to increase the number of images analysed and increase the number of biological replicates for each experiment. Finally, quantification and statistical analysis of the results needs to be done on biological replicates not technical/single images.

-The same critical point applies to some of the experiments performed in mice (Figure 6G-O and Figure S6E-J). In Figure 6G-H, each data point represents a single individual (n = 5 mice) and thus they can be considered biological replicates. In Figure 6I-O and S6E-J, however, the data points (n = 10-20) appear to refer to single images/MNs derived from n = 5 mice or n = 3 mice, so they do not represent biological replicates. Quantification and statistical analysis of data from Figure 6I-O and S6E-J need to be performed on the biological replicates (as in Figure 6G-H). Figure legends should indicate how many images/MNs were quantified per biological replicate.

Referee #3 (Comments on Novelty/Model System for Author):

while the identification of ISO as a mitophagy enhancer with therapeutic potential in ALS is exciting, the current dataset lacks several key functional validations. More rigorous experimentation is required to establish mitochondrial integrity, validate mitophagy enhancement, and clarify ISO's precise molecular mechanism.

Referee #3 (Remarks for Author):

The study conducted by Dr. Ang Li identifies Isoginkgetin (ISO), a bioflavonoid derived from Ginkgo biloba, as a potent inducer of mitophagy. ISO exerts its effects primarily through the enhancement of the PINK1-Parkin pathway and stabilization of the PINK1/TOM complex. The authors further demonstrate that ISO mitigates ALS-related pathology in both cell and animal models by rescuing defective mitophagy and enhancing mitochondrial function. These findings are certainly compelling and indicate a potential therapeutic avenue for ALS. However, while the conclusions are promising, there are several critical concerns that must be addressed to validate the claims and strengthen the mechanistic insights presented.

Foremost, the manuscript emphasizes that ISO promotes mitophagy "without compromising the normal function of mitochondria." Unfortunately, this claim is not adequately substantiated, as no direct mitochondrial function assays are provided. While quantification of mitochondrial membrane potential (MMP) or MTCO2 expression provides limited information, these metrics alone are insufficient to draw definitive conclusions about mitochondrial health or regulation. More comprehensive assessments are needed, such as Mitochondrial respiration analysis (e.g., Seahorse assay) and Assessment of mitochondrial biogenesis content via mtDNA copy number

The authors report that ISO at concentrations up to 10 μ M maintains MMP and enhances mitophagy in cells treated with CCCP, a mitochondrial uncoupler. While these findings are notable, it is equally important to evaluate whether ISO can promote mitophagy in basal conditions (i.e., in the absence of CCCP or other stressors), which would provide insight into its potential as a physiological modulator of mitochondrial quality control.

Additionally, the role of mitophagy in ALS remains highly debated. While multiple studies report impaired PINK1/Parkin signaling and reduced MMP in ALS cellular models, some patient-derived cells exhibit increased mitophagy activity and hyperpolarized mitochondria. Furthermore, emerging data suggest that mitochondrial dysfunction in ALS may not be a static phenotype, but rather a progressive and heterogeneous process that evolves over the course of the disease. Given these complexities, mitophagy defects in ALS models should be corroborated using a broader range of quantitative assays. Co-localization of MTCO2 and LAMP2, while informative, lacks the resolution and specificity required for definitive mitophagy quantification. More robust assays are necessary to firmly establish mitophagy dynamics and validate ISO's mode of action.

Other questions:

The manuscript also notes a lack of LC3-II accumulation in cells treated with 5 μ M CCCP despite observed Parkin recruitment. This raises important questions about the downstream autophagic flux. Is ISO uncoupling Parkin recruitment from canonical autophagosome formation, or is there an alternative mitophagy pathway being activated? Furthermore, the role of ISO in modulating Parkin stability or recruitment efficiency in the context of wild-type versus mutant PINK1 remains unexplored.

Point-to-point responses to the reviewers' comments

Referee #1 (Comments on Novelty/Model System for Author):

ALS-associated phenotypes in *C. elegans*, SOD1-G93A transgenic mice, and patient-derived iPSC motor neurons across three genotypes (C9orf72, SOD1, TDP-43).

Referee #1 (Remarks for Author):

This study identifies Isoginkgetin (ISO), a bioactive flavonoid derived from *Ginkgo biloba*, as a novel and potent inducer of PINK1-Parkin-dependent mitophagy. The authors demonstrate that ISO robustly stabilizes the PINK1/TOM complex, enhances mitophagy, and improves mitochondrial quality control without inducing overt mitochondrial toxicity. Notably, ISO ameliorates ALS-associated phenotypes in *C. elegans*, SOD1-G93A transgenic mice, and patient-derived iPSC motor neurons representing three major genotypes (C9orf72, SOD1, and TDP-43). The development of self-assembled Nano-ISO overcomes blood-brain barrier limitations, providing a compelling translational delivery strategy. Overall, this is a well-designed, mechanistically insightful, and translationally relevant study. The identification of ISO as a mitophagy activator with broad efficacy across ALS models is significant.

Response: Thank you for your very positive evaluation of our discovery and your values in both science and translation.

Comments and Suggestions Clarify the direct molecular target of ISO:

1. The mechanistic data convincingly show that ISO promotes mitophagy through stabilization of the PINK1/TOM complex and upregulation of phosphorylated ubiquitin (pSer65-Ub). While ISO's effect on the PINK1/TOM complex is clearly demonstrated, its direct molecular target remains undefined. A brief discussion of known or proposed mechanisms of ISO, or acknowledgement of this gap, would strengthen the manuscript.

Response: Thank you for your instructive suggestion. We agree with the gap, and have presented a hypothesis on the possible mechanism by which ISO might stabilize the PINK1/TOM complex in the Discussion section of the manuscript (in green color):

“The direct molecular target(s) of ISO, including potential binding sites on/in mitochondria, remain unidentified. In the context of PINK1-Parkin-mediated mitophagy, PINK1 is normally imported into mitochondria via the TOM and TIM complexes and subsequently cleaved and degraded in healthy mitochondria. However, PINK1 instead accumulates on the outer mitochondrial membrane when mitochondrial membrane potential is lost, and forming a super complex with TOM subunits such as TOM20, which facilitates PINK1 activation, thereby initiating Parkin recruitment and mitophagy [73]. There are no interactions between ISO and TOM complex subunit that have been established, but it is noteworthy that compounds targeting the TOM complex can enhance PINK1 stabilization [37] suggesting a potential, but as yet untested mechanistic link.”

2. The inclusion of Nano-ISO as a delivery system adds further innovation and clinical relevance. However, the discussion would benefit from a more detailed comparison between ISO and currently approved ALS treatments, particularly in terms of mechanism and potential for synergy or additive effects.

Response: In the revised discussion section, we have added a paragraph focusing on the comparison of mechanisms and potential functions between ISO and two US FDA-approved drugs:

“It is believed that the excitatory amino acid neurotransmitter glutamate is involved in the pathogenesis of ALS, and accordingly one US FDA-approved anti-ALS drug is the anti-glutamate agent Riluzole [61]. While oxidative stress plays a role in ALS progression, Edaravone, an antioxidant compound was approved as a 2nd ALS drug [62]. However, these two anti-ALS drugs do not show strong effects for stopping or even slowing down the progress of ALS, possibly due to ALS high heterogeneity and fast disease progress [63]. Thus, it is important to identify novel and robust anti-ALS drugs. Our etiological focus on ALS is mitochondrial dysfunction in ALS motor neurons, and correspondingly, its links to excitotoxicity and oxidative stress [64, 65]. Our ISO data enable us to propose that combining antioxidants and/or anti-excitotoxic agents with strategies that enhance mitophagy to remove damaged mitochondria may offer additive/synergistic therapeutic outcomes in some ALS patients, such as those who are suffering with mutations of specific genes. Additionally, the intranasal delivery employed by Nano-ISO can enhance cerebral distribution and reduce the risk of peripheral side effects, thereby providing improved benefits and adaptability in long-term treatment.”

3. Finally, the study is focused on PINK1-Parkin, but omits analysis of other mitophagy pathways (e.g. BNIP3/NIX, FUNDC1). These factors can be tested to improve mechanistic specificity.

Response: We have performed additional experiments to investigate whether ISO-induced mitophagy involves other pathways such as receptor-mediated mitophagy. Our western blotting data shows ISO had no effect on the expression of other key mitophagy proteins including BNIP3, NIX and FUNDC1 (revised Figure 1F, also below).

Figure 1: Identification of small molecule compound ISO as a PINK1-Parkin-dependent mitophagy inducer. (F) YFP-Parkin HeLa cells were treated with ISO (10 μ M) for 24 h. Immunoblots of the indicated proteins. Treatment with 20 μ M CCCP (4 h) served as positive control.

In addition, we knocked down *PINK1*, *BNIP3*, *NIX*, *FUNDC1*, and *BNIP3/NIX* in YFP-Parkin-mt-mKeima HeLa cells. Importantly, ISO-induced mitophagy was abolished only under *PINK1* knocked down conditions, indicating that ISO-induced mitophagy is specifically *PINK1*-dependent (new Figure EV1G-I, and below).

Supplementary Figure EV1: High-content drug screening identifies ISO as a potent PINK1-Parkin mitophagy agonist. (G) Three siRNAs targeting *PINK1*

were transfected into YFP-Parkin-mt-mKeima HeLa cells with Lipo3000, and after 72 h transfection, CCCP was added into cells to induce *PINK1*-dependent mitophagy. Western blotting showed that *PINK1* was successfully knocked down, and the expression of pSer65-Ub was also significantly inhibited. (H) siRNAs targeting *BNIP3*, *NIX*, *FUNDC1* and *BNIP3/NIX* double knock-down (KD) were transfected into YFP-Parkin-mt-mKeima HeLa cells with Lipo3000, and after 72 h transfection, Western blotting showed that each receptor protein was successfully knocked down. (I) Only *PINK1* KD in YFP-Parkin-mt-mKeima HeLa cells can block 10 μ M ISO (8 h) induced mitophagy signals ($n = 5$; five biological repeats). Data are presented as the mean \pm SD. One-way ANOVA followed by Dunnett's multiple comparisons test.

Referee #2 (Remarks for Author):

This is an interesting paper that explores a novel therapeutic approach for ALS, focusing on enhancement of mitophagy - the process of selectively removing damaged mitochondria from cells. An extensive screening of nearly 10,000 compounds, lead the authors to identify isoginkgetin (ISO) as a potent mitophagy inducer, which proved to be protective in several in vitro and in vivo models of ALS. The study, spanning from molecular mechanisms to animal models and human cell lines, is convincing and sheds light on the role of impaired mitophagy in ALS pathology, opening up new avenues for developing treatments based on mitophagy enhancement.

In this respect, the study is novel and deserves publication.

Response: Thank you very much for your very positive review of our study which took us over 5 years of hard work; we really appreciate it.

We do have however a number of crucial requests before the paper can be accepted for publication.

GENERAL/MAJOR COMMENTS:

1. One major concern is that the authors did not describe the molecular mechanism underlying PINK1/Parkin impairment in ALS models. This is crucial to clearly dissect the molecular mechanism of protection triggered by ISO. In fact, if PINK1 activity is impaired in models of ALS, and ISO-dependent protective effect depends on the mitophagic effect of PINK1, unfunctional PINK1 needs to become functional upon ISO treatment. How is this achieved? The molecular mechanism triggered by ISO underlying this inactive/active switch of PINK1 is completely unexplored.

Response: We fully agree that clearly demonstrating the molecular mechanism of PINK1 inactivation is a significant and challenging task. Various hypotheses have been proposed regarding mitophagy dysfunction in neurodegenerative diseases including ALS. Our present study has described the phenomenon of impaired PINK1-Parkin mitophagy that we have observed in ALS patient samples and in the ALS patient iPSC derived motor neurons. In the revised version, we added:

Introduction section

“Impaired PINK1-Parkin-dependent mitophagy, characterized by the decreased expression of FL-PINK1, has been reported in various ALS disease models [29-31]. The specific molecular mechanism remains unknown, and no available drugs are currently known to regulate mitophagy in ALS by activating PINK1 expression or through enhancing the activity of FL-PINK1.”

Discussion section

“However, the molecular mechanisms leading to impaired PINK1-Parkin-dependent mitophagy in ALS MNs are still not clear. We therefore propose two potential hypotheses describing the reduction in PINK1-mitophagy activity in ALS: a) increased cleavage of PINK1 by PARL [67] and/or protein degradation systems (such as UPS

and autophagy itself), and b) compromised homeostasis of FL-PINK1 and its different forms (e.g., pathological TDP-43 may impair PINK1 degradation resulting in increased abundance of cytoplasmic cleaved-PINK and compromised mitophagy) [68]. These hypotheses should be tested in the future.”

Regarding how dysfunctional PINK1 becomes functional after ISO treatment, we have provided a detailed explanation on how FL-PINK1 is activated and involved in the process of mitophagy in the revised version. The detailed information is found in the Introduction of the revised manuscript as follows:

“The PTEN induced kinase 1 (PINK1) and parkin RBR E3 ubiquitin protein ligase (Parkin) pathway is one of the best characterized ubiquitin-mediated mitophagy pathways [12]. In this pathway, full-length PINK1 (FL-PINK1) stabilizes on the outer mitochondria membrane (OMM), forming a high molecular weight complex with the translocase of the outer mitochondrial membrane (TOM) [13], where it dimerizes and auto-activates its kinase function via autophosphorylation [14]. Then, the activated PINK1 phosphorylates ubiquitin at Serine 65 (pSer65-Ub) and the ubiquitin-like domain of Parkin, which activates a positive feed forward loop, initiating mitophagy [15].”

DOI:10.1016/j.molcel.2021.11.012

Activation of PINK1 requires its stabilization on the outer mitochondrial membrane, where it forms a complex with the TOM machinery– known as PINK1/TOM complex, prior to initiating its kinase activity. In our study, we show ISO stabilizes the PINK1/TOM complex (illustrated in Figure 3).

Therefore, we conclude that the impaired PINK1-Parkin-mediated mitophagy in ALS may arise through multiple mechanisms, such as loss-of-function mutations in PINK1, reduced protein stability or expression of PINK1. Among them, reduced expression levels of PINK1 have been widely reported. Based on our findings, we propose that ISO enhances the stabilization of PINK1/TOM complex, thereby maximizing the residual kinase activity of PINK1 in ALS MNs. ISO appears to promote PINK1-dependent mitophagy by indirectly supporting its membrane stabilization, rather than directly enhancing PINK1’s kinase activity.

2. Authors did not take into consideration the existence of mitophagic pathways other than PINK1/Parkin, which can be activated by ISO. The activation of these alternative pathways can compensate for PINK1 impairment in ALS models and be protective in the absence of PINK1. Although authors did show that ISO promotes PINK1/Parkin mitophagy, this does not exclude the possibility that ISO might also activate alternative mitophagic pathways that can compensate for impaired PINK1 in ALS models.

Response: The first referee also mentioned this issue. So we have performed additional experiments to investigate whether ISO is involved in other mitophagy pathways such as receptor-mediated mitophagy. Our western blotting data shows ISO had no effect on the expression of other key mitophagy proteins including BNIP3, NIX and FUNDC1 (revised Figure 1F, also below).

Figure 1: Identification of small molecule compound ISO as a PINK1-Parkin-dependent mitophagy inducer. (F) YFP-Parkin HeLa cells were treated with ISO (10 μM) for 24 h. Immunoblots of the indicated proteins. Treatment with 20 μM CCCP (4 h) served as positive control

In addition, we knocked down *PINK1*, *BNIP3*, *NIX*, *FUNDC1*, and *BNIP3/NIX* in YFP-Parkin-mt-mKeima HeLa cells. Importantly, ISO-induced mitophagy was abolished only under *PINK1* knocked down conditions, indicating that ISO-induced mitophagy is specifically *PINK1*-dependent (new Figure EV1G-I, and below).

Supplementary Figure EV1: High-content drug screening identifies ISO as a potent PINK1-Parkin mitophagy agonist. (G) Three siRNAs targeting *PINK1* were transfected into YFP-Parkin-mt-mKeima HeLa cells with Lipo3000, and after 72 h transfection, CCCP was added into cells to induce PINK1-dependent mitophagy. Western blotting showed that PINK1 was successfully knocked down, and the expression of pSer65-Ub was also significantly inhibited. (H) siRNAs targeting *BNIP3*, *NIX*, *FUNDC1* and *BNIP3/NIX* double knock-down (KD) were transfected into YFP-Parkin-mt-mKeima HeLa cells with Lipo3000, and after 72 h transfection, Western blotting showed that each receptor protein was successfully knocked down. (I) Only *PINK1* KD in YFP-Parkin-mt-mKeima HeLa cells can block 10 μ M ISO (8 h) induced mitophagy signals (n = 5; five biological repeats). Data are presented as the mean \pm SD. One-way ANOVA followed by Dunnett's multiple comparisons test.

To further exclude ISO participates in the activated receptor-mediated mitophagy pathway, we worked on the hypoxia system. We have successfully established this system as evidenced by the high expression of hypoxia-inducible factor 1 α (HIF-1 α) and reduced mitochondrial membrane proteins MFN2 and MTCO2 (Figure EV2C-D). However, ISO did not change the expression of BNIP3 and NIX or FUNDC1 in the hypoxic YFP-Parkin HeLa cells (Figure EV2C), which suggests that ISO has no effects on the receptor-mediated mitophagy. The same experiment was performed in SH-SY5Y cells with similar results (Figure EV2D). Thus, all these data indicate that ISO specifically promotes PINK1-Parkin-dependent mitophagy (new Figure EV2C-D, and below).

Supplementary Figure EV2: ISO promotes mitophagy via mediating PINK1 function. (C) 5 μ M ISO treatment for 24 hours cannot promote hypoxia-induced receptor-mediated mitophagy protein expression in YFP-Parkin HeLa cells. (D) 5 μ M ISO treatment for 24 hours cannot promote hypoxia-induced receptor-mediated mitophagy protein expression in SH-SY5Y cells.

Our data cannot fully exclude the involvement of alternative mitophagy pathways in

the ISO system, and future studies are required to examine whether activation of these pathways can compensate for PINK1 deficiency and provide neuroprotection in ALS models, as outlined in the revised version:

Discussion section

“Furthermore, the present study does not fully address the specific role(s) of other mitophagy pathways in ALS pathology and progression, such as impaired FUNDC1-dependent [74] and BNIP3-dependent mitophagy pathways [31]. Thus, further mechanistic and interventional studies (e.g., with ISO) on other mitophagy pathways within ALS motor neurons are needed.”

3. Stat needs to be seriously improved. Most experiments have been repeated only two to three times; a minimum of five independent biological replicates are mandatory to perform reliable stat analysis.

Response: Thank you for your suggestion. In all the cell experiments, we conducted five independent biological replicates (n=5) for statistical analysis. Please check revised Figures 1-3 as well as revised Figures EV1-2 and their figure legends (in green color) with the source data. We also provide detailed responses to the questions about animal studies and iPSCs studies in the subsequent sections.

4. Introduction and Discussion lack a comprehensive analysis of pertinent literature and an in-depth examination of the study's findings. Both sections would benefit from a more rigorous approach, enhanced clarity in language use, and improved sentence structure.

Response: We have updated the “Introduction” and “Discussion” sections. The logic of the introduction is as follows: background introduction of ALS, mitochondrial dysfunction in ALS as well as the existing mitophagy pathways and the deficiencies in mitophagy research in ALS and the feasibility of using ISO for ALS treatment. The structure of the discussion follows a logical progression: a summary of our findings, highlighting key innovations, acknowledging limitations, and outlining future directions and the feasibility of continued research.

ADDITIONAL POINTS

5. The order used to present the results is not clear. In the first part of the results the authors showed that ISO is able to enhance mitophagy in vitro and they provide a possible molecular mechanism, subsequently in the fourth chapter of the results (Defective PINK1-Parkin-dependent mitophagy in ALS MNs) they showed that mitophagy is impaired in MNs derived from ALS patient, however the results that showed that ISO enhances mitophagy in these models are at the very end, after all the in vivo studies. Consider moving the last chapter after the fourth to improve clarity.

Response: Thanks for the logic. We made some adaptation for the result section according to your suggestion. Now, you can find all results about how ISO effects iPSC-derived motor neurons in the new Figure 5. Then, we demonstrated that ISO could enhance mitophagy and show protective effects in the in vivo models (worms

and mice, new Figure 6 and Figure 7). We hope this new stepwise approach provides a clearer understanding of the rationale of our work.

-6. Please describe in more details in Materials and Methods how the different mitophagy indexes/mitophagy events are calculated.

Response: In the Methods section, we have added two sub-sections to describe the detection methods of mitophagy index in mt-Keima cells, as well as the neuronal mitophagy events in nematodes, mice, patient samples, and iPSC-derived MNs. The detailed information is as follows:

Flow cytometry

Mitophagy index was assessed in mKeima-expressing cells as described previously [37]. Briefly, mt-mKeima HeLa cells were harvested with trypsin and analysed by flow cytometry (BD LSRFortessa X-20) equipped with a 405-nm and 561-nm laser. Lysosomal mito-Keima indicated mitophagy index was detected using dual-excitation ratiometric pH measurements at 405 nm (pH 7) and 561 nm (pH 4) lasers with 695 nm and 670 nm emission filters, respectively. A minimum of 10,000 events were counted and analysed using Diva software (BD Biosciences).

Neuronal mitophagy events detection

Mitophagy was detected in multiple model organisms using several methods as reported previously [19, 83, 84]. Postmortem spinal cord samples of ALS patients and sex-matched healthy controls were obtained from the Netherlands Brain Bank (<https://www.e-nbb.org>, n = 3 per group), and the mitochondrial protein marker MTCO2 and the lysosomal protein marker LAMP2 were detected using immunofluorescence. The Pearson's correlation coefficient of colocalization between MTCO2 and LAMP2 was calculated in NeuN-positive neurons using Image J software (<http://punias.free.fr/ImageJ/colocalization-finder.html>) to assess the mitophagy events. In iPSC-derived MNs models, two methods were employed to assess the level of mitophagy. The first method involves using immunofluorescence to mark the MAP2-positive motor neurons and then analysing the Pearson's correlation coefficient of MTCO2 and LAMP2 proteins to assess the mitophagy events. The second method involves the use of a commercial mitophagy detection kit (Dojindo; MD01). Mtphagy dye is a mitochondrion-localizing pH-sensitive probe. When mitophagy occurs, damaged mitochondria fuse with lysosomes, causing the mtphagy dye to produce a stronger fluorescence. Mitophagy events were evaluated by analysing the average fluorescence intensity of mtphagy dye in *Hb9::GFP* lentivirus-labelled motor neurons. Neuronal mitophagy events were detected in mice using immunofluorescence to mark the ChAT-positive motor neurons and then analyzing the Pearson's correlation coefficient of MTCO2 and LAMP2 proteins to assess the mitophagy events. Neuronal mitophagy in *C. elegans* was measured using two strains. First, an mt-Rosella transgenic worm expressing a pan-neuronal mt-Rosella biosensor that combines a GFP variant sensitive to the acidic environment of the lysosomal lumen, fused to the pH-insensitive DsRed. Mitophagy was calculated as DsRed/DsRed+GFP, thus the higher the ratio of pixel intensity, the higher the level of mitophagy. Second, we

generated transgenic nematodes expressing the DCT-1 mitophagy receptor fused with GFP or BFP, together with the autophagosome marker LGG-1 fused with DsRed in D-type motor neurons or pan-neurons. Mitophagy events were assessed using the colocalization puncta number between LGG-1 and DCT-1.”

7. In Figure 4H the number of LAMP2 puncta in ALS models seems to be lower compared to controls, an effect that appears to be reversed by ISO treatment (Figure 7C). The same effect is also visible in the mouse model (Figure 6L). Could you please comment on this? Does the number of LAMP2 puncta in these models depend on mitophagy levels? Could ISO also affect general autophagy and/or lysosomal biogenesis?

Response: We indeed found lysosomal dysfunction in ALS MNs of different model systems we used. And in our research on the molecular targets of ISO, we discovered that ISO was also actively participating in one lysosomal biogenesis pathway. But this project is still ongoing, and the lysosome dysfunction was not the major focus of the current study, so the relevant data were not presented in this article. For “Does the number of LAMP2 puncta in these models depend on mitophagy levels?”, we think that during the process of mitophagy, ultimately, lysosomes are required to complete the entire degradation process. This is also the principle behind the design of the mt-mKeima and mt-Rosella reporting systems. Therefore, at least a portion of the increase in LAMP2 puncta is related to mitophagy. In the revised Discussion section, we added:

“To note, our unpublished data also shows lysosomal dysfunction in ALS MNs, and whether and how ISO functions in lysosomal biogenesis/homeostasis in ALS MNs are under investigation.”

-8. The DCT-1/LGG-1 and mt-Rosella *C. elegans* models should be better described.

Response: We provided an introduction to these two types of nematodes in the article as follows:

“The ability of ISO to promote mitophagy *in vivo* was investigated using our previously reported transgenic worms expressing either mt-Rosella or DCT-1/LGG-1 [19, 46]. Mt-Rosella transgenic worms were generated by expressing mitochondrial-targeted biosensor protein Rosella, which is fused with GFP and DsRed in worm neurons. Due to the sensitivity of GFP in the acidic lysosomal environment, an increase in the DsRed/DsRed+GFP ratio indicates an increased level of neuronal mitophagy. DCT-1/LGG-1 transgenic worms were generated by expressing the DAF-16/FOXO-controlled germline-tumor affecting-1 (DCT-1) mitophagy receptor fused with GFP together with the autophagosome marker protein LGG-1 fused with DsRed in neurons, and the increased number of the co-localization of DCT-1 and LGG-1 indicates the increased level of neuronal mitophagy.”

-9. Experiments in Figure 5 (from D to J) are performed only on SOD1 G93A transgenic worms. It is important to also add WT as control (treated with vehicle as well as ISO). Please add representative images for quantification in Figure 5J. It is not clear if the KD of Pink-1 and pdr-1 in worm MNs (Figure 5J-K) is performed on WT or SOD1 G93A mutant, please clarify this point and add the appropriate controls (vehicle-Pink-1, vehicle-pdr-1, vehicle-WT, ISO-WT), which are missing in the analysis. Figure S4C-D the higher concentration of ISO used as reported in the text is 15 μ M, correct in the figure.

Response: In Figure 6D-I, we included WT N2 worms as a control. In Figure 6F-H, G93A worms crossed with *unc25::GFP* worms to visualize the connections in motor neurons. N2 crossed with *unc25::GFP* worms were used as the control (updated in main text and Figure legends, in green color). Also, we have added representative images for quantification in Figure 6J. In the main text, we have described that the Figure 6K experiment was completed in SOD1 G93A worms crossed with mt-Rosella and SID-1 worms, and the Figure 6L experiment was completed in SOD1 G93A worms crossed with SID1 worms. At the same time, we have added appropriate control groups. Finally, we have modified the label in Figure EV5C-D as 15 μ M ISO. Because the images are too large, we did not display them here. Please check main text and new Figure 6 and Figure EV5 along with updated figure legends (in green color).

-10. Please clarify in the text that pdr-1 is the homolog of Parkin.

Response: We have added an explanation in the manuscript: "*pdr-1* (*C. elegans* homolog of the mammalian *Parkin*)"

-11. Figure 6D: the y-axis is labeled "Time to disease onset", but it should refer to the percentage of "healthy" mice. Please modify.

Response: Thank you and we modified as suggested (revised Figure 7D, also below).

Figure 7 Protective effects of intranasal administration of Nano-ISO in ALS SOD1 G93A transgenic mice. (D) The onset of tremor in SOD1 G93A mice was delayed in Nano-ISO group, n=12 mice in the Veh group, n=12 mice in the Nano-ISO group, and n=9 mice in the Riluzole group.

-12. A very critical point arises from the data of the experiments performed using

iPSCs-derived MNs (Figure 4H-O and Figure 7). In Figure 4H-O it appears that the 3 cell lines deriving from patients with 3 different mutations are pooled together and used as biological replicates. The same it's true for Figure 7A-C, while in Figure 7E-K the results from the 3 cell lines are kept separate. Moreover, based on the figure legend description, the dots in the quantification images correspond to single images/MNs, they do not represent biological replicates so it is not possible to perform a proper statistical analysis on these data. Please revise all data from Figure 4 and 7 taking into account that to perform a statistical analysis it is required to have three to five biological replicates for each sample, and each biological replicate should come from the analysis of at least 10 images/MNs. Ideally, all data should be presented as in Figure 7F-H-I-K. It might be acceptable to pool together data from the 3 different cell lines and consider them as ALS patients; however it is necessary to increase the number of images analysed and increase the number of biological replicates for each experiment. Finally, quantification and statistical analysis of the results needs to be done on biological replicates not technical/single images.

Response: We sincerely apologize for our improper handling, and we are very grateful for your suggestions. We re-analyzed the data in Figure 4, new Figure 5 and Figure EV4.

The presence of patient samples includes various possible pathogenesis genes. Therefore, three types of ALS patient iPSCs derived motor neurons were employed as the ALS models. We increased the number of images analyzed and the number of biological repetitions as required. Each point represents the average value of 10 MNs for each iPSC-derived MNs; each group contains three biological replicates of three types of MNs. Also, we have revised the figure legends for Figure 4H-Q. Please check the legends of Figure 4 (in green color) as well as the source data we provided. We added:

In Figure 4 legend:

“Each point represents the average value of 10 MNs for each iPSC-derived MNs; each group contains three biological replicates of three types of MNs.”

In new Figure 5 and Figure EV4, to clearly demonstrate the protection effect of ISO in the three ALS patient iPSC-derived MNs, we performed additional experiments based on the original Figure 7F-H-I-K and re-performed the statistical analysis. ALS C9, SOD1, TDP43 MNs were evaluated separately. In the detection of mitophagy and the detection of pSer65-Ub expression levels (new Figure 5A-F), each point represents the average value of 10 MNs for each iPSC-derived MNs; three biological replicates for each group. In the detection of neurites swelling and the *Hb9*:GFP area (new Figure 5I-M), each point represents the average value of 10 images for each iPSC-derived MNs; three biological replicates for each group. Also, we have revised the figure legends for Figure 5 and Figure EV4. Please check main text, figures and the legends of Figure 5 and Figure EV4 (in green color) as well as the source data we provided. We added:

In Figure 5 and Figure EV4 legend:

“Each point represents the average value of 10 MNs for each iPSC-derived MNs; three biological replicates for each group.” Or “Each point represents the average value of 10 images for each iPSC-derived MNs; three biological replicates for each group.”

-13. The same critical point applies to some of the experiments performed in mice (Figure 6G-O and Figure S6E-J). In Figure 6G-H, each data point represents a single individual (n = 5 mice) and thus they can be considered biological replicates. In Figure 6I-O and S6E-J, however, the data points (n = 10-20) appear to refer to single images/MNs derived from n = 5 mice or n = 3 mice, so they do not represent biological replicates. Quantification and statistical analysis of data from Figure 6I-O and S6E-J need to be performed on the biological replicates (as in Figure 6G-H). Figure legends should indicate how many images/MNs were quantified per biological replicate.

Response: We have updated the data and statistical analysis for new Figure 7 and Figure EV7. To evaluate mitophagy in motor neurons (Figure 7L-M) as well as the pSer65-Ub expression (Figure 7N-O) staining, each point represents the average value of 10 MNs for each mouse, and n = 5 mice were used as biological replicates for statistical analysis. In the Iba-1, GFAP, NeuN and ChAT staining (Figure 7G-K, Figure EV7E-J), each point represents the average value of 10 images for each mouse, and n = 5 mice were used as biological replicates for statistical analysis. Also, in the figure legends of Figure 7 and Figure EV7, we have presented the number of images/MNs quantified per biological replicate for each figure. Please check the figures and legends of Figure 7 and Figure EV7 (in green color) as well as the source data we provided. We added:

In Figure 7 and Figure EV7 legend:

“Each point represents the average value of 10 images for each mouse, n=5 mice/group.” Or “Each point represents the average value of 10 MNs for each mouse, n=5 mice/group.”

Referee #3 (Comments on Novelty/Model System for Author):

while the identification of ISO as a mitophagy enhancer with therapeutic potential in ALS is exciting, the current dataset lacks several key functional validations. More rigorous experimentation is required to establish mitochondrial integrity, validate mitophagy enhancement, and clarify ISO's precise molecular mechanism.

Response: Thank you very much for your comments. We have conducted more experiments to further refine our entire research.

Referee #3 (Remarks for Author):

The study conducted by Dr. Ang Li identifies Isoginkgetin (ISO), a bioflavonoid derived from Ginkgo biloba, as a potent inducer of mitophagy. ISO exerts its effects primarily through the enhancement of the PINK1-Parkin pathway and stabilization of the PINK1/TOM complex. The authors further demonstrate that ISO mitigates ALS-related pathology in both cell and animal models by rescuing defective mitophagy and enhancing mitochondrial function. These findings are certainly compelling and indicate a potential therapeutic avenue for ALS. However, while the conclusions are promising, there are several critical concerns that must be addressed to validate the claims and strengthen the mechanistic insights presented.

1. Foremost, the manuscript emphasizes that ISO promotes mitophagy "without compromising the normal function of mitochondria." Unfortunately, this claim is not adequately substantiated, as no direct mitochondrial function assays are provided. While quantification of mitochondrial membrane potential (MMP) or MTCO₂ expression provides limited information, these metrics alone are insufficient to draw definitive conclusions about mitochondrial health or regulation. More comprehensive assessments are needed, such as Mitochondrial respiration analysis (e.g., Seahorse assay) and Assessment of mitochondrial biogenesis content via mtDNA copy number

Response: Based on your important suggestions, we performed two extra experiments, including mitochondrial functional analysis using a seahorse machine and the detection on the changes of mtDNA copy numbers, to investigate whether ISO affected mitochondrial function. We found that ISO did not change oxygen consumption rate or mtDNA copy number in HeLa cells. Our data indicated that ISO did not affect the mitochondrial function in HeLa cells (new Figure 3J-M, also below).

Figure 3: ISO stabilizes the PINK1/TOM complex and sustains the active form of PINK1. (J) Seahorse was used to detect the OCR within the ISO treatment group, 5 μ M ISO does not impair mitochondrial respiration ($n=5$). (K) Based on (J), calculations of the maximum respiration values reached by the cells in the ISO and DMSO groups during the FCCP depolarization process, 5 μ M ISO does not impair mitochondrial respiration ($n=5$). (L) QRT-PCR was used to detect the mitochondrial DNA copy number within the ISO treatment group, 5 μ M ISO does not decrease the mtDNA copy number ($n=5$). (M) Flow cytometry was used to detect TMRE-labelled MMP within the ISO treatment group, 5 μ M ISO does not cause the collapse of MMP ($n=5$).

In the main text:

“Combining our analysis within the ISO treatment group, through the Seahorse oxygen consumption rate detection (Figure 3J-K), as well as the changes in mitochondrial DNA copy number (Figure 3L) and mitochondrial membrane potential (Figure 3M), we summarize that ISO promotes mitophagy by stabilizing the PINK1/TOM complex on the OMM without compromising the normal function of mitochondria.”

2. The authors report that ISO at concentrations up to 10 μ M maintains MMP and enhances mitophagy in cells treated with CCCP, a mitochondrial uncoupler. While these findings are notable, it is equally important to evaluate whether ISO can promote mitophagy in basal conditions (i.e., in the absence of CCCP or other stressors), which would provide insight into its potential as a physiological modulator of mitochondrial quality control.

Response: Thank you for your suggestion. In fact, during the process of screening for ISO (Figure 1), we discovered that prolonged treatment with high concentrations of ISO (10 μ M) can induce mitophagy. Our article consists of two parts. First, high concentration of ISO (10 μ M) can induce mitophagy directly (Figure 1); Second, low concentration of ISO (5 μ M) did not induce mitophagy in our system but could facilitate mitophagy induction in the conditions of external mitochondrial stress (e.g., with CCCP treatment, Figure 2). The usage of ISO is to treat ALS, which has damaged motor neurons with accumulated dysfunctional mitochondria. To mimic the condition, we treated mt-mKeima HeLa cells with CCCP to induce mitochondrial damage for the following mechanistic studies. This model has been reported and applied in the previous study (<https://doi.org/10.1016/j.cell.2020.04.025>). Furthermore, our study demonstrated that ISO protects ALS MNs in varied ALS models via

eliminating damaged mitochondria via PINK1-Parkin-dependent mitophagy.

3. Additionally, the role of mitophagy in ALS remains highly debated. While multiple studies report impaired PINK1/Parkin signaling and reduced MMP in ALS cellular models, some patient-derived cells exhibit increased mitophagy activity and hyperpolarized mitochondria. Furthermore, emerging data suggest that mitochondrial dysfunction in ALS may not be a static phenotype, but rather a progressive and heterogeneous process that evolves over the course of the disease. Given these complexities, mitophagy defects in ALS models should be corroborated using a broader range of quantitative assays. Co-localization of MTCO2 and LAMP2, while informative, lacks the resolution and specificity required for definitive mitophagy quantification. More robust assays are necessary to firmly establish mitophagy dynamics and validate ISO's mode of action.

Response: We agree with the reviewer's comments. We propose that compensatory or self-protective mechanisms may drive dynamic changes in the mitophagy signaling pathway, and that mitochondrial ion channels and membrane potential may also change during ALS progression. However, we need to point out that in our study, the patient samples represent the terminal stage of disease progression. Similarly, in patient iPSC-derived MNs, the detection of mitophagy levels and changes in mitochondrial membrane potential was conducted after the appearance of neurites beads-like structures (one of the disease phenotypes of ALS motor neurons, 10.1016/j.stem.2014.02.004). Therefore, we propose that, once the disease phenotype emerges in ALS MNs, dysfunction across multiple organelles, including mitochondria—leads to impaired mitophagy and a decline in mitochondrial membrane potential, as observed in our study. To better explain this issue, we updated the “Discussion” section and proposed the dynamic process of mitophagy and changes in mitochondrial membrane potential within ALS, as well as related research that can be conducted in the future.

In the main text, we present the time points at which we measured the levels of mitophagy and mitochondrial membrane potential.

“Differentiated MNs were cultured for 28 days to allow full maturation, at which point disease associated phenotypes, such as swollen neurites, were observed in the ALS-iPSC-derived MNs (Figure EV3B-C, white arrow).”

In the Discussion, we made some additions.

“Mitophagy levels and the alterations in mitochondrial membrane potential varied in different pathogenic genotypes of ALS and at different stages of disease progression [31]. Due to practical challenges, we did not systematically monitor mitophagy dynamics throughout the entire progression of ALS in our models. However, it has been reported that mitophagy may be higher in the early stages of ALS, and due to various reasons such as mitochondrial permeability changes, hyperpolarized mitochondria may also appear [69-71]. It has become possible to measure mitophagy *in vivo* through models such as mt-mKeima mice [72]: crossing this mitophagy

reporting mouse model with an ALS mouse strain will enable the recording of tissue- and cell type-specific changes in mitophagy across the ALS disease continuum.”

Secondly, regarding your suggestions, we have updated the co-localization staining results of LAMP2/MTOC2 in the ALS SOD1 G93A mice model (new Figure 7L-M, also below) and the ALS patient iPSC-derived MNs model (Figure 4H-I, also below). Additionally, in the ALS iPSC-derived MNs model, Mitophagy dye (Dojindo; MD01), a commercial dye for detecting mitophagy levels was used, and live-cell imaging was performed within *Hb9::GFP*⁺ motor neurons to elucidate the changes in mitophagy (new Figure 4J-K, also below) and the ability of ISO to activate mitophagy (new Figure 5E-F, also below).

Figure 7 Protective effects of intranasal administration of Nano-ISO in ALS SOD1 G93A transgenic mice. (L) Representative images of MTCO2 and LAMP2 co-localization levels in lumbar ChAT⁺ MNs. Scale bars, 5 μ m. (M) Quantification of (L) MTCO2 and LAMP2 colocalization using Pearson's correlation coefficient showed that ISO enhanced mitophagy in lumbar ChAT⁺ MNs. Each point represents the average value of 10 MNs for each mouse, n=5 mice/group.

Figure 4: Accumulation of dysfunctional mitochondria and impaired

PINK1-Parkin-dependent mitophagy in ALS MNs. (H) Representative images of colocalization of MTCO2 and LAMP2 in three ALS patient iPSC lines derived MNs (MAP2 staining) and three healthy controls. Scale bars, 10 μ m. (I) Quantification of (H) MTCO2 and LAMP2 co-localization using the Pearson's correlation coefficient demonstrating that the basal level of mitophagy was decreased in three ALS patient iPSC derived MAP2⁺ MNs. Each point represents the average value of 10 MNs for each iPSC-derived MNs; each group contains three biological replicates of three types of MNs. (J) Representative living cell images of the mtpagy dye in three ALS patient iPSC derived MNs (*Hb9::GFP*⁺) and three healthy controls. Scale bars, 10 μ m. (K) Quantification of (J) the fluorescence intensity demonstrating that the basal level of mitophagy was decreased in three ALS patient iPSC derived *Hb9::GFP*⁺ MNs. Each point represents the average value of 10 MNs for each iPSC-derived MN; each group contains three biological replicates of three types of MNs.

We also describe in the main text:

“We found that the co-localization frequency of LAMP2 and MTCO2 in the ALS MAP2⁺ motor neurons was 40.3% lower than that in the healthy control group, indicating a mitophagy disorder (Figure 4H-I). To further explore the mitophagy levels in ALS MNs, we used a commercial mitophagy dye (mtpagy) to conduct live-cell imaging within *Hb9::GFP*-positive motor neurons. We found that the mitophagy levels indicated by the mtpagy dye in ALS MNs were also 47% lower than those in the healthy control group, further indicating the presence of damaged mitophagy in ALS MNs (Figure 4J-K).”

Figure 5 ISO improves ALS-related phenotypes in three ALS iPSC lines derived MNs by promoting PINK1-dependent mitophagy. (E) Representative living cell images of the mtpagy dye in three ALS patient iPSC derived MNs (*Hb9::GFP*⁺). Scale bars, 10 μ m. (F) Quantification of (E) the fluorescence intensity demonstrating that ISO enhanced mitophagy in three ALS patient iPSC derived *Hb9::GFP*⁺ MNs. Each point represents the average value of 10 MNs for each iPSC-derived MNs; three biological replicates for each group.

In the main text:

“Increased mtpagy dye signals indicated improved mitophagy in *Hb9::GFP*⁺ MNs (Figure 5E and F).”

Other questions:

4. The manuscript also notes a lack of LC3-II accumulation in cells treated with 5 μ M CCCP despite observed Parkin recruitment. This raises important questions about the downstream autophagic flux. Is ISO uncoupling Parkin recruitment from canonical autophagosome formation, or is there an alternative mitophagy pathway being activated? Furthermore, the role of ISO in modulating Parkin stability or recruitment efficiency in the context of wild-type versus mutant PINK1 remains unexplored.

Response: We have performed additional experiments to explain why CCCP (5 μ M for 4 hours) failed to induce the accumulation of LC3-II (Figure 2D). We believe that caused by the rapid translocation of YFP-Parkin in response to CCCP-induced mitochondrial membrane potential loss in YFP-Parkin HeLa cells. Although CCCP is known to activate PINK1–Parkin–dependent mitophagy, leading to LC3-II accumulation under typical conditions, the rapid dynamics in our model may have limited our ability to capture this accumulation within the treatment window. Therefore, we extended the treatment time of CCCP and conducted repeated experiments using the YFP-Parkin HeLa and SH-SY5Y cell line. In Figure EV2A, after a long period of CCCP treatment (8 h), we found that CCCP could induce the accumulation of LC3-II. Furthermore, in the SH-SY5Y cells, when treated with 5 μ M CCCP for 4 hours, a significant accumulation of LC3-II could be observed (new Figure EV2B, also below).

Supplementary Figure EV2: ISO promotes mitophagy via mediating PINK1 function. (A) 5 μ M ISO treatment for 8 hours can promote CCCP-induced PINK1-Parkin-dependent mitophagy protein expression, mitochondrial membrane protein degradation and LC3 lipidation in YFP-Parkin HeLa cells. (B) 5 μ M ISO treatment for 4 hours can promote CCCP-induced PINK1-Parkin-dependent mitophagy protein expression, mitochondrial membrane protein degradation and LC3 lipidation in SH-SY5Y cells.

Secondly, we believe that ISO enhances PINK1 activity by stabilizing the PINK1/TOM

complex, thereby promoting the recruitment of downstream Parkin proteins. This indicates ISO is involved in mediating the canonical PINK1-Parkin-dependent mitophagy pathway (as reported in this article: 10.1038/nature14893). Also, for the exploration of other mitophagy pathways, in line with the first and second reviewers, we designed experiments to explore whether 10 μ M ISO could induce receptor-mediated mitophagy, and whether 5 μ M ISO could promote hypoxia-induced receptor-mediated mitophagy in Figures 1F, Figures EV1G-I, and Figures EV2C-D, we ruled out the possibility that ISO can induce receptor-mediated mitophagy.

Finally, since the stability and recruitment of Parkin associated with the PINK1 mutant are not the focus of this study, we did not investigate the effect of ISO on the response of Parkin in WT PINK1 and mutant PINK1. However, we added one experiment in Figure EV2F that the Parkin translocation induced by CCCP, which was enhanced by ISO, could be significantly inhibited when we used PRT062607 (a recently reported potent inhibitor of PINK1).

Supplementary Figure EV2: ISO promotes mitophagy via mediating PINK1 function. (E) Immunoblotting showed that PRT, a PINK1 inhibitor, prevented the enhancing effect of ISO on CCCP-induced pSer65-Ub expression and MFN2 degradation, and Parkin translocation (F).

22nd Sep 2025

Dear Dr. Su,

Thank you for the submission of your revised manuscript to EMBO Molecular Medicine. I am pleased to inform you that we will be able to accept your manuscript pending the following final amendments:

1) In the main manuscript file, please do the following:

- Please address all comments suggested by our data editors listed below:

o Figure legends:

1. Please note that the exact p values are not provided in the legends of figures 1I, 2B, 3E, 4K, , N, O, Q; 5B, D, F, H, J; 5B, D, F, H, J; 6G, K, L; 7J, K, M; EV1 F, I; EV4A, C, G; EV5 A, C, D, F.

2. Please note that information related to n is missing in the legend of figure 1E.

3. Please note that the white arrows are not defined in the legends of figures 5C, E; 6B, EV5 B, EV7 F. This needs to be rectified.

4. Please note that the white dotted borders are not defined in the legend of figures 4J, L; 5A, E. This needs to be rectified.

- Remove data not shown (twice on p.9).

- In Methods, provide the statement that informed consent was obtained from all human subjects (next of kin) and that the experiments conformed to the principles set out in the WMA Declaration of Helsinki and the Department of Health and Human Services Belmont Report.

- In Methods, add a paragraph describing the bioinformatic analysis of published datasets GSE76220 and GSE115130. Datasets should be adequately cited here incl. the publications.

- In Methods, provide antibody dilutions that were used for each antibody.

- In Methods, add the following paragraph:

Graphics:

(some of the... OR Figure #... OR synopsis) Graphics were created with BioRender.com.

- Indicate in legends exact n and exact p values, not a range, along with the statistical test used. To keep the figures "clear" some authors found providing an Appendix table Sx with all exact p-values preferable. You are welcome to do this if you want to.

- Author contributions: Please remove it from the manuscript and specify author contributions in our submission system. CRediT has replaced the traditional author contributions section because it offers a systematic machine-readable author contributions format that allows for more effective research assessment. You are encouraged to use the free text boxes beneath each contributing author's name to add specific details on the author's contribution. More information is available in our guide to authors:

<https://www.embopress.org/page/journal/17574684/authorguide#authorshipguidelines>

- Data availability statement should contain information only for the raw data generated in this study and deposited in public databases. Please remove information about data generated in other studies. GSE76220 and GSE115130 should be cited at an appropriate place in Methods. If no data are deposited in public repositories, please replace current text in data availability statement with the following sentence: This study includes no data deposited in external repositories.

Please check "Author Guidelines" for more information.

<https://www.embopress.org/page/journal/17574684/authorguide#availabilityofpublishedmaterial>

- Correct the reference citation in the text and reference list. In the text a reference should be cited by author and year of publication. Include a space between a word and the opening parenthesis of the reference that follows. In the reference list, citations should be listed in alphabetical order. Where there are more than 10 authors on a paper, 10 will be listed, followed by "et al.". Also, please remove DOIs. Please check "Author Guidelines" for more information.

<https://www.embopress.org/page/journal/17574684/authorguide#referencesformat>

2) Tables: Please add a legend of Dataset EV1 to the excel file in a separate tab/worksheet and for Table EV1 at the top of the page.

3) Source data: Please upload the source data for Fig EV1 and EV2 as two separate files.

4) Synopsis:

- Synopsis text: In addition to bullet points please provide a short standfirst (maximum of 300 characters, including space).

Please check recently published papers for reference <https://www.embopress.org/journal/17574684>.

- Synopsis image: Please resize the image to 550 px-wide x 300-600 pixels high and upload it as a high-resolution jpeg file. Also, please increase the font of the text on the image to retain readability.

5) As part of the EMBO Publications transparent editorial process initiative (see our Editorial at

<http://embomolmed.embopress.org/content/2/9/329>), EMBO Molecular Medicine will publish online a Review Process File (RPF) to accompany accepted manuscripts. This file will be published in conjunction with your paper and will include the anonymous referee reports, your point-by-point response and all pertinent correspondence relating to the manuscript. Let us know whether you agree with the publication of the RPF and as here, if you want to remove or not any figures from it prior to publication.

6) Please provide a point-by-point letter INCLUDING my comments as well as the reviewer's reports and your detailed responses (as Word file).

I look forward to reading a new revised version of your manuscript as soon as possible.

Yours sincerely,

Zeljko Durdevic

Zeljko Durdevic
Senior Editor
EMBO Molecular Medicine

*** Instructions to submit your revised manuscript ***

1) a .docx formatted version of the manuscript text (including Figure legends and tables)

2) Separate figure files*

3) supplemental information as Expanded View and/or Appendix. Please carefully check the authors guidelines for formatting Expanded view and Appendix figures and tables at <https://www.embopress.org/page/journal/17574684/authorguide#expandedview>

4) a letter INCLUDING the reviewer's reports and your detailed responses to their comments (as Word file).

5) The paper explained: EMBO Molecular Medicine articles are accompanied by a summary of the articles to emphasize the major findings in the paper and their medical implications for the non-specialist reader. Please provide a draft summary of your article highlighting

6) Author contributions: the contribution of every author must be detailed in a separate section.

7) EMBO Molecular Medicine now requires a complete author checklist (<https://www.embopress.org/page/journal/17574684/authorguide>) to be submitted with all revised manuscripts. Please use the checklist as guideline for the sort of information we need WITHIN the manuscript. The checklist should only be filled with page numbers where the information can be found. This is particularly important for animal reporting, antibody dilutions (missing) and exact values and n that should be indicated instead of a range.

8) Every published paper now includes a 'Synopsis' to further enhance discoverability. Synopses are displayed on the journal webpage and are freely accessible to all readers. They include a short stand first (maximum of 300 characters, including space) as well as 2-5 one sentence bullet points that summarise the paper. Please write the bullet points to summarise the key NEW findings. They should be designed to be complementary to the abstract - i.e. not repeat the same text. We encourage inclusion of key acronyms and quantitative information (maximum of 30 words / bullet point). Please use the passive voice. Please attach these in a separate file or send them by email, we will incorporate them accordingly.

You are also welcome to suggest a striking image or visual abstract to illustrate your article. If you do please provide a jpeg file 550 px-wide x 300-600px high.

9) A Conflict of Interest statement should be provided in the main text

10) Please note that we now mandate that all corresponding authors list an ORCID digital identifier. This takes <90 seconds to complete. We encourage all authors to supply an ORCID identifier, which will be linked to their name for unambiguous name identification.

Currently, our records indicate that the ORCID for your account is 0000-0003-3254-825X.

Link Not Available

11) Include a Reagents and Tools Table as part of the Methods section, which can be downloaded from our author guidelines (<https://www.embopress.org/page/journal/17574684/authorguide#structuredmethods>)

Photos 400-800 DPI

*Additional important information regarding figures and illustrations can be found at <https://bit.ly/EMBOPressFigurePreparationGuideline>. See also figure legend preparation guidelines: <https://www.embopress.org/page/journal/17574684/authorguide#figureformat>

***** Reviewer's comments *****

Referee #2 (Comments on Novelty/Model System for Author):

Authors have addressed all the concerns we raised. The manuscript is of adequate technical quality to deserve publication.

Point-by-point letter

1) In the main manuscript file, please do the following:

- Please address all comments suggested by our data editors listed below:

Figure legends:

1. Please note that the exact p values are not provided in the legends of figures 1I, 2B, 3E, 4K, N, O, Q; 5B, D, F, H, J; 5B, D, F, H, J; 6G, K, L; 7J, K, M; EV1 F, I; EV4A, C, G; EV5 A, C, D, F.

Response: Thank you very much for your suggestion. As recommended below, to keep the figures "clear", we have provided an Appendix Table S1 with a title "Appendix Table S1 Exact *P* values" containing the information about all exact *P*-values and statistical methods used. Meanwhile, we have added the information "Exact *P* values are reported in Appendix Table S1" in the revise legends of each indicated figure.

2. Please note that information related to n is missing in the legend of figure 1E.

Response: We have added the information related to n in the legend of figure 1E. The figure legend for Figure 1E reads: "YFP-Parkin HeLa cells were treated with ISO at various concentrations for 24 h. CCK8 assay was used to detect cell viability (n = 4; four biological repeats)."

3. Please note that the white arrows are not defined in the legends of figures 5C, E; 6B, EV5 B, EV7 F. This needs to be rectified.

Response: Thank you so much for your suggestion. We have defined the white arrows that are present in each figure in the revised legends. We have noted that a number of figures mentioned by the data editor do not contain the white arrows such as Figures 5C, E and EV5B. For clarity, we provide details on defining the white arrows in each individual images as below:

In the legend of Figure 4C, we have added: "NeuN⁺ motor neurons with reduced co-localization of MTOC2 and LAMP2 are marked with white arrows."

In the legend of Figure 4E, we have added: "NeuN⁺ motor neurons with reduced pSer65-Ub expression are marked with the white arrow."

In the legend of Figure 6B, we have added: "*unc25*⁺ D-type motor neurons are marked with white arrows."

In the legend of Figure EV7F we have added: "White arrows indicate motor neurons with shrunken cell bodies."

4. Please note that the white dotted borders are not defined in the legend of figures 4J, L; 5A, E. This needs to be rectified.

Response: Thank you for your suggestion. We have defined the white dotted borders present in each figure in the revised legends.

In the legend of Figure 4J, we have added: "*Hb9*::GFP⁺ motor neurons are marked with white dotted borders."

In the legend of Figure 4L, we have added: “MAP2⁺ motor neurons are marked with white dotted borders.”

In the legend of Figure 5A, we have added: “MAP2⁺ motor neurons are marked with white dotted borders.”

In the legend of Figure 5E, we have added: “*Hb9::GFP*⁺ motor neurons are marked with white dotted borders.”

In the legend of Figure 7G, we have added: “The ventral horn of the spinal cord is marked with white dotted borders.”

In the legend of Figure 7L, we have added: “ChAT⁺ motor neurons are marked with white dotted borders.”

In the legend of Figure 7N, we have added: “ChAT⁺ motor neurons are marked with white dotted borders.”

In the legend of Figure EV4B, we have added: “MAP2⁺ motor neurons are marked with white dotted borders.”

In the legend of Figure EV5B, we have added: “*unc119*⁺ neurons are marked with white dotted borders.”

- Remove data not shown (twice on p.9).

Response: We have removed twice "data not shown" from the manuscript.

- In Methods, provide the statement that informed consent was obtained from all human subjects (next of kin) and that the experiments conformed to the principles set out in the WMA Declaration of Helsinki and the Department of Health and Human Services Belmont Report.

Response: In the "Postmortem ALS Patient Samples" section of the Methods, we have added the following statement: “The informed consent was obtained from all human subjects (next of kin) and that the experiments conformed to the principles set out in the WMA Declaration of Helsinki and the Department of Health and Human Services Belmont Report.”

- In Methods, add a paragraph describing the bioinformatic analysis of published datasets GSE76220 and GSE115130. Datasets should be adequately cited here incl. the publications.

Response: We have added a paragraph describing the bioinformatic analysis of published datasets GSE76220 and GSE115130 in the "Differential Analysis" section of the Methods, along with the corresponding publications in the revision: “A total of 31 LCM-seq samples (including 23 spinal motor neuron samples from ALS patients and 8 from healthy controls) were used in this study; the relevant data were downloaded from the GEO database under accession numbers GSE76220 and GSE115130 (Krach *et al.*, 2018; Nizzardo *et al.*, 2020).”

- In Methods, provide antibody dilutions that were used for each antibody.

Response: Referring to the recent online article in EMBO Molecular Medicine (<https://www.embopress.org/doi/full/10.1038/s44321-025-00311-6#sec-4>), we have

provided antibody dilutions for each antibody in the "Reagents_Tools_Table".

- In Methods, add the following paragraph:

Graphics:

(some of the... OR Figure #... OR synopsis) Graphics were created with [BioRender.com](https://www.biorender.com).

Response: We have added the following paragraph about Graphics in Methods.

"Graphics including synopsis, Figure 7A, Figure EV6C, and Figure EV7K were created with [BioRender.com](https://www.biorender.com)."

- Indicate in legends exact n and exact p values, not a range, along with the statistical test used. To keep the figures "clear" some authors found providing an Appendix table Sx with all exact p-values preferably. You are welcome to do this if you want to.

Response: The exact n and exact *P* values, not a range, along with the statistical test used are provided in the revised figure legends. To keep the figures "clear", we have provided an Appendix Table S1 with a title "Exact P values are reported in Appendix Table S1" containing all exact *P*-values and statistical methods used.

- Author contributions: Please remove it from the manuscript and specify author contributions in our submission system. CRediT has replaced the traditional author contributions section because it offers a systematic machine-readable author contributions format that allows for more effective research assessment. You are encouraged to use the free text boxes beneath each contributing author's name to add specific details on the author's contribution. More information is available in our guide [to authors: https://www.embopress.org/page/journal/17574684/authorguide#authorshipguidelines](https://www.embopress.org/page/journal/17574684/authorguide#authorshipguidelines)

Response: Thank you for your suggestion. We have removed the "Author Contributions" section from the manuscript.

- Data availability statement should contain information only for the raw data generated in this study and deposited in public databases. Please remove information about data generated in other studies. GSE76220 and GSE115130 should be cited at an appropriate place in Methods. If no data are deposited in public repositories, please replace current text in data availability statement with the following sentence: This study includes no data deposited in external repositories.

Please check "Author Guidelines" for more information.

<https://www.embopress.org/page/journal/17574684/authorguide#availabilityofpublishingmaterial>

Response: Thank you for your suggestion. We have described and cited the GSE76220 and GSE115130 datasets in the Methods and revised the Data Availability Statement as below: "This study includes no data deposited in external repositories."

- Correct the reference citation in the text and reference list. In the text a reference should be cited by author and year of publication. Include a space between a word and the opening parenthesis of the reference that follows. In the reference list, citations should be listed in alphabetical order. Where there are more than 10 authors on a paper, 10 will be listed, followed by "et al.". Also, please remove DOIs. Please check "Author Guidelines" for more information.

<https://www.embopress.org/page/journal/17574684/authorguide#referencesformat>

Response: Thank you for your reminder. We have corrected the reference citation formats in the text and the reference list.

2) Tables: Please add a legend of Dataset EV1 to the excel file in a separate tab/worksheet and for Table EV1 at the top of the page.

Response: We have created a sheet named "Legend of Dataset EV1" in Dataset EV1 and added the legend for Dataset EV1 into it. Similarly, we have added the legend for Table EV1 at the top of the page Table EV1.

3) Source data: Please upload the source data for Fig EV1 and EV2 as two separate files.

Response: We have uploaded the source data for Fig EV1 and EV2 as two separate files.

4) Synopsis:

- Synopsis text: In addition to bullet points please provide a short standfirst (maximum of 300 characters, including space). Please check recently published papers for reference <https://www.embopress.org/journal/17574684>.

- Synopsis image: Please resize the image to 550 px-wide x 300-600 pixels high and upload it as a high-resolution jpeg file. Also, please increase the font of the text on the image to retain readability.

Response: We have uploaded the synopsis text (as Word file, 280 characters, including space) and synopsis image (550 px-wide x 474 pixels high, as jpeg file).

5) As part of the EMBO Publications transparent editorial process initiative (see our Editorial at <http://embomolmed.embopress.org/content/2/9/329>), EMBO Molecular Medicine will publish online a Review Process File (RPF) to accompany accepted manuscripts. This file will be published in conjunction with your paper and will include the anonymous referee reports, your point-by-point response and all pertinent correspondence relating to the manuscript. Let us know whether you agree with the publication of the RPF and as here, if you want to remove or not any figures from it prior to publication. Please note that the Authors checklist will be published at the end of the RPF.

Response: Thank you. We agree with the publication of the RPF, and no figures need to be removed from it.

6) Please provide a point-by-point letter INCLUDING my comments as well as the reviewer's reports and your detailed responses (as Word file).

Response: Thank you. A point-by-point letter including your comments as well as the reviewer's reports (as Word file) is provided.

Referee #2 (Comments on Novelty/Model System for Author):

Authors have addressed all the concerns we raised. The manuscript is of adequate technical quality to deserve publication.

Response: We are really appreciated with your positive comments.

1st Oct 2025

Dear Dr. Su,

We are pleased to inform you that your manuscript is accepted for publication and is now being sent to our publisher to be included in the next available issue of EMBO Molecular Medicine.

Zeljko Durdevic
Senior Editor
EMBO Molecular Medicine
